# Why Is the Counterintuitive Phenomenon of Likelihood Rare in Tabular Anomaly Detection with Deep Generative Models?

## Abstract

Deep generative models with tractable and analytically computable likelihoods, exemplified by normalizing flows, offer an effective basis for anomaly detection through likelihood-based scoring. We demonstrate that, unlike in the image domain where deep generative models frequently assign higher likelihoods to anomalous data, such counterintuitive behavior occurs far less often in tabular settings. We first introduce a domain-agnostic formulation that enables consistent detection and evaluation of the counterintuitive phenomenon, addressing the absence of precise definition. Through extensive experiments on 47 tabular datasets and 10 CV/NLP embedding datasets in ADBench, benchmarked against 12 baseline models, we demonstrate that the phenomenon, as defined, is consistently rare in general tabular data. We further investigate this phenomenon from both theoretical and empirical perspectives, focusing on the roles of data dimensionality and feature correlation difference. We find that likelihood-only detection with normalizing flows offers a practical and reliable approach for anomaly detection in tabular domains.

## 1 Introduction

Generative models, including variational autoencoders (VAEs) (Kingma & Welling, 2014), normalizing flows (NFs) (Dinh et al., 2015), and generative adversarial networks (GANs) (Goodfellow et al., 2020), are widely used to model complex data distributions across diverse applications such as industrial diagnostics, medical imaging, and financial risk assessment. Among these, normalizing flows are particularly well-suited for anomaly detection due to their ability to compute estimated likelihoods, providing a straightforward mechanism for detecting out-of-distribution (OOD) samples. [1] The simplest approach of anomaly detection using normalizing flows is to assume that normal data $\mathbf{x} \in \mathbb{R}^d$ follows the distribution $P$ of normal data, and anomalous data $\mathbf{x}' \in \mathbb{R}^d$ follows a distribution $Q \neq P$, and to determine that a given data $\mathbf{x}_{test} \in \mathbb{R}^d$ is an anomaly if its likelihood $\phi_P(\mathbf{x})$ is lower than a predefined threshold $\alpha$ when tested. We refer to this methodology using normalizing flow as NF-SLT (Normalizing Flow with Simple Likelihood Test).

This methodology is based on the intuition that anomalous data are less likely to be observed in the distribution of normal data. However, the image domain illustrates that in-distribution data utilized as training data in models that can obtain the likelihood of the input data indirectly or directly, such as normalizing flow, exhibit similar or even lower likelihoods than out-of-distribution data. Nalisnick et al. (2019a) demonstrates that when CIFAR-10 (Alex, 2009) is used as training data (In-distribution) and SVHN (Netzer et al., 2011) is used as the test data (Out-of-distribution) of a model that can obtain the likelihood of input data, SVHN has a higher likelihood than CIFAR-10. This is counterintuitive because the likelihood of OOD data is higher than that of in-distribution data. Therefore, it can be inferred that if anomaly detection is performed using only the likelihood of the input data, detection may fail in certain cases (i.e., occurrence of a counterintuitive phenomenon). Refer to Section 2.2 for more details about counterintuitive phenomenon.

However, the following question arises: does this phenomenon also occur in tabular data anomaly detection? Kirichenko et al. (2020) demonstrates that although the likelihood of in-distribution/OOD

---

[1]Although the two tasks slightly differ, we consider OOD detection and anomaly detection to be the same task, and we will utilize the term anomaly detection. Task definitions are presented in Appendix A.

data overlaps for the normalizing flow in the tabular data anomaly detection, it is limited by the fact that only two datasets are shown by setting each as in-distribution data/OOD data. In addition, there is no comparison with other comparison models. A common argument is that assigning likelihoods higher than that of normal data to anomalies is sufficient to demonstrate a conterintuitive phenomenon. Regardless, the interpretation has its limitations. First, the view is contradictory since the argument would consider any result outside 100% AUROC as counterintuitive. Second, likelihood inversion can arise from intrinsic dataset difficulty, for example, when normal and abnormal samples are hard to distinguish, rather than from the phenomenon itself.

This calls for a more sophisticated approach for determining counterintuitive phenomenon, such as by comparing the generative model's performance against other models (e.g., DeepSVDD, OCSVM), as limitation arises when simple approach (such as direct comparison between abnormal/normal data's likelihood) is made. Hence, it is not yet identified whether the counterintuitive likelihood phenomenon occurs in the tabular domain. To address this gap, for currently vaguely-defined counterintuitive phenomenon, we first propose a clearer definition based on the observation in likelihood-based tests for models with estimated likelihoods, allowing the concept to be applied across different domains. Building on this definition, we conduct an extensive set of experiments to examine whether the simple likelihood test, previously criticized for its limitations in image anomaly detection, remains effective in the context of tabular anomaly detection. Consequently, we empirically demonstrate that almost all datasets in ADBench (Han et al., 2022), a tabular AD benchmark dataset, do not exhibit counterintuitive phenomena in the tabular domain, and even NF-SLT outperform comparison models in simple likelihood tests. Furthermore, we demonstrate its success in the tabular domain theoretically and empirically from the perspective of dimensionality and feature correlation.

To explain why this counterintuitive phenomenon does not occur in the tabular domain, we use the following two facts:

**Fact 1.1** (Lower Dimension). *Images typically have three dimensions: height, width, and channel, while tabular data generally have lower dimensionality, consisting of a single feature vector without spatial structure.*

**Fact 1.2** (Correlation of Features). *Images exhibit strong local pixel correlations, which allows models like CNN to effectively capture spatial relationships between neighboring pixels. In contrast, tabular data does not assume any specific structural relationship between features.*

Taking ADBench, as an example, most of the datasets have a dimension lower than 100. However, CIFAR-10, one of the image datasets with small dimensions, has a dimension of 3072. This shows that the curse of dimensionality may be more severe in the image domain than in the tabular domain, and we analyze how this affects the likelihood test using normalizing flow. Additionally, Kirichenko et al. (2020) argued that in image OOD detection, normalizing flows fail to capture semantic information effectively because images exhibit local pixel correlations. Based on Fact 1.2, we extend this discussion to the tabular domain and claim that normalizing flows are less affected by feature correlation in this setting. To justify this claim, we quantify overall feature correlation by measuring the reduction of intrinsic dimension (ID) relative to the ambient dimension. We then explain why this reduction reflects the effect of correlation, and compare the degree of ID reduction observed in tabular and image data. Although there are datasets in the tabular domain that have higher dimensions than images or strong correlation (e.g., genomics, see Appendix C.4), these have very different characteristics from typical tabular datasets, so it is reasonable to assume that the trends of the two domains follow the examples described above.

In conclusion, the contribution of our study can be described as threefold.

- We provide a **domain-agnostic definition of the counterintuitive phenomenon** in simple likelihood tests and empirically show that simple likelihood testing with normalizing flows in the tabular domain rarely leads to this phenomenon, outperforming comparison models.

- We verify our results using all **47 tabular datasets and 10 CV/NLP embedding datasets** from ADBench without selection bias (Shwartz-Ziv & Armon, 2022) and compare against 12 anomaly detection baselines.

- We demonstrate **a theoretical and empirical analysis** of why the counterintuitive phenomenon does not occur in the tabular domain, unlike in images, by linking it to the difference in dimension and feature correlation.

## 2 RELATED WORK

### 2.1 NORMALIZING FLOW

Normalizing flow is one of the generative models that converts input data $\mathbf{x} \in \mathbb{R}^d$, which follows an unknown distribution called $p_\mathbf{x}$, into $\mathbf{z} \in \mathbb{R}^d$; in addition, it follows a simple distribution $p_\mathbf{z}$ that is typically chosen as standard Gaussian $\mathcal{N}(0, I_d)$ (Dinh et al., 2017), using an invertible function $f : \mathbb{R}^d \to \mathbb{R}^d$ that consists of complex functions such as neural networks (Dinh et al., 2015), such that $p_\mathbf{x}$ can be written as a formula in terms of $p_\mathbf{z}$. At this point, $p_\mathbf{x}$ is expressed as the determinant of Jacobian of $\mathbf{x}$ and $\mathbf{z}$ and $p_\mathbf{z}$ by the change-of-variable rule, and is expressed as Equation 1.

$$\log p_\mathbf{x}(\mathbf{x}) = \log p_\mathbf{z}(\mathbf{z}) + \log |J|, J = \det \frac{\partial \mathbf{z}}{\partial \mathbf{x}} \tag{1}$$

In general, it learns in the direction of maximizing the likelihood $\log p_\mathbf{x}(\mathbf{x})$ of the learning input data, and approximates the distribution of the input data (Caterini & Loaiza-Ganem, 2022). Normalizing flows can be categorized by whether the determinant of the Jacobian (i.e., the volume term) is fixed (Dinh et al., 2015) or varies with the input (Rezende & Mohamed, 2015; Dinh et al., 2017; Kingma & Dhariwal, 2018; Behrmann et al., 2019; Chen et al., 2019; Durkan et al., 2019). When sampling new data, sampling is performed by extracting it from the pre-defined $p_\mathbf{z}$ and inputting it as the input of $f^{-1}$. The normalizing flow has the advantage of being able to obtain the estimated likelihood of the input data, unlike models such as variational autoencoder and generative adversarial network. Additionally, the normalizing flow has the advantage of not requiring the additional likelihood approximate inference techniques (Nalisnick et al., 2019a). However, normalizing flow has two constraints: (1) the computational amount of Jacobian must not become too large, and (2) the inverse of $f$ must exist. Therefore, the following methodologies were utilized to ensure the ease of Jacobian calculation and the existence of the inverse $f^{-1}$: methods such as a coupling layer (Dinh et al., 2015; 2017; Kingma & Dhariwal, 2018), special-form transformations (Rezende & Mohamed, 2015), and power-series approximations with Lipschitz constraint (Behrmann et al., 2019; Chen et al., 2019) are commonly used.

### 2.2 COUNTERINTUITIVE PHENOMENON OF LIKELIHOOD

Nalisnick et al. (2019a) reported that a counterintuitive phenomenon regarding likelihood assignment occurs in models that can obtain estimated likelihood, such as normalizing flow, in the image domain. This study lays the foundation for identifying the cause of this phenomenon or suggesting solutions. Kirichenko et al. (2020); Schirrmeister et al. (2020) improved anomaly detection performance by modifying flow architectures. In particular, the latter introduced an approach that reflects the hierarchical data structure, thereby improving detection performance. Serrà et al. (2020) quantified complexity through a general compression algorithm such as PNG, based on experimental results, demonstrating that simple images exhibit higher likelihood, and presented an anomaly score combining the likelihood and complexity terms. Kamkari et al. (2024) used Local Intrinsic Dimension (LID) to measure an image's simplicity and proposed a dual thresholding method for LID and likelihood to improve anomaly detection performance. Morningstar et al. (2021); Osada et al. (2024); Ahmadian et al. (2021) mitigated the drawback of using only a single likelihood score by estimating the density of a vector that combines the likelihood with several auxiliary statistics (e.g., complexity, the log-determinant of the Jacobian). Nalisnick et al. (2019b) demonstrated the perspective that detection may fail because in-distribution data are located in the typicality set (Cover, 1999) and OOD data is in the high density set. Zhang et al. (2021) presented the view that the counterintuitive phenomenon occurs due to misestimation of the model. Le Lan & Dinh (2021) demonstrated that even with a perfect model, simple likelihood-based methods can fail due to variants in the representation. Ren et al. (2019) improved detection performance by using the likelihood ratio between the background and semantic models and Caterini & Loaiza-Ganem (2022) explained the cause of this phenomenon from an entropic perspective and why the likelihood ratio model works well.

## 3 Definition of Counterintuitive Phenomenon

Earlier research (Kirichenko et al., 2020) noted instances where in-distribution and OOD data had overlapping likelihoods in tabular datasets, but these findings were limited to only a few datasets and lacked comprehensive comparisons with other anomaly detection models. To address limitations in prior work's explanations of the counterintuitive phenomenon, we propose a generalized definition of the counterintuitive phenomenon that applies to diverse domains. To formalize this phenomenon, we begin by establishing two core assumptions:

**Assumption 3.1** (Relatively Low Performance). *If a counterintuitive phenomenon occurs, most comparison models should outperform the generative model on an anomaly detection task.*

**Assumption 3.2** (High Performance Gap). *Even if the above condition is satisfied, the performance gap between the generative model and comparison models must be significant to qualify as a counterintuitive phenomenon. If the gap is small, it cannot be considered counterintuitive.*

We now formalize this phenomenon using these assumptions.

**Definition 3.3** (Occurrence of Counterintuitive Phenomenon). *Let* $\mathrm{AUROC}_0$ *denote the AUROC of the likelihood-only test using the generative model* $P_{\theta_0}$ *on a normal/abnormal dataset pair* $(P, Q)$, *and let* $\mathrm{AUROC}_i$ *denote that of the* $i$*-th comparison model for* $i = 1, \ldots, k$. *We say that a counterintuitive phenomenon occurs if both conditions hold:*

$$\frac{1}{k} \sum_{i=1}^{k} \mathbb{1}\{\mathrm{AUROC}_i > \mathrm{AUROC}_0\} > \beta, \tag{2}$$

$$\min_{i:\mathrm{AUROC}_i > \mathrm{AUROC}_0} (\mathrm{AUROC}_i - \mathrm{AUROC}_0) > \gamma. \tag{3}$$

Definition 3.3 states that a counterintuitive phenomenon occurs when the proportion of the comparison models whose AUROC exceeds that of the generative model $P_{\theta_0}$ exceeds $\beta$, and the minimum AUROC difference between $P_{\theta_0}$ and the models that outperform $P_{\theta_0}$ is greater than $\gamma$. Consequently, Definition 3.3 enables performance comparisons using relative AUROC, allowing us to determine whether a counterintuitive phenomenon has occurred, rather than merely inferring its presence from a low AUROC. The fully rigorous formulation of Definition 3.3 is provided in Appendix B.

Consider the CIFAR-10 (in-distribution) vs. SVHN (out-of-distribution). According to Morningstar et al. (2021), a simple likelihood test using the Glow (Kingma & Dhariwal, 2018) yielded an AUROC of 6.4%. In contrast, Sun et al. (2022) achieved AUROC scores exceeding 90% with their proposed method and comparison models. Based on Definition 3.3, this case clearly demonstrates a counterintuitive phenomenon, as the generative model performs significantly worse than the comparison models. To explore whether this phenomenon occurs in tabular data, we conducted experiments to test if a counterintuitive phenomenon, as defined in Definition 3.3, appears in tabular anomaly detection datasets.

## 4 Experiment

**Dataset and Preprocessing** The experiment was conducted using the data split protocol in Zong et al. (2018). To explain this protocol, 50% of normal data is used for training, and the remaining 50% of normal and abnormal data are used as test data. We used **all 47 tabular and 10 CV/NLP embedding datasets** presented in ADBench. Using the entire dataset was motivated by Shwartz-Ziv & Armon (2022), who criticized that researchers often introduced selection bias by choosing specific datasets to inflate performance. To address this, we included all proposed benchmark datasets without exclusion. All models except the NeuTraLAD model utilized RobustScaler provided by the Python library Scikit-learn (Pedregosa et al., 2011) to standardize the input data. The reason for excluding NeuTraLAD is that a significant performance decrease was observed when scaling.

**Models** We compared the performance of 6 shallow AD models and 6 deep AD models. We implemented the shallow models using PyOD (Zhao et al., 2019) and Scikit-learn (Pedregosa et al., 2011). The compared shallow models are PCA (Shyu et al., 2003), LOF (Breunig et al., 2000), IF (Liu et al., 2008), OCSVM (Schölkopf et al., 1999), COPOD (Li et al., 2020), and ECOD (Li et al., 2022). The compared deep models are DAGMM (Zong et al., 2018), DeepSVDD (Ruff et al., 2018), GOAD (Bergman & Hoshen, 2020), NeuTraLAD (Qiu et al., 2021), ICL (Shenkar & Wolf, 2022),

Table 1: (Top): Evaluation performance of 47 tabular data. (Bottom): Evaluation performance of 10 CV/NLP embedding data. The Top2 Ratio indicates the proportion of datasets where model ranked within the top2 for AUROC, the Fail Ratio shows the proportion of datasets where a model's AUROC rank was 9th or lower.

| Method | AUROC ↑ | AUPRC ↑ | Avg. Rank ↓ | Top2 Ratio ↑ | Fail Ratio ↓ |
|---|---|---|---|---|---|
| PCA | 0.7715 | 0.5209 | 6.51 | 0.17 | 0.40 |
| LOF | 0.8169 | 0.5606 | 5.53 | 0.21 | 0.23 |
| IF | 0.8014 | 0.5060 | 5.62 | 0.19 | 0.13 |
| OCSVM | 0.6562 | 0.3833 | 9.47 | 0.06 | 0.72 |
| COPOD | 0.7471 | 0.4419 | 7.68 | 0.11 | 0.49 |
| ECOD | 0.7425 | 0.4530 | 7.98 | 0.09 | 0.49 |
| DAGMM | 0.6467 | 0.3468 | 10.51 | 0.00 | 0.85 |
| DeepSVDD | 0.7687 | 0.5388 | 6.96 | 0.06 | 0.34 |
| GOAD | 0.6086 | 0.4114 | 9.72 | 0.04 | 0.60 |
| NeuTraLAD | 0.8081 | 0.5694 | 5.57 | 0.26 | 0.26 |
| ICL | 0.8208 | 0.6170 | 5.17 | 0.32 | 0.23 |
| MCM | 0.7864 | 0.5383 | 6.70 | 0.11 | 0.23 |
| NF-SLT | **0.8575** | **0.6398** | **3.43** | **0.45** | **0.02** |

| Dataset | DeepSVDD | GOAD | NeuTraLAD | ICL | MCM | NF-SLT |
|---|---|---|---|---|---|---|
| CIFAR-10 | 0.9103 | 0.9335 | 0.9405 | 0.9254 | 0.9381 | **0.9527** |
| FashionMNIST | 0.9117 | 0.9060 | 0.9360 | 0.9267 | 0.9380 | **0.9455** |
| MNIST-C | 0.8348 | 0.7741 | 0.8519 | 0.8257 | 0.8836 | **0.8950** |
| MVTecAD | 0.7543 | 0.7960 | 0.8874 | 0.8874 | 0.8408 | **0.9100** |
| SVHN | 0.5466 | 0.5366 | 0.5774 | 0.5626 | 0.5770 | **0.5842** |
| 20news | 0.5547 | 0.5438 | 0.6001 | 0.6087 | 0.5995 | **0.6547** |
| agnews | 0.6630 | 0.5857 | 0.6510 | 0.6697 | 0.7252 | **0.7591** |
| amazon | 0.5833 | 0.5613 | 0.6010 | 0.6022 | 0.6050 | **0.6194** |
| imdb | 0.5090 | **0.5398** | 0.5393 | 0.5098 | 0.5090 | 0.5013 |
| yelp | 0.6490 | 0.6138 | 0.6620 | 0.6690 | 0.6750 | **0.6971** |

MCM (Yin et al., 2024), and NF-SLT with NICE (Dinh et al., 2015). For NF-SLT with NICE, we used 10 coupling layer and trained the model for 200 epochs with weight decay 1e-4. We optimized the negative log-likelihood of the latent variables using AdamW (Loshchilov & Hutter, 2019) with a CosineAnnealingWarmRestarts learning rate scheduler (Loshchilov & Hutter, 2017). The batch size was set to 512, and the latent prior was fixed to $\mathcal{N}(0, I_d)$. Overall hyperparameter settings and implementation details are provided in Appendix F.

**Evaluation** We evaluate these AD models using AUROC and AUPRC. We performed 10 repeated experiments on the tabular datasets and recorded the average AUROC scores and the relative rank of each model as summarized in Table 1. For each dataset, after experimenting with all combinations in the hyperparameter searching space with 10 repeated experiments, the hyperparameter combination with the highest average AUROC for all datasets is selected as the representative hyperparameter combination to demonstrate the performance of the model. The hyperparameter search space for each model and hyperparameter sensitivity experiment is recorded in Appendix F. Additionally, the results of applying other flows to NF-SLT are included in Appendix G.

**Experiment Result** Consider Definition 3.3; if a counterintuitive phenomenon is also frequent in the tabular domain, it should have a high fail ratio even if it works well on a particular dataset resulting in a high top2 ratio. In addition, the failed dataset should have a large minimum performance difference from the other models. However, based on the results in Table 1, we can observe that NF-SLT has a lower fail ratio than the shallow and deep models, and outperforms other metrics. Furthermore, on the 'yeast' dataset where NF-SLT exhibited relatively low performance, the minimum performance difference between MCM and AUROC is 0.02; hence, we cannot assume that it exhibited low performance due to a counterintuitive phenomenon. Furthermore, NF-SLT outperforms deep models on ADBench's CV/NLP embedding datasets, excluding the 'imdb' dataset. Although it shows worse performance than other models on the 'imdb' dataset, the difference in performance with the comparison model is very small, so it cannot be judged that a counterintuitive phenomenon has occurred because it does not satisfy the second condition of Definition 3.3. Furthermore, we report

the detection performance on datasets dominated by categorical features and various anomaly types in Appendix E, and the results of the experiments also demonstrate the superiority of NF-SLT. To verify the consistency, we compared its performance with other test methodologies such that typicality test (Nalisnick et al., 2019b), with results in Appendix H.

## 5 Why Is The Simple Likelihood Test Successful in Tabular Data?

### 5.1 High Dimension Perspective

Based on Fact 1.1, we explain why tabular data can be successful in likelihood testing because of their lower dimensionality. It has been reported that the case where the likelihood of normal and anomaly data is inverted in the image domain usually occurs when the normal data has a more complex texture than the anomaly data, that is, when the complexity of the normal data is higher than that of the anomaly data (Serrà et al., 2020). Additionally, it can be thought that the high complexity of data sampled from a specific distribution means that the entropy of the distribution is high. Hence, to explain why the counterintuitive phenomenon rarely occurs in the tabular domain, we extend the likelihood-gap expression of Caterini & Loaiza-Ganem (2022), which characterizes the expected likelihood difference between normal and abnormal data in terms of entropy, and link it to Fact 1.1.

Let the distribution of normal data be $P$, the distribution of abnormal data be $Q$, and let $P_\theta$ be a model such as normalizing flow that estimates the density of $P$. Then, the gap of the likelihood of each distribution estimated by $P_\theta$ can be expressed as follows:

$$\mathbb{E}_{\mathbf{x} \sim P}[\log P_\theta(\mathbf{x})] - \mathbb{E}_{\mathbf{x} \sim Q}[\log P_\theta(\mathbf{x})] = D_{KL}(Q||P_\theta) - D_{KL}(P||P_\theta) + \mathbb{H}(Q) - \mathbb{H}(P) \quad (4)$$

where $\mathbb{H}(P)$ is entropy of distribution $P$, and $D_{KL}(Q||P_\theta)$ is the KL-divergence of distribution $Q$ and density estimation model like normalizing flow $P_\theta$. In Equation 4, if the difference in entropy between the two distributions $\mathbb{H}(Q) - \mathbb{H}(P)$ is a very small negative number , the expectation gap of the likelihood can become negative. However, in the previous study in Caterini & Loaiza-Ganem (2022), the effect of the dimension in expectation of likelihood gap was not analyzed, so we analyzed how the dimension can affect the expectation of likelihood gap and included it in Theorem 5.4 and the proof of this is reported in Appendix D.

**Theorem 5.4** (Impact of Dimensionality on Likelihood Gap). *Let $P = \prod_{i=1}^{d} p_i(x_i)$ and $Q = \prod_{i=1}^{d} q_i(x_i)$ be independent $d$-dimensional continuous probability density models in $\mathbb{R}^d$ with same conditions as Lemma 5.1. Let $P_\theta$ be a well-trained density estimation model approximates $P$ (i.e., $p_\theta(x) \to p(x)$ pointwisely as $\theta \to \theta_0$). If $\mathbb{H}(P) - \mathbb{H}(Q) > D_{KL}(Q||P)$, the lower bound of gap between the expectation of the likelihood for $P$ and $Q$ decreases linearly with respect to $d$.*

According to Theorem 5.4, even when $P_\theta$ is almost perfect model, if $P$ and $Q$ are $d$-dimensional independent distributions and the difference in entropy between the two distributions is greater than a $D_{KL}(Q||P)$, it can be verified that the lower bound of gap between the expectation of the likelihood for $P$ and $Q$ is negative and decreases linearly with the dimension. This shows that as the dimension increases, the phenomenon of inversion of the likelihood expectation of data sampled from each distribution can become more severe. Additionally, we show in Corollary 5.6 that under additional assumptions on the entropy of the distribution, not only the likelihood gap but also the upper bound of the AUROC, which is a practical and widely used evaluation metric, is inversely proportional to the dimension. The proof of this result is provided in Appendix D.

**Corollary 5.6** (Dimensionality and AUROC Upper Bound). *Building on the assumptions of Corollary 5.5, suppose the $n$-th absolute central moment of the log-likelihood difference, $\log p_\theta(Y) - \log p_\theta(X)$, scales as $\mathcal{O}(d^k)$ for some $n > 1$ and $k < n$. In this case, if the average log-likelihood gap becomes negative, the maximum achievable AUROC for distinguishing samples from $P$ and $Q$ is inversely related to the dimensionality $d$. This indicates that as the dimension increases, the likelihood test becomes fundamentally less effective at separating normal and abnormal samples.*

According to Corollary 5.6, the upper bound on the achievable AUROC decreases. This is consistent with Definition 3.3: a smaller AUROC implies a higher likelihood that the counterintuitive phenomenon occurs. To validate this prediction, we conducted dimensionality-reduction experiments. Specifically, we applied ICA (Hyvärinen & Oja, 2000) to high-dimensional image data and retained a varying number of independent components. Using RealNVP, we then measured the AUROC as

Table 2: AUROC scores for likelihood tests as a function of dimensionality (number of PCs) using RealNVP with MLP (image preprocessed by ICA). The region to the left of the bold vertical line indicates cases where $\mathbb{H}(P) > \mathbb{H}(Q)$, and the region to the right is the opposite.

| In-dist ($P$) / Out-dist ($Q$) | 1024 | 512 | 256 | 30 | In-dist ($P$) / Out-dist ($Q$) | 1024 | 512 | 256 | 30 |
|---|---|---|---|---|---|---|---|---|---|
| CIFAR-10 / SVHN | 0.2311 | 0.2924 | 0.2984 | 0.3143 | SVHN / CIFAR-10 | 0.9917 | 0.9843 | 0.9486 | 0.8520 |
| CIFAR-100 / SVHN | 0.0843 | 0.1160 | 0.2036 | 0.3490 | SVHN / CIFAR-100 | 0.9933 | 0.9536 | 0.9137 | 0.8622 |
| CelebA / SVHN | 0.1207 | 0.1782 | 0.2745 | 0.4711 | SVHN / CelebA | 0.9976 | 0.9811 | 0.9722 | 0.9481 |

a function of the retained dimension in Table 2. This setup isolates the effect of dimensionality on likelihood ranking and provides empirical support for our theoretical claims.

The results in Table 2 show that, when $\mathbb{H}(P) > \mathbb{H}(Q)$ holds, the AUROC increases as the dimensionality decreases. Notably, the improvement remains substantial even when the dimensionality is reduced to almost 1% of the original dimension. In contrast, the cases to the right of the bold vertical line show decreasing AUROC as the dimension decreases. This matches the trivial behavior obtained by reversing the entropy condition, i.e., $\mathbb{H}(P) < \mathbb{H}(Q)$, in Theorem 5.4 and Corollary 5.6. Therefore, even if the $\mathbb{H}(P) > \mathbb{H}(Q)$ condition is satisfied, tabular data can be considered more advantageous in the simple likelihood test because they are less exposed to the problems that arise in high dimensions, as indicated by Fact 1.1.

We also adjusted the dimension of the image using the bilinear interpolation resize method provided by torchvision (Marcel & Rodriguez, 2010) for the raw image, and performed a likelihood test after obtaining the likelihood of the image through Glow which is consist of a CNN, the results are included in Table 3. Since this experiment uses raw images, independence between pixels is not guaranteed, so the theorem presented in Appendix D cannot be applied. However, this experiment was conducted to check the performance trend according to dimension in a situation where there is no independence assumption.

Table 3: AUROC scores for likelihood tests as a function of image size using Glow (image resized by bilinear interpolation). The region to the left of the bold vertical line indicates cases where $\mathbb{H}(P) > \mathbb{H}(Q)$, and the region to the right is the opposite.

| In-dist ($P$) / Out-dist ($Q$) | 32x32 | 16x16 | 8x8 | In-dist ($P$) / Out-dist ($Q$) | 32x32 | 16x16 | 8x8 |
|---|---|---|---|---|---|---|---|
| CIFAR-10 / SVHN | 0.0716 | 0.3586 | 0.4512 | SVHN / CIFAR-10 | 0.9902 | 0.9777 | 0.9195 |
| CIFAR-100 / SVHN | 0.0846 | 0.4448 | 0.3918 | SVHN / CIFAR-100 | 0.9900 | 0.9798 | 0.9481 |
| CelebA / SVHN | 0.1541 | 0.3056 | 0.7037 | SVHN / CelebA | 0.9850 | 0.9968 | 0.9982 |
| CIFAR-100 / CIFAR-10 | 0.4857 | 0.4933 | 0.5016 | CIFAR-10 / CIFAR-100 | 0.5259 | 0.5446 | 0.5567 |
| CelebA / CIFAR-10 | 0.7481 | 0.7137 | 0.7557 | CIFAR-10 / CelebA | 0.5087 | 0.6181 | 0.6751 |

Surprisingly, we can see that there are cases where the AUROC exceeds 0.5 when reducing the size of the two images in Table 3 (see CelebA vs SVHN case). In addition, when SVHN is set to in-distribution and CelebA is set to out-of-distribution, the performance tends to increase as the dimension decreases, which is a result that conflicts with the theorems in Appendix D. We argue that this is because resizing an image via bilinear interpolation strengthens the correlation between image pixels, which significantly reduces the entropy of the image distribution where each texture is complex (i.e., high entropy). Therefore, although it is difficult to confirm the effect of dimension on AUROC through the methodology, it can be seen that not only can the performance be improved by the simple image resize methodology for cases where likelihood inversion occurs, but also it is possible to increase the AUROC to more than 0.5.

We reported experimental setting of Table 2 and the impact of dimensionality when simply applying PCA, and the effect of dimension in real tabular data in Appendix C.4. In addition, we conducted an experiment to distinguish between two Gaussian distributions with different means using a likelihood test using a NICE and RealNVP consisting of ReLU-like functions, and found that as the dimension increases, the AUROC approaches 0.5. Since this phenomenon is also a case where AUROC seriously decreases simply as the dimension increases, we present experiments and our theoretical analysis about flow's latent vector in high dimensional space in Appendix C.

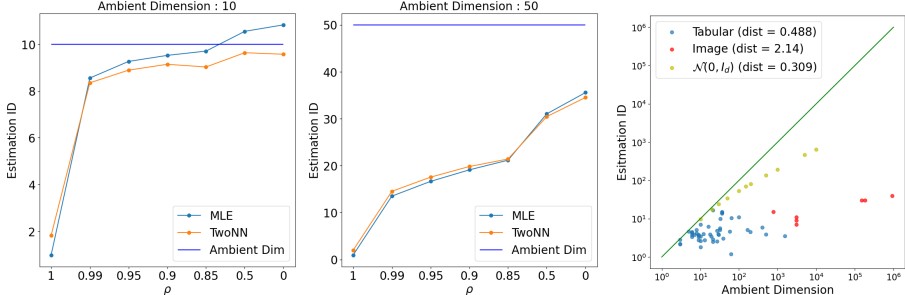

Figure 1: (Left, Center) : Estimation of ID according to changes in $\rho$. The x-axis represents the value of $\rho$ and the y-axis represents dimension. The horizontal line represents the ambient dimension. (Right) : Log-scale scatter plot for ID estimate and ambient dimension using the TwoNN method of real dataset and synthetic dataset sampled by $\mathcal{N}(0, I_d)$. The estimated ID for the image dataset are from Table 1 in Pope et al. (2021). The distance recorded in the legend represents the average value of the distance between each point and the green line indicating when the ID estimate and ambient dimension are the same. Each blue/red point corresponds to one ADBench/image dataset.

## 5.2 FEATURE CORRELATION PERSPECTIVE

In prior work, Kirichenko et al. (2020) showed that likelihood inversion can arise in normalizing flows on image data due to strong local pixel correlations. Meanwhile, Schirrmeister et al. (2020) reported that, in the image domain, OOD detection performance can improve when flow architectures use multi-layer perceptron (MLP) rather than convolutional neural network (CNN)in certain settings. As noted by Battaglia et al. (2018), CNNs exhibit an inductive bias known as locality, making them particularly effective for image data where local pixel correlations are strong. In contrast, MLPs have a weak inductive bias and are thus more suitable for tabular data, where no strong correlation between features is assumed.

Leveraging Fact 1.2 and noting that architectural inductive bias is shaped by domain-specific characteristics, we argue that the counterintuitive phenomenon is infrequent in tabular data. This is because tabular features are inherently heterogeneous, unlike the homogeneous features of image data. Images have homogeneity because all features are composed of the same type of "pixel" and have values in the same interval from 0 to 255 and local pixels have strong correlations. However, tabular features are composed of many types such as continuous, discrete, and categorical, and the interval (or category) of these values can be set arbitrarily, and no assumptions are made between features so it has heterogeneity.

Hence, we quantify heterogeneity and homogeneity to interpret the differences between the two domains from the perspective of correlation. To quantify feature heterogeneity and homogeneity, we measure overall feature correlation, which captures the strength of relationships between features. This is because correlation indicates the strength of the relationship between features, and if this strength is high, it can be interpreted as a strong tendency to follow a specific pattern (e.g., non-linear relationship), making it possible to determine whether the features are heterogeneous or homogeneous. However, since common feature correlation methodologies (e.g., Pearson correlation coefficient) are based on the relationship between two features, they do not capture global correlation structure. Hence, we quantify correlation indirectly by showing that for data $\mathbf{x} \in \mathbb{R}^d$ as the strength of the correlation between features increases, the intrinsic dimension (ID) $d'$ becomes smaller relative to the ambient dimension $d$. Please refer to Camastra & Staiano (2016) for a definition of ID. Since it is impossible to know the exact ID, we estimate the ID using MLE (Levina & Bickel, 2004) and TwoNN (Facco et al., 2017), which are popular methods for estimating the ID based on fractal theory.

First, we demonstrate our claimed relationship between correlation and ID using random variables that follow a Gaussian distribution as a toy example. Let $X \sim \mathcal{N}(0, \Sigma)$, $\Sigma$ can be organized into an

Table 4: (Top) : ID estimates for real dataset. (Bottom) : The ratio of datasets with a rank more than or equal to 3 according to the $d$ Ratio. $d$ Ratio is the ratio of intrinsic dimension estimated by TwoNN to ambient dimension. Additionally, the results about image dataset were recorded with reference to the results recorded in Pope et al. (2021).

| Dataset | MNIST | CIFAR-10 | CIFAR-100 | SVHN | magicgamma | satellite | landsat | waveform |
|---------|-------|----------|-----------|------|------------|-----------|---------|----------|
| MLE | 13 | 26 | 23 | 19 | 7 | 12 | 11 | 16 |
| TwoNN | 15 | 11 | 9 | 7 | 7 | 15 | 14 | 17 |
| $d$ Ratio | 0.019 | 0.003 | 0.002 | 0.002 | 0.700 | 0.417 | 0.389 | 0.810 |

| $d$ Ratio Threshold | 0.1 | 0.2 | 0.3 | 0.4 | 0.5 | 0.6 | 0.7 | 0.8 |
|---------------------|-----|-----|-----|-----|-----|-----|-----|-----|
| Rank $\geq 3$ Dataset Ratio | 0.160 | 0.440 | 0.640 | 0.760 | 0.840 | 0.840 | 0.92 | 1.000 |

$d$-dimensional autoregressive covariance structure and represented by a formula as follows

$$
\Sigma = \begin{bmatrix}
1 & \rho & \rho^2 & \cdots & \rho^{d-1} \\
\rho & 1 & \rho & \cdots & \rho^{d-2} \\
& & \vdots & & \\
\rho^{d-1} & \rho^{d-2} & \rho^{d-3} & \cdots & 1
\end{bmatrix}, \rho \in [0,1].
\tag{5}
$$

We set the covariance $\Sigma$ as in Equation 5 so that adjusting $\rho$ controls the strength of correlations among variables. If $\rho$ is closer to 1, the correlation between each variable will be stronger. Then, we use TwoNN and MLE ($k = 10$) ID estimators to describe the change in ID according to $\rho$ in left and center subplot of Figure 1 for $X$ with an ambient dimension of 10 and 50. Through left and center subplot of Figure 1, we can interpret that both ID estimators estimate smaller ID values when $\rho$ increases. Therefore, it can be seen that stronger correlation between variables leads the ID to take values considerably smaller than the ambient dimension. The plot on the center of Figure 1 shows that even when $\rho = 0$, it underestimates the ID, but this is because the estimator tends to underestimate the ID when it has a large truth ID, so it is reasonable to look at the estimate as a lower bound when dataset has a large truth ID (Ansuini et al., 2019; Sharma & Kaplan, 2022).

Based on the results obtained from the preceding experiments, we define the ratio of the intrinsic dimension to the ambient dimension as the $d$ Ratio to measure the overall feature correlation. A higher degree of feature correlation results in a lower intrinsic dimension estimate, and thus a smaller $d$ Ratio, whereas weaker correlation yields a larger $d$ Ratio.

Subsequently, to validate the findings from synthetic data experiments on real datasets, we estimate the ID of real-world image and tabular datasets and compare these estimates to their corresponding ambient dimensions. To compare this, we report ID estimates in Table 4 using MNIST, CIFAR-10, CIFAR-100, SVHN image datasets, which are mainly used in image benchmarks and ADBench's dataset, using MLE ($k = 20$) and TwoNN. According to Table 4, all four image datasets have a $d$ Ratio of about 1%, whereas the tabular datasets exhibit substantially higher $d$ Ratio values compared to the images.

Additionally, we recorded a log-scale plot with each dimension as the axis in Figure 1 to check the tendency of the ID estimation values and ambient dimensions of the tabular and image datasets. As a result of comparing the average distance between the green line in Figure 1, it is perceived that the image has a larger average distance than the tabular. In addition to the numerical results, it can be seen visually that the blue points (tabular dataset) are much closer from the green line than the red points (image dataset), and we can see that the yellow points follow $\mathcal{N}(0, I_d)$ which are theoretically perfectly uncorrelated data points are formed close to the green line. Through these results, it can be seen that tabular data has an ID closer to the ambient dimension than image data, so it can be concluded that they exhibit a lower correlation between features than image data.

Furthermore, for 25 datasets where NF-SLT does not achieve top performance (rank $\geq 3$), we show the fraction of datasets with a $d$ Ratio below a certain threshold in Table 4. These experimental results show that NF-SLT fails to achieve high performance on most datasets with low $d$ Ratio, even within the tabular domain. Therefore, we conclude that one factor behind the high detection performance of tabular data is the heterogeneous nature of its features. We further argue that this effect may act

in combination with the improvement in anomaly detection performance obtained using MLPs, as reported in (Schirrmeister et al., 2020).

Moreover, to account for the absence of counterintuitive phenomena on CV/NLP embeddings in Section 4, we estimate the intrinsic dimension of the ADBench CIFAR-10 and SVHN embedding representations using TwoNN. The estimated intrinsic dimensions are 23 and 18, respectively, whereas the ambient embedding dimension is 1000 (smaller than the 3072-dimensional raw pixel space). Despite this reduced ambient dimension, the embeddings exhibit higher intrinsic dimensionality than the original images, implying a larger $d$ Ratio. This suggests that the embedding features are less strongly correlated and span a higher-dimensional manifold than raw pixels. Consequently, the high-dimensional issues that degrade likelihood ranking are mitigated, making NF-SLT effective on these embeddings. This explanation is consistent with Kirichenko et al. (2020), which reported that the counterintuitive phenomenon is alleviated when using semantic embedding representations instead of raw pixels. Hence, unlike images, the tabular domain generally has low feature correlation, which contributes to its heterogeneous nature and makes it difficult to satisfy the assumption in Definition 3.3 that the generative model's relative performance should be low.

## 6 CONCLUSION

This paper examined whether the counterintuitive phenomenon in image anomaly detection also appears in tabular data. We first provided a domain-agnostic definition of this phenomenon, allowing it to be analyzed consistently across different data types. Using theoretical and empirical analyses with extensive experiments, we showed that this phenomenon rarely occurs in tabular data with simple likelihood tests using normalizing flows. Our results show that flow-based likelihood tests effectively detect tabular anomalies, outperforming traditional models without facing image domain challenges. For future work, we hope to see the development of flow architectures that can better capture semantic information in tabular data, as well as theoretical and empirical studies that extend these methods to high-dimensional tabular datasets with correlation structures comparable to those in image data.

## REPRODUCIBILITY STATEMENT

To ensure the reproducibility of our work, we provide details on our experimental setup. Appendix C.4 and F provide information necessary for reproducibility, including the sources of the comparison models and the hyperparameters used in the experiments. We also provide code of NF-SLT for reproducing our experiments in the supplementary materials.

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

## A  DEFINITION OF OOD DETECTION AND ANOMALY DETECTION

The anomaly detection task aims to construct a classifier capable of detecting OOD abnormal instances (Golan & El-Yaniv, 2018). This is the same as the goal of OOD detection, which treats OOD detection and anomaly detection under the same goal. OOD detection aims to identify data points that follow a different distribution than the in-distribution, while anomaly detection does not assume a specific distribution for anomalies. However, since this assumption does not alter the fundamental goal of anomaly detection, we regard the two tasks as equivalent for the purposes of this work.

## B  RIGOROUS FORMULATION OF DEFINITION 3.3

**Definition B.1** (Occurrence of Counterintuitive Phenomenon). *Let $\mathbf{x} \sim P$, where $P_{\theta_0}$ provides an approximately exact likelihood estimate of the input $\mathbf{x}$, and let $P_{\theta_k}$ represent $k$-th comparison models that do not provide likelihood estimates. Assume all models are well-trained.*

*Let $\varphi_{P_{\theta_0}}(\mathbf{x})$ represent the likelihood estimate from the generative model $P_{\theta_0}$, and let $\varphi_{P_{\theta_k}}(\mathbf{x})$ represent the test statistic (e.g., anomaly score) from the $k$-th comparison model. Let*

$$R = \left\{ i \in [k] \mid \Pr(\varphi_{P_{\theta_0}}(\mathbf{x}) > \varphi_{P_{\theta_0}}(\mathbf{y})) < \Pr(\varphi_{P_{\theta_i}}(\mathbf{x}) > \varphi_{P_{\theta_i}}(\mathbf{y})) \right\}. \tag{6}$$

*We define that a counterintuitive phenomenon occurs if the following two conditions are satisfied for $\mathbf{y} \sim Q$:*

1. *The majority of comparison models outperform the generative model:*

$$\frac{1}{k} \sum_{i=1}^{k} \mathbb{1} \left\{ \Pr(\varphi_{P_{\theta_0}}(\mathbf{x}) > \varphi_{P_{\theta_0}}(\mathbf{y})) < \Pr(\varphi_{P_{\theta_i}}(\mathbf{x}) > \varphi_{P_{\theta_i}}(\mathbf{y})) \right\} > \beta. \tag{7}$$

2. *The minimum performance gap between the generative model and the outperforming comparison models is significant:*

$$\min_{i \in R} \left( \Pr(\varphi_{P_{\theta_i}}(\mathbf{x}) > \varphi_{P_{\theta_i}}(\mathbf{y})) - \Pr(\varphi_{P_{\theta_0}}(\mathbf{x}) > \varphi_{P_{\theta_0}}(\mathbf{y})) \right) > \gamma. \tag{8}$$

## C  ADDITIONAL DESCRIPTION FROM SECTION 5.1

In the following, we examine why anomaly detection using normalizing flows can fail as dimensionality increases in theoretical perspectives. To do so, we will review some concentration behavior of the Euclidean norm in high dimensions because it plays a key role in understanding the performance of normalizing flows in distinguishing between normal and anomalous data. This is because the normalizing flow model $f$ is trained on normal data $X$ such that $f(X)$ follows a standard Gaussian distribution, i.e., $f(X) \sim N(0, I)$ and the training objective is to maximize the log-likelihood, $\log P_{N(0,I)}(f(X) \mid \Theta) \propto -\|f(X)\|_2^2$, where $\Theta$ represents the parameters of the normalizing flow and $\| \cdot \|_2$ denotes the Euclidean norm. Our analysis in Section C.1 & C.2 cannot directly analyze the volume term in models that do not preserve volume, such as RealNVP and Glow, but since Osada et al. (2024) shows that the volume term and the latent likelihood are positively correlated, we indirectly study the relationship with the volume term as a behavior of the latent vector. Additionally, in Section C.3 we show that dimension size has a negative impact on the likelihood test, regardless of the volume term.

### C.1  EUCLIDEAN NORM GETS MORE CONCENTRATED AS DIMENSION INCREASES

If $Z \sim \mathcal{N}(0, I_d)$, then $\|Z\|_2^2$ concentrates near $d$. This means that the test statistic $\|f(X)\|_2^2$ is close to $d$ if X is normal data and the normalizing flow model $f$ is well trained so that $f(X)$ follows $\mathcal{N}(0, I_d)$. Thus, if the normalizing flow is well-trained, the transformed normal data $f(X)$ concentrate on a sphere of radius $\sqrt{d}$ (where $d$ is the dimensionality of $X$) due to the tail bound properties of sum of independent normal distributions. First, we prove Proposition C.1, which states that as the dimension increases, the Euclidean norm tends to concentrate near $\sqrt{d}$.

**Proposition C.1.** *If $Z \sim \mathcal{N}(0, I_d)$, then for all $0 < t < d$ :*

$$\Pr\left(\left|||Z||_2^2 - d\right| \geq t\right) \leq 2e^{-\frac{t^2}{8d}} \tag{9}$$

*Proof.* Take a random variable $Z \sim \mathcal{N}(0, I_d)$ in $\mathbb{R}^d$. Then for each $Z_i \sim N(0,1)$, $\mathbb{E}\left[e^{\lambda(Z_i^2 - 1)}\right] = \frac{e^{-\lambda}}{\sqrt{1-2\lambda}} \leq e^{4\lambda^2/2}$ for all $|\lambda| < 1/4$. Thus, $||Z||_2^2$ is sub-exponential with parameters $(2\sqrt{d}, 4)$ and by the properties of sub-exponential random variables, we obtain the concentration bound : $\Pr\left(\left|||Z||_2^2 - d\right| \geq t\right) \leq 2e^{-\frac{t^2}{8d}}$ for $0 < t < d$. $\square$

Since we can show by Proposition C.1 that $|Z|_2^2$ is a sub-exponential random variable with parameter containing $d$, we can verify that the probability that the Euclidean norm is observed far from $\sqrt{d}$ has a sub-exponential tail bound. Then, we prove that the variance of the Euclidean norm tends to concentrate more shrinks relative to the increase in dimensionality.

**Theorem C.2** (Klartag (2007); Fleury et al. (2007); Guédon (2014)). *$X$ is a log-concave isotropic random variable in $\mathbb{R}^d$. If $\exists \epsilon_d(\epsilon_d \to 0)$ such that $\Pr\left(\left|\frac{||X||_2}{\sqrt{d}} - 1\right| \geq \epsilon_d\right) \leq \epsilon_d$, then $\lim_{d \to \infty} \frac{\text{Var}||X||_2}{d} = 0$*

**Proposition C.3.** *If $Z \sim \mathcal{N}(0, I_d)$,*

$$\lim_{d \to \infty} \frac{\text{Var}||Z||_2}{d} = 0 \tag{10}$$

*Proof.* From the Proposition C.1, take $t = dt$. Then $\Pr\left(\left|||Z||_2^2 - d\right| \geq dt\right) \leq 2e^{-\frac{dt^2}{8}}$ for $0 < t < 1$. Since there exists $\epsilon_d$ such that $\max\{t, 2e^{-\frac{dt^2}{8}}\} < \epsilon_d$ and $\epsilon_d \to 0$ and $\Pr\left(\left|\frac{||Z||_2}{\sqrt{d}} - 1\right| \geq t\right) = \Pr\left(\left|\frac{||Z||_2^2}{d} - 1\right| \geq t\right)$, there exists $\epsilon_d(\epsilon_d \to 0)$ such that $\Pr\left(\left|\frac{||Z||_2}{\sqrt{d}} - 1\right| \geq \epsilon_d\right) \leq \epsilon_d$. By Theorem C.2, $\lim_{d \to \infty} \frac{\text{Var}||Z||_2}{d} = 0$ $\square$

Based on the Proposition C.3, as dimensionality increases, the Euclidean norm of a normal random variable tends to concentrate more quickly relative to the increase in dimensionality. The propositions show that the latent variable of normalizing flow following $\mathcal{N}(0, I_d)$ has a variance that grows more slowly than the increase in dimension, and that its norm deviates from $\sqrt{d}$. This infers that as dimension increases, the latent variable of a normalizing flow corresponding to the normal data is concentrated in a sphere with radius $\sqrt{d}$.

Therefore, the model becomes more vulnerable to slight misestimation or error as dimension increases. Also, if the norm of the latent variable corresponding to anomalous data, after being passed through the flow model trained on high-dimensional data, is smaller than $\sqrt{d}$, the detection performance may degrade when using a simple likelihood-based test. As a result, outcomes may emerge that align with assumptions about counterintuitive phenomena.

## C.2    EUCLIDEAN NORM IS ALMOST IDENTICAL IN HIGH-DIMENSIONAL SPACE

In high-dimensional spaces, the Euclidean norm becomes a less effective statistical measure, as data points from distinct distributions exhibit nearly identical norms. We summarize and demonstrate that all the isotropic and log-concave random variables become indistinguishable from normal distribution in terms of Euclidean norm as the dimensionality increases. Before starting the analysis, we assume that the distribution of latent vectors obtained when passing the anomaly data through the flow is not $N(0, I_d)$ but is isotropic and log-concave, as the histograms shown in Appendix C.3 are almost similar.

**Theorem C.4** (Guédon & Milman (2011)). *For a log-concave and isotropic random variable $X$ in $\mathbb{R}^d$, there exists a constant $C$ such that for any $t > 0$,*

$$\Pr\left(||X||_2 - \sqrt{d} \geq t\sqrt{d}\right) \leq Ce^{-c\sqrt{d}\min\{t^3, t\}}$$

**Conjecture C.5** (Thin-Shell Conjecture). *For a log-concave and isotropic random variable $X$ in $\mathbb{R}^d$, there exists a constant $C$ such that for any $t > 0$,*

$$\Pr\left(\left|\|X\|_2 - \sqrt{d}\right| \geq t\sqrt{d}\right) \leq 2e^{-Ct\sqrt{d}}$$

Although the Thin-Shell conjecture has not yet been proven, there have been several breakthroughs by the works including Eldan (2013) and Chen (2021). As the Thin-Shell Conjecture and the results of Guédon & Milman (2011) show, all the log-concave and isotropic random variables have their Euclidean norm near $\sqrt{d}$. This makes it hard for normalizing flow model to distinguish various other log-concave distributions from normal distribution as dimension increases.

**Theorem C.6** (Anttila et al. (2003)). *$X$ is log-concave isotropic random variable in $\mathbb{R}^d$. If there exists $\epsilon_d \to 0$ as $d \to \infty$ such that $\Pr\left(\left|\frac{\|X\|_2}{\sqrt{d}} - 1\right| \geq \epsilon_d\right) \leq \epsilon_d$, then there exists $\theta \in S^{d-1}$*

$$\sup_{t>0}\left|\Pr\left(\sum_{i=1}^d \theta_i X_i \leq t\right) - \frac{1}{\sqrt{2\pi}}\int_{-\infty}^t e^{-v^2/2}dv\right| \leq \eta_d$$

*, where $\eta_d \to 0$*

This theorem by Anttila et al. (2003) demonstrates that if the Euclidean norm of a random variable in $\mathbb{R}^d$ concentrates near $\sqrt{d}$, there exists a linear functional of $X$ that closely approximates a normal distribution. Klartag (2007) extended this result, showing that almost every linear functional of $X$ becomes approximately normally distributed as $d \to \infty$. These results imply that in high-dimensional spaces, the concentration of the Euclidean norm is nearly identical across distributions, which reduces the effectiveness of hypothesis tests based on the Euclidean norm in distinguishing between distributions. In fact, as shown in the experimental results in Appendix C.3, the likelihood histograms reveal that although the normal and anomaly data are clearly derived from different distributions, the distributions of their likelihoods overlap as the dimensionality increases.

**Theorem C.7.** *Let $X, Y$ be random vectors with isotropic and log-concave density in $\mathbb{R}^d$ and $\sigma$ be the uniform probability measure on sphere $\mathbb{S}^{d-1}$. Then there exist $\epsilon_d \to 0, \delta_d \to 0$ and subset $\Theta$ in sphere $\mathbb{S}^{d-1}$ such that $\sigma(\Theta) > 1 - \delta_d$ and for any $\theta, \phi \in \Theta$,*

$$d_{TV}(\langle X, \theta\rangle, \langle Y, \phi\rangle) < \epsilon_d$$

It is straightforward to derive Theorem C.7 from the theorem in Klartag (2007). This theorem demonstrates that two random vectors from isotropic and log-concave distributions are almost indistinguishable by comparing the linear measurements. Specifically, since the directions $\theta$ and $\phi$ in the theorem C.7 are drawn uniformly at random and belong to a subset $\Theta$ with measure approaching 1, the likelihood ordering based on projections strongly correlates with that based on norms. So,

$$d_{TV}(\|X\|_2, \|Y\|_2) = d_{TV}(\langle X, \frac{X}{\|X\|_2}\rangle, \langle Y, \frac{Y}{\|Y\|_2}\rangle) < \epsilon_d \tag{11}$$

which holds with high probability under the assumptions of Theorem C.7. Since normalizing flow model uses Euclidean norm of latent vector, this explains the degraded performance of the normalizing flow model based anomaly detection for high-dimensional data.

## C.3 Experiments for Sections C.1 & C.2 Using Synthetic Data

In this section, we discuss experimental implementations on synthetic data that demonstrate a phenomenon consistent with the results in Sections C.1 & C.2. Although Sections C.1 & C.2 hold trivially for zero-centered distributions, our analysis extends beyond this case by confirming that the Euclidean norms remain identical even when the distributions are not zero-centered.

The normal and anomaly distributions are set to Gaussian distributions but have different parameters $\mu$ and $\Sigma$. Randomly sample $10^4$ data points from the Gaussian distribution and set them as the training dataset, learn the NICE and RealNVP consist of LeakyReLU, and then randomly sample $10^4$ data points from the Gaussian distribution and $10^4$ data points from the abnormal distribution has a different mean from the normal distribution and set them as the test dataset. A simple likelihood

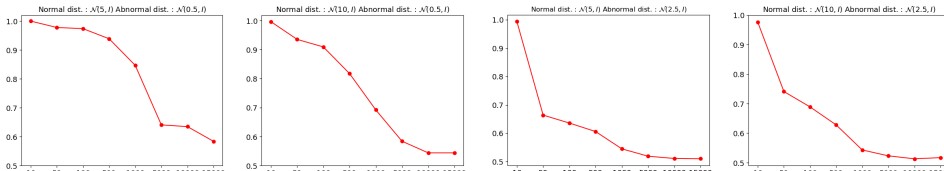

Figure 2: Performance of NF-SLT implemented by NICE across different dimensions. The y-axis represents AUROC, and the x-axis indicates the dimensionality of the data. The titles of each subfigure specify distributions' parameters. Subplots 1 and 2 have a larger In/Out of Distribution difference than Subplots 3 and 4, and the speed of performance decay as the dimension increases becomes faster as the distribution difference becomes smaller.

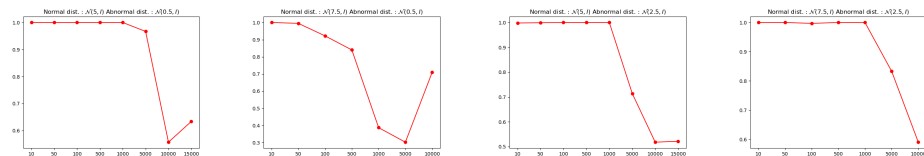

Figure 3: Performance of NF-SLT implemented by RealNVP across different dimensions. The y-axis represents AUROC, and the x-axis indicates the dimensionality of the data. The titles of each subfigure specify distributions' parameters.

test was performed, and the differences in AUROC scores according to dimensions using NICE are recorded in Figure 2, and for RealNVP in Figure 3. The default dimension setting in experiment was set to [10, 50, 100, 500, 1000, 5000, 10000, 15000]. Due to numerical stability issues, the experiment using RealNVP was conducted with the $\mu$ was set to 7.5 instead of 10, and when the $\mu$ of the in-distribution was 7.5, the experiment was not conducted with a dimension of 15000.

As shown in Figure 2, which is the experimental result using NICE, the performance in the 1st and 2nd subfigures degrades significantly between dimensions 1000 and 5000. The 3rd and 4th subfigures begin to degrade in performance at a dimension of 50, smaller than the first two figures, and continue to degrade as the dimension increases. Also we conducted an experiment by changing the means of the in-distribution and out-of-distribution, and confirmed that the AUROC was measured as 1 in most cases. Hence, we found that this phenomenon does not occur symmetrically, similar to the OOD detection experiments on CIFAR-10 and SVHN. In addition, Figure 3, an experiment using RealNVP, shows a similar pattern to Figure 2 in that AUROC decreases rapidly as the dimension increases, except for the second subplot. In Figure 3, the second subplot exhibits an inconsistent trend as dimensionality increases. This instability is likely due to the significant mean difference between in-distribution and out-of-distribution, which introduces numerical stability challenges. This observation underscores the critical impact of distribution alignment on performance scaling.

Additionally, we present a histogram of the log-likelihood of normal and abnormal data along the dimensions from each experiment corresponding to Figure 2 in Figure 4 to 7. The title of each figure indicates the dimension, the x-axis indicates the log-likelihood, and the y-axis indicates the number of data corresponding to the bin. In addition, the orange and blue histograms represent abnormal data and normal data, respectively. In experiments on RealNVP, to see how the volume term affects the likelihood, we visualized the latent norm and volume as histograms in the experiments for the first and third subplots of Figure 3 in Figures 8 to 11.

According to Equation 1 for input data $\mathbf{x}$, the log-likelihood $\log p(\mathbf{x})$ can be calculated with the log-likelihood of the latent $\mathbf{z}$ corresponding to $\mathbf{x}$ and the volume term for the distribution set as the prior of the flow. Through the histogram results, we can see that when the AUROC approaches 0.5 (i.e., dimension approaches to 10000), the latent norm becomes identical and the size of the volume term is reversed. This means that when calculating the log-likelihood with Equation 1, the volume term rather has a negative effect on the likelihood test, which is consistent with the results of Figure 4(a) and 4(b) of Nalisnick et al. (2019a). However, in Figures 8 to 11 , we can see that

the scale of the volume term is much smaller than the latent norm, which can be considered as the log-likelihood scale of the latent because $\mathcal{N}(0, I_d)$ is set as the prior of the flow in general and the log-likelihood is proportional to $-||\mathbf{z}||_2^2$. Therefore, the phenomenon that AUROC approaches 0.5 when the dimension increases can be interpreted as the fact that in the case of a volume-preserving model such as NICE, the latent norms become identical, and in the case of a non-volume-preserving model such as RealNVP, the latent norms become identical and the volume term shows a behavior that actually hinders the performance, but the effect of the volume term is so small.

From another perspective, let's express this case using Equation 4:

$$
\begin{aligned}
& \mathbb{E}_{\mathbf{x} \sim P}[\log P_\theta(\mathbf{x})] - \mathbb{E}_{\mathbf{x} \sim Q}[\log P_\theta(\mathbf{x})] \\
& = D_{KL}(Q||P_\theta) - D_{KL}(P||P_\theta) + \mathbb{H}(Q) - \mathbb{H}(P) \\
& = D_{KL}(Q||P_\theta) - D_{KL}(P||P_\theta) \ (\because \mathbb{H}(P) = \mathbb{H}(Q))
\end{aligned}
\tag{12}
$$

In this case, if $P_\theta$ is a perfect model, the likelihood gap is guaranteed to be positive, so it can be argued that the reason the term becomes negative is because $D_{KL}(P||P_\theta)$ eventually increases for high-dimensional data (i.e., $P_\theta$ is not a perfect model). Therefore, it can be argued that this problem occurs because it is difficult to approximate $P_\theta$ as a perfect model, such as in optimization problems in high-dimensional space. However, the important thing is that even this phenomenon does not occur symmetrically. Therefore, we argue that it is difficult to think that the reason for this phenomenon is simply an optimization problem that occurs in high dimensions.

Consequently, these results support the claim that the norm shown in Theorem C.7 of Section C.2 becomes almost identical in high dimensional space even when the distributions are not zero-centered. In addition, it can be confirmed that the phenomenon occurs regardless of whether the model has a constant volume term or not (i.e., whether the model is volume-preserving) so it can be thought that the presence or absence of a volume term does not help to alleviate the phenomenon. However, we did not observe this phenomenon in real data, and we confirmed that this phenomenon did not occur when the flow activation function was configured as a hyperbolic tangent rather than a ReLU-like function in flow. Therefore, we propose future research to investigate why norms become identical in certain cases when using ReLU-like functions in flow (or why anomalous data are mapped to isotropic and log-concave distributions based on Section 5.1) and to explore whether this phenomenon is reproduced in real data.

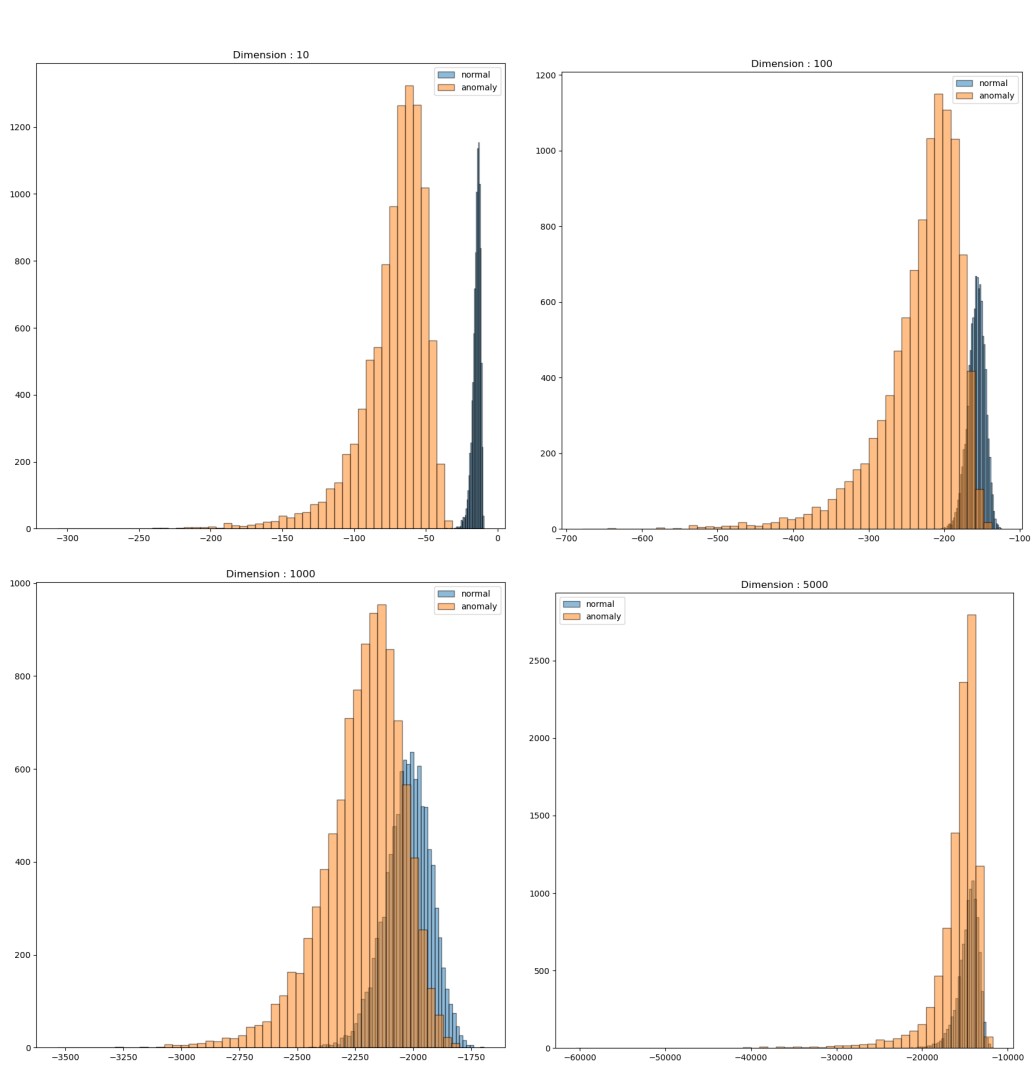

Figure 4: Histogram of log-likelihood values for the 1st subfigure in Figure 2

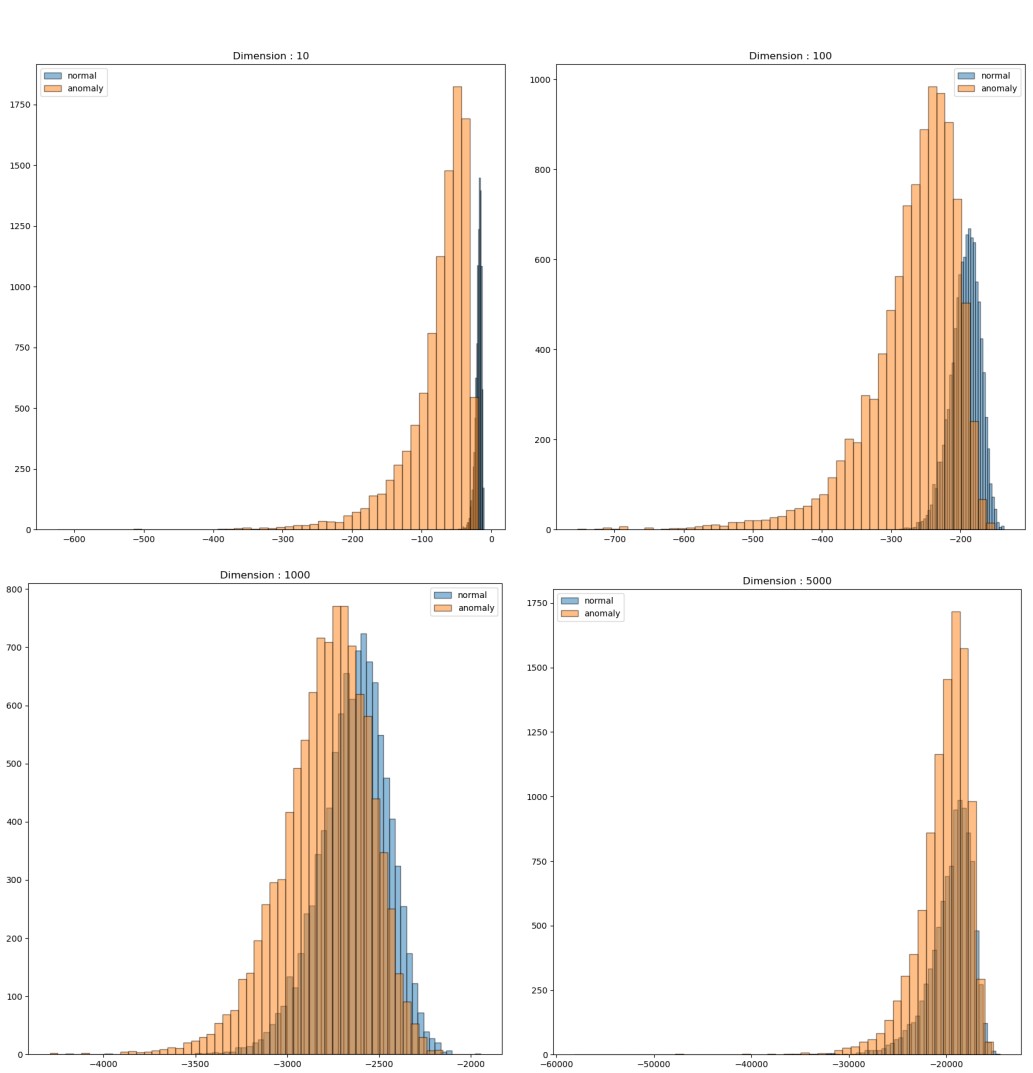

Figure 5: Histogram of log-likelihood values for the 2nd subfigure in Figure 2

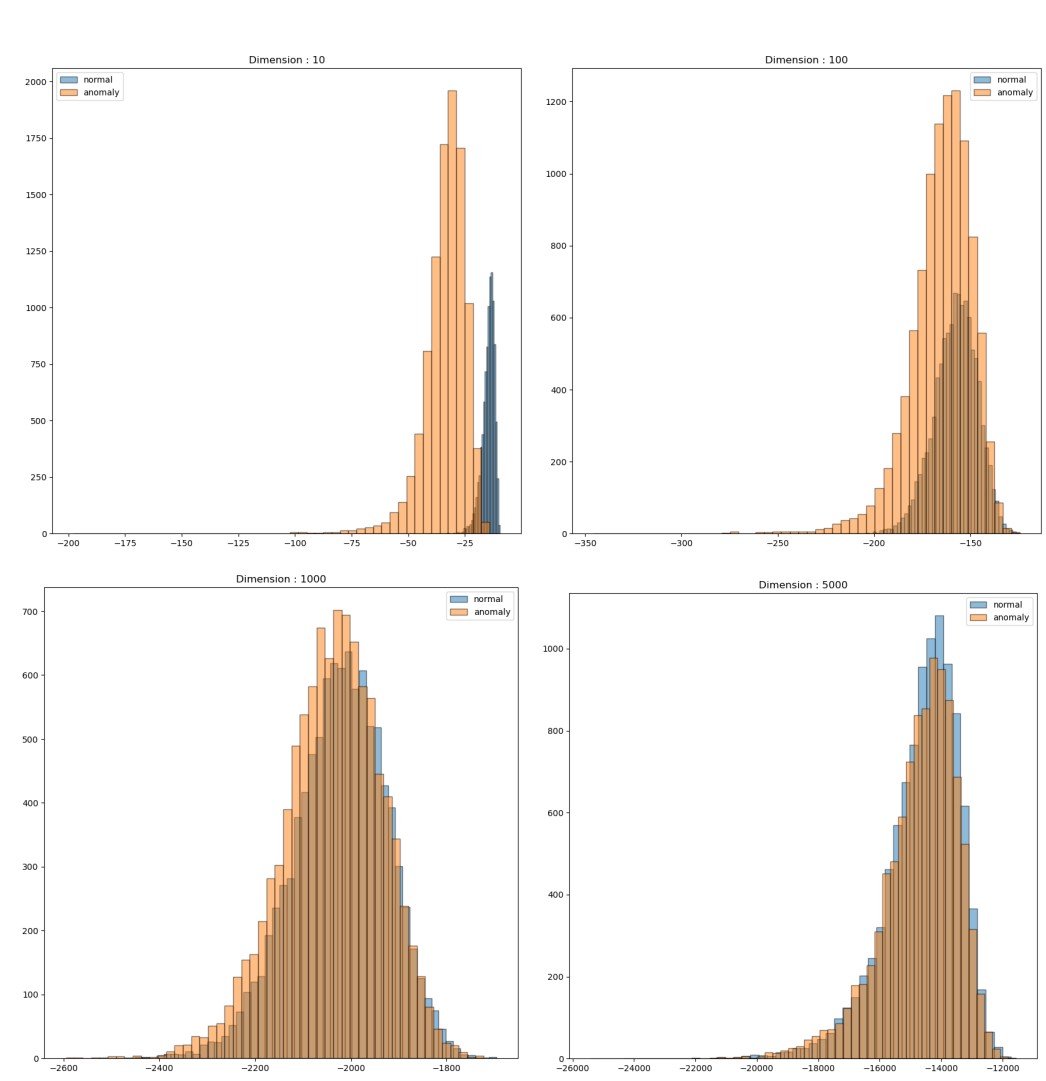

Figure 6: Histogram of log-likelihood values for the 3rd subfigure in Figure 2

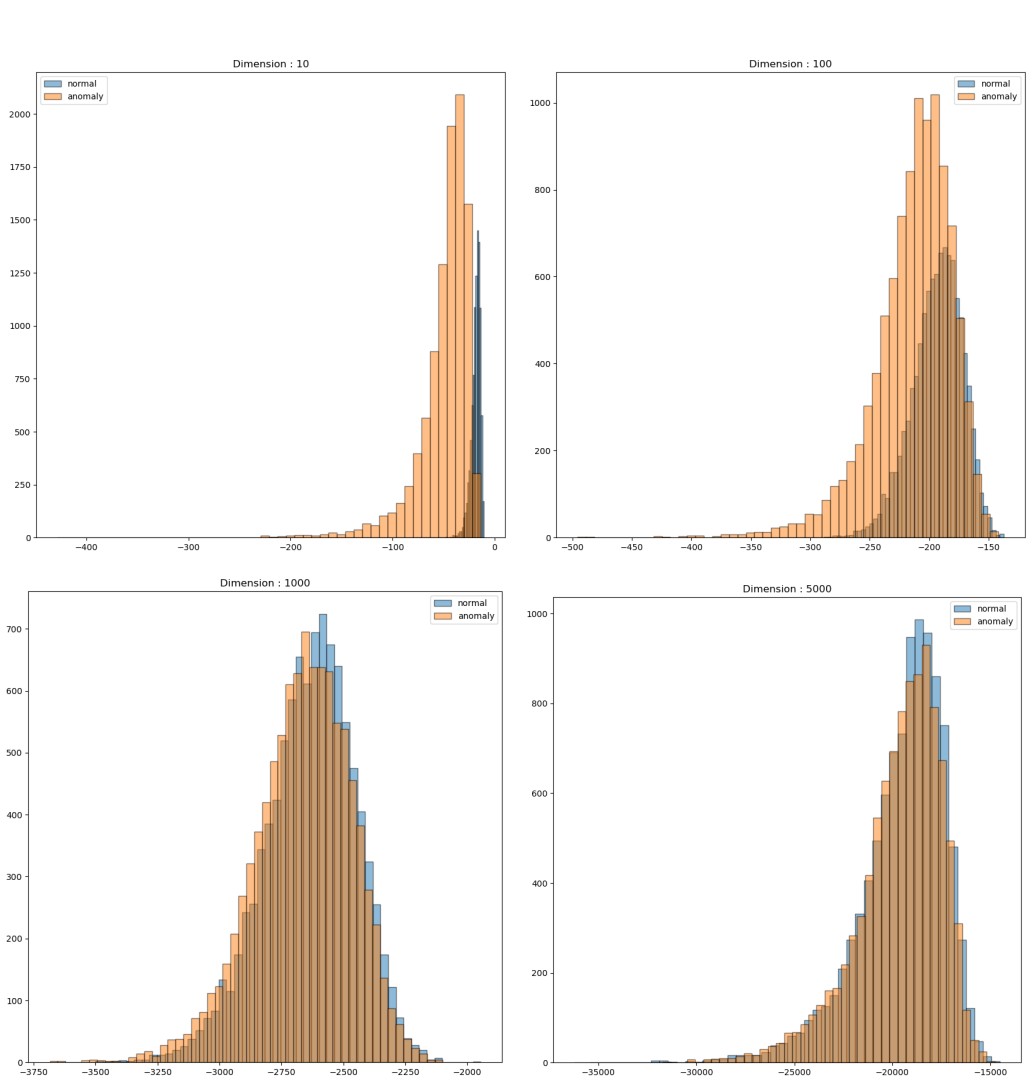

Figure 7: Histogram of log-likelihood values for the 4th subfigure in Figure 2

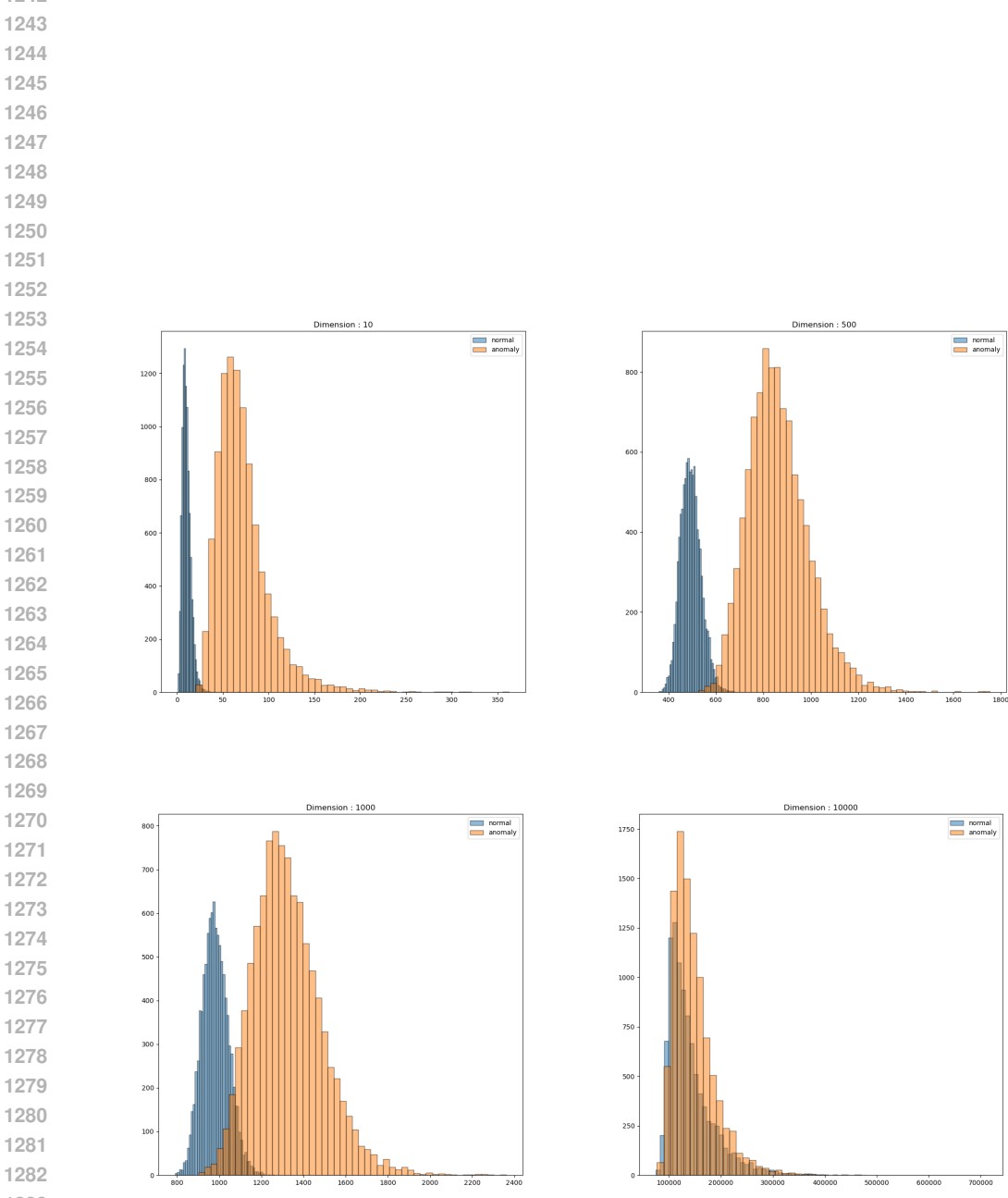

Figure 8: Histogram of latent norm values for the 1st subfigure in Figure 3

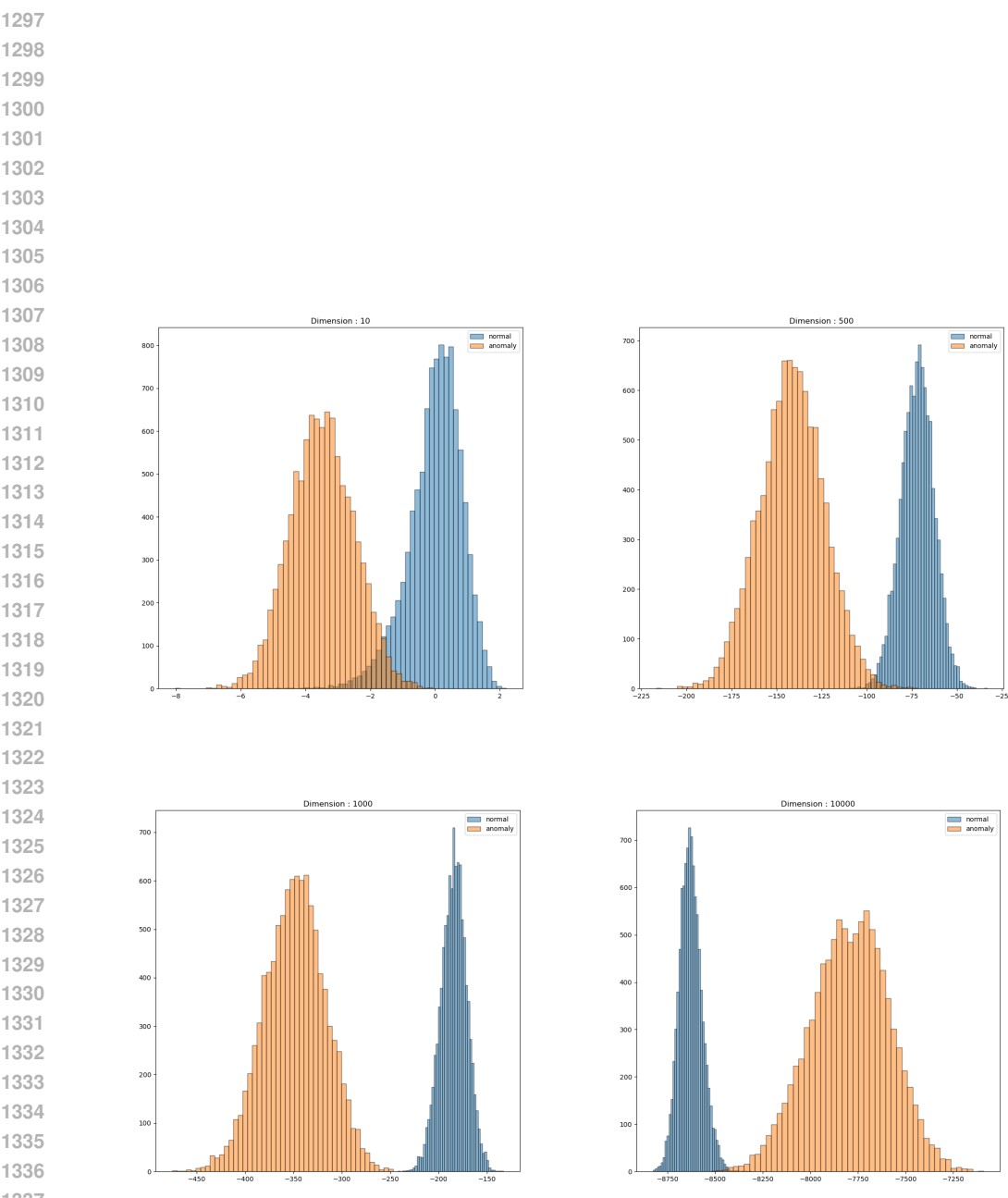

Figure 9: Histogram of volume values for the 1st subfigure in Figure 3

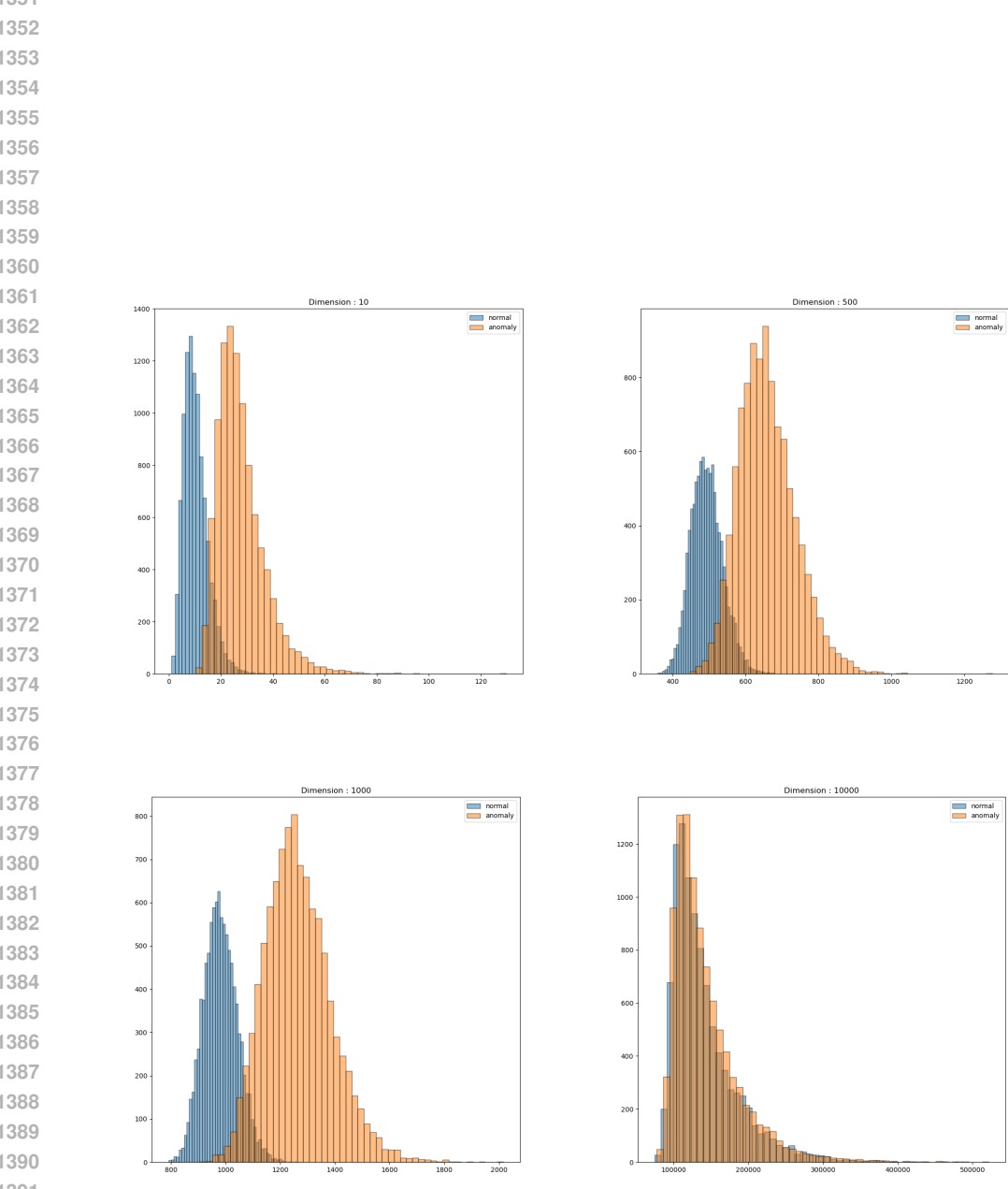

Figure 10: Histogram of latent norm values for the 3rd subfigure in Figure 3

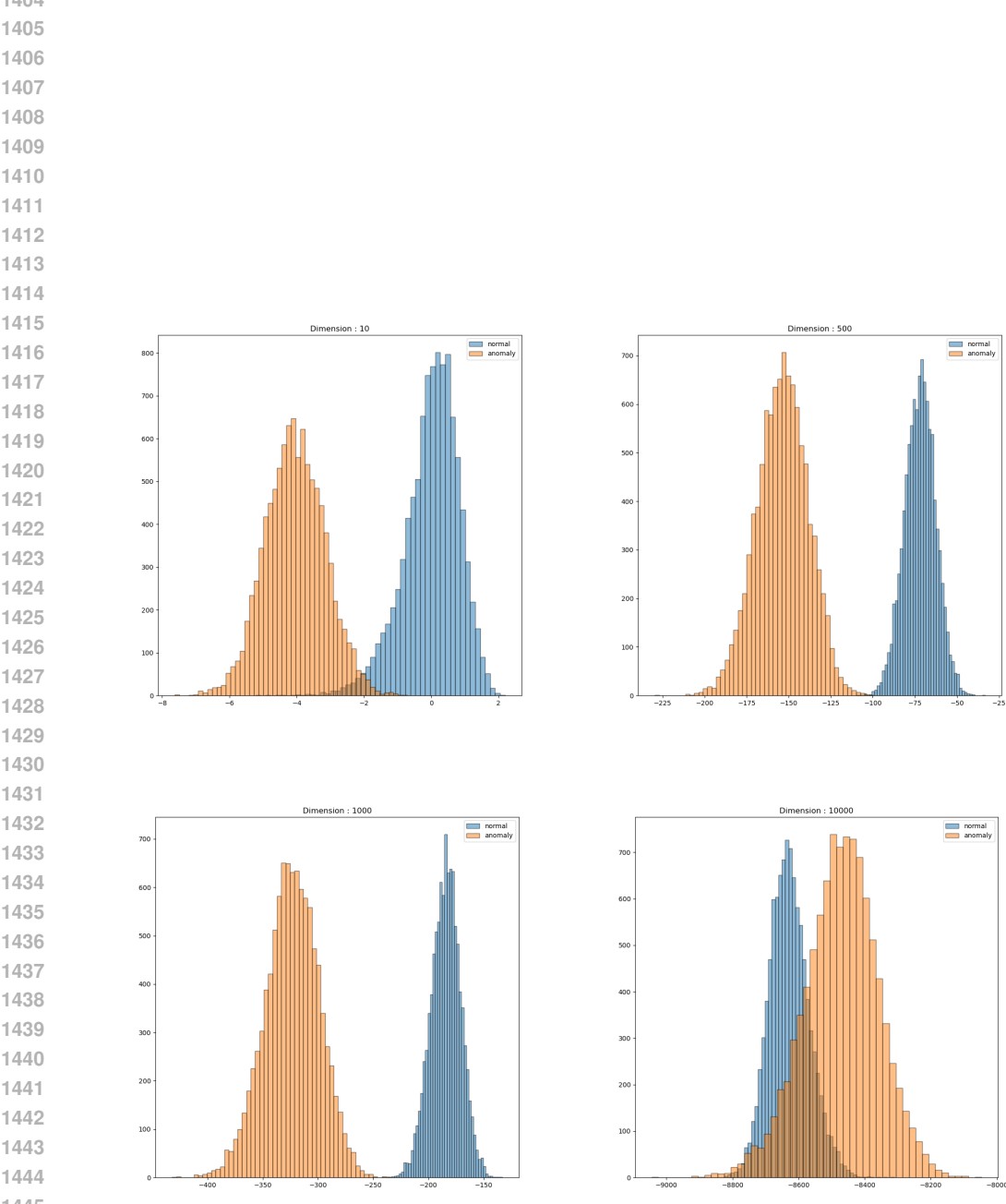

Figure 11: Histogram of volume values for the 3rd subfigure in Figure 3

## C.4 EXPERIMENTS OF SECTION 5.1 WITH REAL DATA

Table 5: AUROC scores for likelihood tests as a function of dimensionality (number of PCs) using RealNVP with MLP (image preprocessed by PCA). The region to the left of the bold vertical line indicates cases where $\mathbb{H}(P) > \mathbb{H}(Q)$, and the region to the right is the opposite.

| In-dist ($P$) / Out-dist ($Q$) | 1024 | 512 | 256 | 30 | In-dist ($P$) / Out-dist ($Q$) | 1024 | 512 | 256 | 30 |
|---|---|---|---|---|---|---|---|---|---|
| CIFAR-10 / SVHN | 0.1262 | 0.095 | 0.1729 | 0.3547 | SVHN / CIFAR-10 | 0.9932 | 0.9807 | 0.9460 | 0.8670 |
| CIFAR-100 / SVHN | 0.0668 | 0.0975 | 0.2015 | 0.3410 | SVHN / CIFAR-100 | 0.9926 | 0.9783 | 0.9436 | 0.8723 |
| CelebA / SVHN | 0.1598 | 0.2179 | 0.2988 | 0.4728 | SVHN / CelebA | 0.9969 | 0.9857 | 0.9709 | 0.9389 |
| CIFAR-100 / CIFAR-10 | 0.4804 | 0.5058 | 0.4982 | 0.5021 | CIFAR-10 / CIFAR-100 | 0.5415 | 0.5287 | 0.5257 | 0.5630 |
| CelebA / CIFAR-10 | 0.8424 | 0.8511 | 0.8344 | 0.6492 | CIFAR-10 / CelebA | 0.6350 | 0.5878 | 0.6747 | 0.7326 |

Table 6: Additional AUROC scores for likelihood tests as a function of dimensionality (number of PCs) using RealNVP with MLP (image preprocessed by ICA). The region to the left of the bold vertical line indicates cases where $\mathbb{H}(P) > \mathbb{H}(Q)$, and the region to the right is the opposite.

| In-dist ($P$) / Out-dist ($Q$) | 1024 | 512 | 256 | 30 | In-dist ($P$) / Out-dist ($Q$) | 1024 | 512 | 256 | 30 |
|---|---|---|---|---|---|---|---|---|---|
| CIFAR-10 / SVHN | 0.2311 | 0.2924 | 0.2984 | 0.3143 | SVHN / CIFAR-10 | 0.9917 | 0.9843 | 0.9486 | 0.8520 |
| CIFAR-100 / SVHN | 0.0843 | 0.1160 | 0.2036 | 0.3490 | SVHN / CIFAR-100 | 0.9933 | 0.9536 | 0.9137 | 0.8622 |
| CelebA / SVHN | 0.1207 | 0.1782 | 0.2745 | 0.4711 | SVHN / CelebA | 0.9976 | 0.9811 | 0.9722 | 0.9481 |
| CIFAR-100 / CIFAR-10 | 0.5133 | 0.5169 | 0.5046 | 0.5037 | CIFAR-10 / CIFAR-100 | 0.4970 | 0.5081 | 0.5146 | 0.5588 |
| CelebA / CIFAR-10 | 0.8511 | 0.8501 | 0.8364 | 0.6515 | CIFAR-10 / CelebA | 0.3988 | 0.4041 | 0.4692 | 0.6725 |

First, we will explain the experimental settings in Table 2, 5, 6 for experiment on RealNVP. In both experiments, the model used RealNVP with 16 coupling layers, and each coupling layer consisted of 2 layers, and the hidden dimension of the corresponding layer was set to 256. AdamW was used as the optimizer, and the learning rate was set to 5e-4 and the weight decay was set to 1e-4. The batch size was 256, the scheduler was CosineAnnealingWarmRestarts, and the period was applied as the entire epoch. The epoch was set to 300 for entire cases. The train and test datasets used the train and test datasets specified in advance in torchvision for each dataset. The implementation of the model is referenced from Stimper et al. (2023).

Although PCA does not satisfy the independence assumption of Theorem 5.4, as it does not preserve statistical independence across components, it remains a widely adopted dimensionality reduction method. Therefore, we evaluated the impact of dimensionality adjustment using PCA on AUROC, as presented in Tables 5. Additionally, Table 6 shows the additional result of Table 2. In this experiment, 16 Glow layer blocks were used, and a model consisting of 3 multi-scale layers was used. Other experimental settings are the same as Table 2.

In the results, it can be confirmed that the AUROC increases as the dimension decreases for the three case from the top above in Table 5 that satisfy the $\mathbb{H}(P) > \mathbb{H}(Q)$ condition. However, the two cases below in Table 5 and 6 do not show any tendency, which can be interpreted as not showing any specific tendency because the difference in entropy between the two distributions is not large. The reason why the difference in entropy between the two distributions is not large is based on the fact that the complexity histograms of the two images in Figure 2 of Serrà et al. (2020) mostly overlap.

Furthermore, we performed dimension reduction using PCA on the InternetAds dataset, which has the largest dimension ($d = 1555$) in ADBench, and checked the trend of performance changes. The results are included in Table 7.

Table 7: AUROC for InternetAds dataset of NF-SLT and MCM according to dimension change. The ratio represents the percentage of PCs used among 500 PCs, which is the result of reducing the dimension to 500 through PCA.

| Ratio | 1 | 0.95 | 0.9 | 0.85 | 0.8 | 0.75 |
|---|---|---|---|---|---|---|
| NF-SLT | 0.8180 | 0.8561 | 0.8579 | 0.8377 | 0.8246 | 0.8240 |
| MCM | 0.7242 | 0.7217 | 0.7189 | 0.7161 | 0.7158 | 0.6978 |

Through the results in Table 7, we can see that MCM's performance decreases due to information loss as the dimension decreases. However, NF-SLT's performance increases when the dimension is

reduced, and even when the ratio is reduced to 0.75, the performance is higher than when the entire PC is used. Surely, it is correct according to the theorem that when the dimension decreases, the performance increases and when the gap of entropy is large, it approaches 0.5. However, since it is possible for entropy to be generated from various distributions that are larger or smaller than the normal sample, we explain in Table 7 that not only the AUROC may exceed 0.5, but the performance trend may not increase steadily. This is the basis for the claim that performance degradation can occur even in tabular datasets when the dimension is large, and that performance can be good and counterintuitive phenomenon can rarely occur because the dimension of the tabular dataset is small.

In practice, extremely high-dimensional tabular datasets, such as genomic or metagenomic data, often contain tens of thousands of features. However, these datasets present specific challenges for anomaly detection (AD) due to their inherent feature homogeneity and extreme dimensional complexity. Unlike general tabular data, which typically includes heterogeneous features (e.g., categorical, continuous, nominal), genomic data is characterized by a large number of biologically similar variables with high mutual correlation and tightly constrained value distributions. This fundamental difference means that directly applying standard AD methods to raw genomic or metagenomic data is generally impractical without substantial preprocessing. For instance, metagenomic datasets often include tens of thousands of microbial species abundances or gene expression profiles, where each feature can be highly correlated with others due to shared biological pathways or taxonomic hierarchies. This extreme feature homogeneity can lead to severe overfitting and degraded AD performance if handled without prior dimensionality reduction. Therefore, standard bioinformatics workflows commonly incorporate domain-specific dimensionality reduction techniques, such as sparse Canonical Correlation Analysis (CCA) (Witten et al., 2009), Gene Set Variation Analysis (GSVA) (Hänzelmann et al., 2013), or biologically-informed network modeling (Wang et al., 2024), to project these high-dimensional inputs into lower-dimensional, interpretable representations. For example, Hänzelmann et al. (2013) demonstrated the use of GSVA to condense over 25,000 genes into biologically meaningful gene sets, significantly reducing the feature space and improving downstream analysis.

Additionally, a recent study by Wang et al. (2024) on BioNet highlighted that even deep learning models for high-dimensional biological data often rely on domain-specific preprocessing to improve interpretability and reduce overfitting. This approach integrates biological knowledge to guide feature selection, effectively reducing the dimensional complexity of the input data. Such preprocessing steps are critical for achieving meaningful predictions and robust model performance, as demonstrated in the context of glioblastoma heterogeneity assessment (Wang et al., 2024). Given these domain-specific preprocessing requirements, our study focuses on heterogeneous tabular data that include diverse feature types such as categorical, continuous, and nominal variables, rather than directly addressing raw, extremely high-dimensional genomic or metagenomic datasets. This choice aligns with the practical constraints observed in bioinformatics, where specialized preprocessing is essential for effective analysis of such data. Lastly, if there are heterogeneous high-dimensional tabular datasets that do not require such specialized preprocessing, we would be open to exploring them as part of future work, as our current study aims to provide a broadly applicable approach for anomaly detection across diverse real-world datasets.

## D  PROOF OF THEOREMS

First, we prove that when the entropy of a normal distribution $P$ is larger than the entropy of an abnormal distribution $Q$ and $P$, $Q$ are $d$-dimensional i.i.d. distributions, the expectation gap of likelihood estimated by the density estimation model $P_\theta$ trained on data sampled from the normal distribution $P$ increases as $d$ increases.

**Lemma 5.1.** *Let $P, P_\theta, Q$ be continuous probability distributions with its bounded probability distribution function $p, p_\theta, q$. If $p_\theta(x)$ converges to $p(x)$ pointwisely as $\theta \to \theta_0$, $Q$ has bounded support on the intersection of supports of $P$ and $P_{\theta_0}$, then $\lim_{\theta \to \theta_0} D_{KL}(Q||P_\theta) = D_{KL}(Q||P)$*

*Proof.*

$$D_{KL}(Q||P_\theta) = \int q(x) \log \frac{q(x)}{p_\theta(x)} dx$$

$$= \int q(x) \log \frac{q(x)}{p(x)} dx + \int q(x) \log \frac{p(x)}{p_\theta(x)} dx \qquad (13)$$

$$= D_{KL}(Q||P) + \int q(x) \log \frac{p(x)}{p_\theta(x)} dx$$

Thus, it suffices to show $\int q(x) \log \frac{p(x)}{p_\theta(x)} dx$ converges to 0 when $\theta$ converges to $\theta_0$.

Since $p, p_\theta$ are bounded and $Q$ has support on the intersection of supports of $P$ and $P_\theta$, there is constant $M$ such that $e^{-M} p_\theta(x) < p(x) < e^M p_\theta(x)$ for every $x \in Supp(Q)$. Thus, $\{\log \frac{p(x)}{p_\theta(x)}\}$ is uniformly integrable on the support of Q as $|\log \frac{p(x)}{p_\theta(x)}| < M$. Thus, $\lim_{\theta \to \theta_0} \int q(x) \log \frac{p(x)}{p_\theta(x)} dx = \int \lim_{\theta \to \theta_0} q(x) \log \frac{p(x)}{p_\theta(x)} dx$ by Vitali convergence theorem. Since $p_\theta(x)$ converges to $p(x)$ pointwisely, $\lim_{\theta \to \theta_0} \log \frac{p(x)}{p_\theta(x)} = 0$ and $\lim_{\theta \to \theta_0} \int q(x) \log \frac{p(x)}{p_\theta(x)} dx = 0$

$\square$

**Theorem 5.2** (Impact of Dimensionality on Likelihood Gap: i.i.d Case). *Let $P = \prod_{i=1}^d p(x_i)$ and $Q = \prod_{i=1}^d q(x_i)$ be identical and independent $d$-dimensional continuous probability density models in $\mathbb{R}^d$ with same conditions as Lemma 5.1. Let $P_\theta$ is well-trained density estimation model approximates $P$ (i.e., $p_\theta(x) \to p(x)$ pointwisely). If $\mathbb{H}(p) - \mathbb{H}(q) > D_{KL}(q||p)$, gap between the expectation of the likelihood for $P$ and $Q$ decreases linearly with respect to d.*

*Proof.* Gap between the expectation of the likelihood for $P$ and $Q$ can be defined as follows as expressed in Caterini & Loaiza-Ganem (2022).

$$\mathbb{E}_{\mathbf{x} \sim P}[\log p_\theta(\mathbf{x})] - \mathbb{E}_{\mathbf{x} \sim Q}[\log p_\theta(\mathbf{x})] \qquad (14)$$

Let $\mathbb{H}(P)$ be entropy of distribution $P$ and $D_{KL}(P||Q)$ be KL-divergence between $P$ and $Q$. Then, each term can be decomposed as follows:

$$\mathbb{E}_{\mathbf{x} \sim P}[\log P_\theta(\mathbf{x})] = -D_{KL}(P||P_\theta) - \mathbb{H}(P)$$
$$\mathbb{E}_{\mathbf{x} \sim Q}[\log P_\theta(\mathbf{x})] = -D_{KL}(Q||P_\theta) - \mathbb{H}(Q) \qquad (15)$$

So, we can derive as follow:

$$\mathbb{E}_{\mathbf{x} \sim P}[\log P_\theta(\mathbf{x})] - \mathbb{E}_{\mathbf{x} \sim Q}[\log P_\theta(\mathbf{x})]$$
$$= -D_{KL}(P||P_\theta) - \mathbb{H}(P) + D_{KL}(Q||P_\theta) + \mathbb{H}(Q)$$
$$= D_{KL}(Q||P_\theta) - D_{KL}(P||P_\theta) + \mathbb{H}(Q) - \mathbb{H}(P) \qquad (16)$$
$$\approx D_{KL}(Q||P_\theta) + \mathbb{H}(Q) - \mathbb{H}(P) \ (\because p_\theta(x) \to p(x) \text{ p.w.})$$

Then, since P and Q are distributed i.i.d.,

$$D_{KL}(Q||P_\theta) + \mathbb{H}(Q) - \mathbb{H}(P)$$
$$= D_{KL}(Q||P_\theta) + \mathbb{E}_{\mathbf{x} \sim Q}[-\log Q(\mathbf{x})] - \mathbb{E}_{\mathbf{x} \sim P}[-\log P(\mathbf{x})]$$
$$= D_{KL}(Q||P_\theta) + \mathbb{E}_{\mathbf{x} \sim Q}[-\log Q(x_1, x_2, ..x_n)] - \mathbb{E}_{\mathbf{x} \sim P}[-\log P(x_1, x_2, ...x_n)]$$
$$= D_{KL}(Q||P_\theta) + \mathbb{E}_{\mathbf{x} \sim Q}[-\log \prod_{i=1}^d q(x_i)] - \mathbb{E}_{\mathbf{x} \sim P}[-\log \prod_{i=1}^d p(x_i)]$$
$$\qquad (17)$$
$$= D_{KL}(Q||P_\theta) + \mathbb{E}_{\mathbf{x} \sim Q}[-\sum_{i=1}^d \log q(x_i)] - \mathbb{E}_{\mathbf{x} \sim P}[-\sum_{i=1}^d \log p(x_i)]$$
$$= D_{KL}(Q||P_\theta) + d(\mathbb{E}_{x \sim q}[-\log q(x)] - \mathbb{E}_{x \sim p}[-\log p(x)])$$
$$\approx D_{KL}(Q||P) + d(\mathbb{E}_{x \sim q}[-\log q(x)] - \mathbb{E}_{x \sim p}[-\log p(x)]) \ (\because \text{Lemma 5.1})$$
$$= d(D_{KL}(q||p) + \mathbb{E}_{x \sim q}[-\log q(x)] - \mathbb{E}_{x \sim p}[-\log p(x)])$$
$$= d(D_{KL}(q||p) - (\mathbb{E}_{x \sim p}[-\log p(x)] + \mathbb{E}_{x \sim q}[-\log q(x)]))$$

$\square$

Then, if $\mathbb{H}(p) - \mathbb{H}(q) > D_{KL}(q||p)$, gap between the expectation of the likelihood for $P$ and $Q$ decreases linearly with respect to $d$. Through Theorem 5.2, even if the generative model approximates the actual model well, the likelihood inversion phenomenon can occur if the difference in the entropy of $q$ and $p$ is larger than the KL-divergence of $q$ and $p$, and this issue escalates with an increase in dimensionality.

**Proposition 5.3** ($\mathbb{H}(P) - \mathbb{H}(Q) > D_{KL}(Q||P)$ for Gaussian Case). *Let $P$ and $Q$ be $\mathcal{N}(\mu_P, \sigma_P^2 I), \mathcal{N}(\mu_Q, \sigma_Q^2 I)$ such that $\sigma_P, \sigma_Q$ are non-zero constant. Then, $\mathbb{H}(P) - \mathbb{H}(Q) > D_{KL}(Q||P)$ holds if $||\mu_P - \mu_Q||^2 < d(\sigma_P^2 - \sigma_Q^2)$.*

*Proof.* Entropy of $P$ and $Q$ can be expressed as follows:

$$\mathbb{H}(P) = \frac{d}{2} \log(2\pi e \sigma_P^2)$$
$$\mathbb{H}(Q) = \frac{d}{2} \log(2\pi e \sigma_Q^2) \tag{18}$$

Therefore, we can derive entropy gap of $P$ and $Q$ below:

$$\mathbb{H}(P) - \mathbb{H}(Q) = \frac{d}{2} \log\left(\frac{\sigma_P^2}{\sigma_Q^2}\right) \tag{19}$$

Also, since the KL divergence of $Q$ and $P$ is known to be in closed form in the case of Gaussian, it can be expressed as follows:

$$D_{KL}(Q||P) = \frac{1}{2}\left[\sigma_P^{-2}(\mu_P - \mu_Q)^T(\mu_P - \mu_Q) + \log\left(\frac{\det \sigma_P^2 I}{\det \sigma_Q^2 I}\right) + \text{tr}(\sigma_P^{-2}\sigma_Q^2 I) - d\right]$$
$$= \frac{1}{2}\left[\frac{||\mu_P - \mu_Q||^2}{\sigma_P^2} + 2d\log\left(\frac{\sigma_P}{\sigma_Q}\right) + \frac{d\sigma_Q^2}{\sigma_P^2} - d\right] \tag{20}$$

Then, we can derive:

$$D_{KL}(Q||P) - \mathbb{H}(P) + \mathbb{H}(Q)$$
$$= \frac{1}{2}\left[\frac{||\mu_P - \mu_Q||^2}{\sigma_P^2} + 2d\log\left(\frac{\sigma_P}{\sigma_Q}\right) + \frac{d\sigma_Q^2}{\sigma_P^2} - d\right] - \frac{d}{2}\log\left(\frac{\sigma_P^2}{\sigma_Q^2}\right)$$
$$= \frac{1}{2}\left[\frac{||\mu_P - \mu_Q||^2}{\sigma_P^2} + 2d\log\left(\frac{\sigma_P}{\sigma_Q}\right) + \frac{d\sigma_Q^2}{\sigma_P^2} - d - 2d\log\left(\frac{\sigma_P}{\sigma_Q}\right)\right] \tag{21}$$
$$= \frac{1}{2}\left[\frac{||\mu_P - \mu_Q||^2}{\sigma_P^2} + \frac{d\sigma_Q^2}{\sigma_P^2} - d\right]$$
$$= \frac{1}{2}\left[\frac{||\mu_P - \mu_Q||^2}{\sigma_P^2} + d\left(\frac{\sigma_Q^2}{\sigma_P^2} - 1\right)\right]$$

Therefore, if $||\mu_P - \mu_Q||^2 < d(\sigma_P^2 - \sigma_Q^2)$, $D_{KL}(Q||P) - \mathbb{H}(P) + \mathbb{H}(Q) < 0$ □

Proposition 5.3 refers that if the variance of $P$ is sufficiently larger than the variance of $Q$ so that the difference between the variances of $P$ and $Q$ is greater than the norm of the mean difference, the inequality holds and likelihood inversion can happen. This is consistent with experimental results of Figure 5(a) of Nalisnick et al. (2019a) because for the CIFAR-10 vs SVHN case where likelihood inversion occurs, we can confirm that SVHN is concentrated on the pixel mean of CIFAR-10. Since it is impossible to accurately estimate entropy and KL divergence for the actual data distribution, it is not possible to confirm whether the inequality really holds, but it can be indirectly confirmed that the inequality is satisfied. Additionally, we extend Theorem 5.2 by relaxing the identically assumption.

**Theorem 5.4** (Impact of Dimensionality on Likelihood Gap). *Let $P = \prod_{i=1}^{d} p_i(x_i)$ and $Q = \prod_{i=1}^{d} q_i(x_i)$ be independent d-dimensional continuous probability density models in $\mathbb{R}^d$ with same conditions as Lemma 5.1. Let $P_\theta$ is well-trained density estimation model approximates $P$ (i.e., $p_\theta(x) \to p(x)$ pointwisely as $\theta \to \theta_0$). If $\mathbb{H}(P) - \mathbb{H}(Q) > D_{KL}(Q||P)$, the lower bound of gap between the expectation of the likelihood for $P$ and $Q$ decreases linearly with respect to $d$.*

*Proof.* Similarly proof of Theorem 5.2, we can derive as follow:

$$
\begin{aligned}
&\mathbb{E}_{\mathbf{x}\sim P}[\log P_\theta(\mathbf{x})] - \mathbb{E}_{\mathbf{x}\sim Q}[\log P_\theta(\mathbf{x})] \\
&= -D_{KL}(P||P_\theta) - \mathbb{H}(P) + D_{KL}(Q||P_\theta) + \mathbb{H}(Q) \\
&= D_{KL}(Q||P_\theta) - D_{KL}(P||P_\theta) + \mathbb{H}(Q) - \mathbb{H}(P) \\
&\approx D_{KL}(Q||P_\theta) + \mathbb{H}(Q) - \mathbb{H}(P) \; (\because p_\theta(x) \to p(x) \text{ p.w.}) \\
&= D_{KL}(Q||P_\theta) + \mathbb{E}_{\mathbf{x}\sim Q}[-\log Q(\mathbf{x})] - \mathbb{E}_{\mathbf{x}\sim P}[-\log P(\mathbf{x})] \\
&= D_{KL}(Q||P_\theta) + \mathbb{E}_{\mathbf{x}\sim Q}[-\log Q(x_1, x_2, ..x_n)] - \mathbb{E}_{\mathbf{x}\sim P}[-\log P(x_1, x_2, ...x_n)] \\
&= D_{KL}(Q||P_\theta) + \mathbb{E}_{\mathbf{x}\sim Q}[-\log \prod_{i=1}^{d} q(x_i)] - \mathbb{E}_{\mathbf{x}\sim P}[-\log \prod_{i=1}^{d} p(x_i)] \\
&= D_{KL}(Q||P_\theta) + \mathbb{E}_{\mathbf{x}\sim Q}[-\sum_{i=1}^{d} \log q(x_i)] - \mathbb{E}_{\mathbf{x}\sim P}[-\sum_{i=1}^{d} \log p(x_i)] \\
&= D_{KL}(Q||P_\theta) + \mathbb{E}_{\mathbf{x}\sim Q}[-\sum_{i=1}^{d} \log q(x_i)] - \mathbb{E}_{\mathbf{x}\sim P}[-\sum_{i=1}^{d} \log p(x_i)] \\
&= D_{KL}(Q||P_\theta) + \sum_{i=1}^{d}(\mathbb{E}_{x_i\sim q_i}[-\log q_i(x_i)] - \mathbb{E}_{x_i\sim p_i}[-\log p_i(x_i)]) \\
&\approx D_{KL}(Q||P) + \sum_{i=1}^{d}(\mathbb{E}_{x_i\sim q_i}[-\log q_i(x_i)] - \mathbb{E}_{x_i\sim p_i}[-\log p_i(x_i)]) \; (\because \text{Lemma 5.1})
\end{aligned}
\tag{22}
$$

Then, we define $k$ as follows:

$$
k = \underset{i\in[d]}{\arg\min} \; \mathbb{E}_{x_i\sim q_i}[-\log q_i(x_i)] - \mathbb{E}_{x_i\sim p_i}[-\log p_i(x_i)]
\tag{23}
$$

Because of $\mathbb{H}(P) - \mathbb{H}(Q) > D_{KL}(Q||P) \geq 0$, $\mathbb{H}(Q) - \mathbb{H}(P)$ has negative value. Therefore, there exists $\exists j \in [d]$ such that $\mathbb{E}_{x_j\sim q_j}[-\log q_j(x_j)] - \mathbb{E}_{x_j\sim p_j}[-\log p_j(x_j)]$ is negative. So,

$$
\begin{aligned}
0 &> \mathbb{E}_{x_j\sim q_j}[-\log q_j(x_j)] - \mathbb{E}_{x_j\sim p_j}[-\log p_j(x_j)] \\
&\geq \mathbb{E}_{x_k\sim q_k}[-\log q_k(x_k)] - \mathbb{E}_{x_k\sim p_k}[-\log p_k(x_k)]
\end{aligned}
\tag{24}
$$

In conclusion, Equation 24 gives us the following lower bound:

$$
\begin{aligned}
0 &> D_{KL}(Q||P) + \sum_{i=1}^{d}(\mathbb{E}_{x_i\sim q_i}[-\log q_i(x_i)] - \mathbb{E}_{x_i\sim p_i}[-\log p_i(x_i)]) \\
&\geq \sum_{i=1}^{d}(\mathbb{E}_{x_i\sim q_i}[-\log q_i(x_i)] - \mathbb{E}_{x_i\sim p_i}[-\log p_i(x_i)]) \\
&\geq d(\underset{i\in[d]}{\min} \mathbb{E}_{x_i\sim q_i}[-\log q_i(x_i)] - \mathbb{E}_{x_i\sim p_i}[-\log p_i(x_i)]) \\
&= d(\mathbb{E}_{x_k\sim q_k}[-\log q_k(x_k)] - \mathbb{E}_{x_k\sim p_k}[-\log p_k(x_k)])
\end{aligned}
\tag{25}
$$

$\square$

Through Theorem 5.4, if $\mathbb{H}(P) - \mathbb{H}(Q) > D_{KL}(Q||P)$ is satisfied, we can see that the lower bound of the gap of the likelihood estimated by $p_\theta$ for $\mathbf{x}$ that follows $P$ and $Q$ decreases linearly with the dimension. Furthermore, we show in Corollary 5.6 that the upper bound of the likelihood gap can be made smaller in dimension proportional to Theorem 5.4 by making additional assumptions.

**Corollary 5.5** (Special Case of Theorem 5.4: Guaranteed Negative Difference). *Let $P = \prod_{i=1}^{d} p_i(x_i)$ and $Q = \prod_{i=1}^{d} q_i(x_i)$ be independent $d$-dimensional continuous probability density models in $\mathbb{R}^d$ with same conditions as Lemma 5.1. Let $P_\theta$ is well-trained density estimation model approximates $P$ (i.e., $p_\theta(x) \to p(x)$ pointwisely as $\theta \to \theta_0$). If $\forall i \in [d]\ \mathbb{H}(p_i) - \mathbb{H}(q_i) > D_{KL}(q_i||p_i)$, the lower and upper bound of gap between the expectation of the likelihood for $P$ and $Q$ decreases linearly with respect to $d$.*

*Proof.* We define $k$ as follows:

$$k = \arg\max_{i \in [d]} \mathbb{E}_{x_i \sim q_i}[-\log q_i(x_i)] - \mathbb{E}_{x_i \sim p_i}[-\log p_i(x_i)] \tag{26}$$

Because of $\mathbb{H}(p_i) - \mathbb{H}(q_i) > D_{KL}(q_i||p_i) \geq 0$ for $\forall i \in [d]$, $\mathbb{H}(q_i) - \mathbb{H}(p_i) + D_{KL}(q_i||p_i)$ has negative value. So,

$$\begin{aligned}
0 &> D_{KL}(q_k||p_k) + \mathbb{E}_{x_k \sim q_k}[-\log q_k(x_k)] - \mathbb{E}_{x_k \sim p_k}[-\log p_k(x_k)] \\
&\geq D_{KL}(q_j||p_j) + \mathbb{E}_{x_j \sim q_j}[-\log q_j(x_j)] - \mathbb{E}_{x_j \sim p_j}[-\log p_j(x_j)]
\end{aligned} \tag{27}$$

Then, we can derive:

$$\begin{aligned}
&\mathbb{E}_{\mathbf{x} \sim P}[\log P_\theta(\mathbf{x})] - \mathbb{E}_{\mathbf{x} \sim Q}[\log P_\theta(\mathbf{x})] \\
&\approx D_{KL}(Q||P) + \sum_{i=1}^{d}(\mathbb{E}_{x_i \sim q_i}[-\log q_i(x_i)] - \mathbb{E}_{x_i \sim p_i}[-\log p_i(x_i)]) \\
&= \sum_{i=1}^{d}(D_{KL}(q_i||p_i) + \mathbb{E}_{x_i \sim q_i}[-\log q_i(x_i)] - \mathbb{E}_{x_i \sim p_i}[-\log p_i(x_i)]) \\
&\leq d(\max_{i \in [d]} D_{KL}(q_i||p_i) + \mathbb{E}_{x_i \sim q_i}[-\log q_i(x_i)] - \mathbb{E}_{x_i \sim p_i}[-\log p_i(x_i)]) \\
&= d(D_{KL}(q_k||p_k) + \mathbb{E}_{x_k \sim q_k}[-\log q_k(x_k)] - \mathbb{E}_{x_k \sim p_k}[-\log p_k(x_k)]) \\
&< 0
\end{aligned} \tag{28}$$

$\square$

The results of Corollary 5.5 confirm that the lower and upper bounds of the expectation gap decrease with dimension, and we show in Corollary 5.6 that the upper bound of AUROC can decrease with dimension if we include additional assumption on moment.

**Corollary 5.6** (Dimensionality and AUROC Upper Bound). *Building on the assumptions of Corollary 5.5, suppose the $n$-th absolute central moment of the log-likelihood difference, $\log p_\theta(Y) - \log p_\theta(X)$, scales as $\mathcal{O}(d^k)$ for some $n > 1$ and $k < n$. In this case, if the average log-likelihood gap becomes negative, the maximum achievable AUROC for distinguishing samples from $P$ and $Q$ is inversely related to the dimensionality $d$. This indicates that as the dimension increases, the likelihood test becomes fundamentally less effective at separating normal and abnormal samples.*

*Proof.* By Zhang et al. (2021), AUROC of likelihood test is defined by

$$\text{AUROC} = \Pr(\log p_\theta(X) > \log p_\theta(Y)),\ X \sim P,\ Y \sim Q \tag{29}$$

Let $\mu = \mathbb{E}_{X \sim P, Y \sim Q}[\log p_\theta(Y) - \log p_\theta(X)] > 0$. Then we can derive:

$$\begin{aligned}
&1 - \text{AUROC} \\
&= \Pr(\log p_\theta(X) < \log p_\theta(Y)) \\
&= \Pr(\log p_\theta(Y) - \log p_\theta(X) > 0) \\
&= \Pr(\mu - (\log p_\theta(Y) - \log p_\theta(X)) < \mu) \\
&\geq \Pr(|\mu - (\log p_\theta(Y) - \log p_\theta(X))| < \mu) \\
&\geq 1 - \frac{\mathbb{E}[|\log p_\theta(Y) - \log p_\theta(X) - \mu|^n]}{\mu^n} \quad (\because \text{polynomial Markov inequality})
\end{aligned} \tag{30}$$

So, AUROC bounded above:

$$\text{AUROC} \leq \frac{\mathbb{E}[|\log p_\theta(Y) - \log p_\theta(X) - \mu|^n]}{\mu^n} \tag{31}$$

Since we know from Corollary 5.5 that upper bound of $-\mu = \mathbb{E}_{X \sim P, Y \sim Q}[\log p_\theta(X) - \log p_\theta(Y)]$ decreases linearly in $d$, we can derive the following inequality:

$$
\begin{aligned}
\mu &= \mathbb{E}_{X \sim P, Y \sim Q}[\log p_\theta(Y) - \log p_\theta(X)] \\
&\geq -d(\max_{i \in [d]} D_{KL}(q_i||p_i) + \mathbb{E}_{x_i \sim q_i}[-\log q_i(x_i)] - \mathbb{E}_{x_i \sim p_i}[-\log p_i(x_i)]) \\
&= d(\min_{i \in [d]} \mathbb{E}_{x_i \sim p_i}[-\log p_i(x_i)] - \mathbb{E}_{x_i \sim q_i}[-\log q_i(x_i)] - D_{KL}(q_i||p_i)) \\
&> 0
\end{aligned}
\tag{32}
$$

Then, we can obtain inequality below:'

$$\frac{1}{\mu^n} \leq \frac{1}{d^n(\min_{i \in [d]} \mathbb{E}_{x_i \sim p_i}[-\log p_i(x_i)] - \mathbb{E}_{x_i \sim q_i}[-\log q_i(x_i)] - D_{KL}(q_i||p_i))^n} \tag{33}$$

Therefore, $\exists M \in \mathbb{R}$ such that

$$
\begin{aligned}
\text{AUROC} &\leq \frac{\mathbb{E}[|\log p_\theta(Y) - \log p_\theta(X) - \mu|^n]}{d^n(\min_{i \in [d]} \mathbb{E}_{x_i \sim p_i}[-\log p_i(x_i)] - \mathbb{E}_{x_i \sim q_i}[-\log q_i(x_i)] - D_{KL}(q_i||p_i))^n} \\
&= \frac{\mathcal{O}(d^k)}{d^n(\min_{i \in [d]} \mathbb{E}_{x_i \sim p_i}[-\log p_i(x_i)] - \mathbb{E}_{x_i \sim q_i}[-\log q_i(x_i)] - D_{KL}(q_i||p_i))^n} \\
&\leq \frac{d^{k-n}|M|}{(\min_{i \in [d]} \mathbb{E}_{x_i \sim p_i}[-\log p_i(x_i)] - \mathbb{E}_{x_i \sim q_i}[-\log q_i(x_i)] - D_{KL}(q_i||p_i))^n}
\end{aligned}
\tag{34}
$$

Because $k - n < 0$, upper bound of AUROC is inversely proportional to dimension. $\qquad\square$

We note that the assumption that absolute $n$-th central moment of $\log p_\theta(Y) - \log p_\theta(X)$ is $\mathcal{O}(d^k)$ such that $k < n$ is not a strong assumption. For example, $X$ and $Y$ are isotropic Gaussian, $\log p_\theta(X)$ is a form that adds a constant to a variable that follows a Chi-square distribution with $d$ degrees of freedom. Then, the second central moment of Chi-square distribution with $d$ degrees of freedom, so second moment of $\log p_\theta(Y) - \log p_\theta(X)$ satisfies $\mathcal{O}(d)$ ($\because \text{Var}(X - Y) = \text{Var}(X) + \text{Var}(Y)$), and most distributions satisfy this condition. In this example, it is possible to satisfy the assumption of Corollary 5.6 because the variance increases less than $\mathcal{O}(d^2)$ with respect to dimension. In addition, the assumption is satisfied if at least one of the $n$-th absolute central moments the growth is slower than $\mathcal{O}(d^n)$. Therefore, using this assumption does not impose many restrictions, since it is satisfied except in extreme cases (e.g., when a term in the distribution function includes $d$).

# E  DETECTION PERFORMANCE FOR VARIOUS ANOMALY TYPES & CATEGORICAL DATA

Table 8: Anomaly type detection performance for deep models

|  | Metrics | GOAD | DeepSVDD | NeutraLAD | ICL | MCM | NF-SLT |
|---|---|---|---|---|---|---|---|
| Local | AUROC | 0.7297 | 0.9605 | 0.9664 | 0.9646 | **0.9948** | 0.9919 |
|  | AUPRC | 0.7922 | 0.9678 | 0.9702 | 0.9697 | **0.9953** | 0.9926 |
| Clustered | AUROC | 1 | 1 | 0.9174 | 1 | 1 | 1 |
|  | AUPRC | 1 | 1 | 0.8620 | 1 | 1 | 1 |
| Global | AUROC | 1 | 1 | 1 | 1 | 1 | 1 |
|  | AUPRC | 1 | 1 | 1 | 1 | 1 | 1 |
| Dependency | AUROC | 0.9986 | 0.9960 | 0.9974 | 0.9997 | 0.9990 | **0.9999** |
|  | AUPRC | 0.9983 | 0.9953 | 0.9966 | 0.9996 | 0.9983 | **1** |

We apply the protocol for generating four types of anomalies suggested in Han et al. (2022) to the real dataset Satellite to verify whether NF-SLT can detect various types of anomalies well. Descriptions of the four types of anomalies and how synthetic anomaly data were generated are as follows:

- **Local anomalies** are data points that significantly differ from others in their immediate surroundings. After generating normal data using Gaussian Mixture Model (GMM, Milligan (1985); Steinbuss & Böhm (2021)), local anomalies are generated by multiplying the covariance by the scaling factor $\alpha = 5$.

- **Global anomalies** are data points distributed widely across the space but remain distinctly separate from the normal data distribution. Normal data is created in the same way as when creating local anomalies, and global anomalies are sampled from uniform distribution $Unif(\alpha \cdot \min(X^k), \alpha \cdot \max(X^k))$ with the scaling factor $\alpha$ set to 1.1. $\mathbf{X^k}$ refers to the $k$-th feature and creates anomalies by adjusting the maximum and minimum boundary values of the feature.

- **Dependency anomalies** are data points that fail to conform to the usual relationships or dependencies observed in normal data. Create normal data by capturing the dependency between features of normal data using Vine Copula (Aas et al., 2009). By estimating the probability density function of each feature through Kernel Density Estimation (KDE, Hastie (2009)) and dependency anomalies with independent features are created.

- **Clustered anomalies** are groups of similar data points that collectively differ greatly from the normal data distribution. Normal data is created in the same way as when creating local anomalies, clustered anomalies are generated by multiplying the mean vector estimated through GMM by the scaling factor $\alpha = 5$.

The comparison models were tested using the six deep models used as comparison models in the main experiment of the paper, and the performance was measured using AUROC and AUPRC. According to the results of Table 8, it can be confirmed that NF-SLT has the best detection performance among deep models in all anomaly types except for local anomaly types. In addition, since it shows performance next to MCM even for local anomaly types, the results verify the performance of NF-SLT in detecting various anomaly types.

Additionally, we evaluated NF-SLT on datasets with significant categorical composition using simple preprocessing (e.g., one-hot encoding). We conducted experiments using categorical-rich datasets from ADBench (InternetAds, campaign, census) and ODDs (nsl-kdd)(Rayana, 2016), and scaled the data with sklearn's RobustScaler for the datasets except nsl-kdd. As in the previous experiments, no scaling was performed for NeuTraLAD. The results for this experiment are shown in Table 9, and the results show that NF-SLT performs well even on datasets where categorical features dominate.

Table 9: AUROC for datasets with mostly categorical features. The numbers next to AUROC are the relative ranks for the 6 deep models.

| Dataset | DeepSVDD | GOAD | NeuTraIAD | ICL | MCM | NF-SLT |
|---------|----------|------|-----------|-----|-----|--------|
| InternetAds | 0.6013 / 6 | 0.7356 / 4 | 0.7112 / 5 | **0.8815 / 1** | 0.7973 / 3 | 0.8746 / 2 |
| campaign | 0.7929 / 3 | 0.2837 / 6 | 0.7814 / 4 | 0.6933 / 5 | 0.7987 / 2 | **0.8059 / 1** |
| census | 0.6421 / 3 | 0.4140 / 5 | 0.4991 / 4 | 0.3917 / 6 | **0.7478 / 1** | 0.7186 / 2 |
| nsl-kdd | 0.8951 / 4 | 0.9026 / 3 | 0.9374 / 2 | 0.8798 / 5 | 0.8757 / 6 | **0.9598 / 1** |

## F    MODEL DETAILS AND HYPERPARAMETER SEARCHING SPACE

Anomaly detection in tabular data has garnered significant attention due to its relevance in various real-world applications, such as fraud detection, medical diagnosis, etc. (Yin et al., 2024). Anomaly detection tasks often suffer from imbalance problems; hence several unsupervised methods have been worked because they have the advantage of not having from this problem. The learning method of anomaly detection varies depending on the presence of a label; however, this section only introduces unsupervised methods that learn without having information about the label. Because existing approaches can be broadly categorized into shallow models and deep models, it is divided into two categories and explained in the corresponding sections.

**Shallow Models** Principal Component Analysis (PCA, Wold et al. (1987)) is a dimension reduction method that projects data to a lower dimension than the original using Singular Value Decomposition (SVD). This algorithm can also be utilized as an anomaly detection method by using the reconstruction error when the latent vector projected in low dimensions is reconstructed to the original data as an anomaly score (Shyu et al., 2003).The Local Outlier Factor (LOF, (Breunig et al., 2000)) is a method that adopts neighbors to determine anomaly scores from a local perspective. Isolation forest (IF, (Liu et al., 2008) uses the concept that normal data is easier to isolate than outliers, one-class SVM(OCSVM, (Schölkopf et al., 1999)) determines the decision boundary by determining a support vector that can sufficiently explain the given training data well, and determines that data that exist outside this boundary are outliers. COPOD (Li et al., 2020) and ECOD (Li et al., 2022) are similar anomaly detection algorithms that utilizing the Empirical Cumulative Distribution Function (ECDF), and measure anomaly scores using the extremeness of input data under ECDF.

**Deep Models** DAGMM (Zong et al., 2018) utilizes a deep autoencoder to generate a low-dimensional representation and reconstruction error for each input data point, which is further fed into a Gaussian Mixture Model (GMM). DeepSVDD (Ruff et al., 2018) trains to bring the input data representation closer to the predefined center using a neural network, sets the boundary of the hypersphere and determines that data existing outside this are outliers. GOAD (Bergman & Hoshen, 2020) is a model that generalizes the transformation-based self-supervised method used in the image domain and applies it to the tabular domain. NeuTraLAD (Qiu et al., 2021) and ICL (Shenkar & Wolf, 2022) applied the contrastive learning method, one of the self-supervised methods, to the tabular domain to improve performance. DO2HSC (Zhang et al., 2024) improves the limitations of DeepSVDD with a hypersphere assumption through orthogonal projection and double hypersphere decision boundary. MCM (Yin et al., 2024) performs anomaly detection by extending the masking self-supervised method adopted in the NLP or image domain to the tabular domain. In addition, DO2HSC (Zhang et al., 2024) was excluded from the experiment because, after checking their implementation code, it was confirmed that the model uses the statistics of the test data when performing orthogonal projection, which was judged to be biased when compared to other models. NPT-AD (Thimonier et al., 2024) was excluded from the experiments due to its excessively long training time for high-dimensional datasets with numerous data points.

Additionally, we report details of the 13 models we implemented and the hyperparameter search space for each model. Because it would be impractical to perform a hyperparameter search for every hyperparameter that exists per model, we have selected the hyperparameters that we believe are important per model. For each hyperparameter searching space, we set commonly used hyperparameter values, ran experiments 10 times on 47 tabular datasets provided by ADBench for all combinations of hyperparameters, and selected the hyperparameter combination with the highest average AUROC for all datasets as a representative hyperparameter that demonstrates the performance of the model. In Table 10, we record the hyperparameter search space used in our experiments, and in Table 11, we record the optimal hyperparameter combination finally selected from the hyperparameter search space. In the description of the deep model where the implementation code resides, we only specify the hyperparameters we modified.

**PCA** We implemented PCA via the PyOD library, and chose n_component_ratio as the hyperparameter to explore. n_component_ratio is a hyperparameter for what percentage of the dimensionality of the original input data should be reduced.

**LOF** We implemented LOF via the PyOD library and chose $N$ as the hyperparameter to explore. $N$ is a hyperparameter for how many neighbors to consider.

**IF** We implemented IF via the PyOD library and chose $N$ as the hyperparameter to explore. $N$ is the hyperparameter for how many trees to ensemble.

**OCSVM** We have implemented OCSVM via the scikit-learn library and have chosen kernel as the hyperparameter to explore. Kernel is the hyperparameter for which function to choose as the kernel function of OCSVM.

**COPOD** We implemented COPOD via the PyOD library and did not set a hyperparameter searching space because the model is hyperparameter-free.

**ECOD** We implemented ECOD via the PyOD library and did not set a hyperparameter searching space because the model is hyperparameter-free like COPOD.

**DAGMM** We ran DAGMM experiments based on the model implemented in `https://github.com/mperezcarrasco/PyTorch-DAGMM`. We set the batch size to 512, the epoch to 200. We chose $n_{gmm}$, $\lambda_{energy}$, and $\lambda_{cov}$ as hyperparameters to explore. $n_{gmm}$ is the hyperparameter for how many Gaussian mixture components to assume, $\lambda_{energy}$ is the weight of the sample energy term in the loss function, and $\lambda_{cov}$ is the weight of the regularization term on the covariance to avoid the singularity problem. Additionally, we set the hyperparameters that we thought were important in DAGMM as follows, but got nearly the same performance from all combinations.

**DeepSVDD** We implemented DeepSVDD ourselves using A as a reference. First, we pretrain using Autoencoder for 100 epochs and define the average value of the latent vector obtained by encoding the train data using the corresponding encoder part as the center. Then, we implemented DeepSVDD by training the encoder for 200 epochs with the loss function of the MSE (Mean Squared Error) of the center. The encoder consists of 3 layers with the same hidden size, the batch size is 512, the learning rate is 1e-3, the optimizer is AdamW (Loshchilov & Hutter, 2019), activation function to ReLU, the weight decay is 1e-4, and the learning rate scheduler is CosineAnnealingWarmRestarts (Loshchilov & Hutter, 2017). We set learning rate and latent dimension as the hyperparameters to search for (latent dimension and hidden size are the same).

**GOAD** We ran GOAD experiments based on the model implemented in `https://github.com/lironber/GOAD`. We set the batch size to 512, the epoch to 200. We chose $n_{rot}$, $d_{latent}$, and $d_{out}$ as hyperparameters to explore. $n_{rot}$ is a hyperparameter for how many transformations to perform, and $d_{latent}$ and $d_{out}$ represent the latent dimension and the dimension after transforming the input data, respectively.

**NeuTraLAD** We ran NeuTraLAD experiments based on the model implemented in `https://github.com/boschresearch/NeuTraL-AD`. We set the batch size to 512, the epoch to 200, number of transformation to 11. The hidden and latent dimensions were selected using the automatic dimension selection implemented in the implementation code. We chose $n_{enclayer}$, $n_{translayer}$, and trans_methods hyperparameters to explore. $n_{enclayer}$ represents the number of encoder layers, and $n_{translayer}$ represents the number of layers in the block that performs the transformation. Finally, trans_method is a hyperparameter for how we want to perform the transformation.

**ICL** We ran ICL experiments based on author's official supplementary at `https://openreview.net/forum?id=_hszZbt46bT`. We set the batch size 512. We chose $\tau$ and $d_{latent}$ as hyperparameters to explore. Since earlystopping is implemented in the implementation code, epoch used 2000 as it was specified in the implementation code. $\tau$ represents the softmax temperature, $d_{latent}$ represents the latent dimension.

**MCM** We ran MCM experiments based on author's official supplementary at `https://openreview.net/forum?id=lNZJyEDxy4`. According to the paper, the authors tuned only $\lambda$ and learning rate per dataset, so we set these two hyperparameters to explore. $\lambda$ represent the weight of mask diversity loss.

**NF-SLT** We implemented NICE based on `https://github.com/DakshIdnani/pytorch-nice`. The coupling layer block was set to 10, the epoch was set to 200, and the weight decay was set to 1e-4. The optimizer was AdamW, and the learning rate scheduler was CosineAnnealingWarmRestarts. The batch size was set to 512. The prior distribution of the latent vector was set to $\mathcal{N}(0, I_d)$, and then trained to minimize the negative log-likelihood over the training latent vector. Additionally, due to the implementation structure of the coupling layer, it cannot be implemented when an odd-dimensional vector is input, which was solved by simply adding a single zero padding. We chose learning rate, $d_{latent}$, and $n_{layer}$ as hyperparameters to explore. $d_{latent}$ represents the latent dimension of the coupling layer block, and $n_{layer}$ represents the number of layers in the coupling layer block.

In addition, we recorded the difference in metrics between Table 15 and Table 1 to measure the hyperparameter sensitivity of NF-SLT in Table 12. The reason for choosing this methodology is to compare the overall sensitivity since each deep model has a different hyperparameter searching space defined in Table 10 making it difficult to compare the sensitivity to specific common hyperparameters. The AUROC difference will be large if the model is sensitive to hyperparameters, and the average rank and fail ratio will show negative values. The results in Table 12 show that even if the optimal hyperparameter combination reported in Table 11 is selected, the AUROC of NF-SLT does not increase significantly compared to other models, and the rank index and fail ratio index, which

indicate the relative performance between models, actually worsen when the optimal hyperparameters are selected.

Table 10: Hyperparameter search space each models. $\eta$ is learning rate.

| Model | Hyperparameter Searching Space |
|---|---|
| PCA | n_component_ratio $\in \{0.75, 0.8, 0.85, 0.9, 0.95\}$ |
| LOF | $N \in \{1, 3, 5, 10, 20\}$ |
| IF | $N \in \{30, 50, 100, 200, 400\}$ |
| OCSVM | kernel $\in \{$"linear", "rbf"$\}$ |
| COPOD | None |
| ECOD | None |
| DAGMM | $n_{gmm} \in \{3,4,5\}$
$\lambda_{energy} \in \{0.1, 0.5, 1\}$
$\lambda_{cov} \in \{5e\text{-}3, 1e\text{-}2, 5e\text{-}2\}$ |
| DeepSVDD | $\eta \in \{1e\text{-}4, 5e\text{-}4, 1e\text{-}3\}$
$d_{latent} \in \{64, 128, 256\}$ |
| GOAD | $n_{rot} \in \{16, 32, 64\}$
$d_{latent} \in \{8, 16, 32\}$
$d_{out} \in \{4, 8, 16\}$ |
| NeuTraLAD | $n_{enclayer} \in \{5, 6, 7\}$
$n_{translayer} \in \{2, 3, 4\}$
trans_method $\in \{$'mul', 'residual'$\}$ |
| ICL | $\tau \in \{1e\text{-}2, 1e\text{-}1, 1\}$
$d_{latent} \in \{100, 200, 300\}$ |
| MCM | $\lambda \in \{1e\text{-}2, 1e\text{-}1, 1, 10\}$
$\eta \in \{1e\text{-}4, 5e\text{-}4, 1e\text{-}3, 5e\text{-}3, 1e\text{-}2\}$ |
| NF-SLT | $\eta \in \{1e\text{-}4, 5e\text{-}3, 1e\text{-}3\}$
$d_{latent} \in \{64, 128, 256\}$
$n_{layer} \in \{2, 3, 4\}$ |

Table 11: Optimal hyperparameter combination for each model

| Model | Optimal Hyperparameter |
|---|---|
| PCA | n_component_ratio = 0.8 |
| LOF | $N = 20$ |
| IF | $N = 400$ |
| OCSVM | kernel = "linear" |
| COPOD | None |
| ECOD | None |
| DAGMM | $n_{gmm} = 3$
$\lambda_{energy} = 0.1$
$\lambda_{cov} = 5e\text{-}3$ |
| DeepSVDD | $\eta = 1e\text{-}4$
$d_{latent} = 256$ |
| GOAD | $n_{rot} = 32$
$d_{latent} = 32$
$d_{out} = 4$ |
| NeuTraLAD | $n_{enclayer} = 5$
$n_{translayer} = 3$
trans_method = 'mul' |
| ICL | $\tau = 1e\text{-}1$
$d_{latent} = 300$ |
| MCM | $\lambda = 0.1$
$\eta = 5e\text{-}3$ |
| NF-SLT | $\eta = 1e\text{-}3$
$d_{latent} = 256$
$n_{layer} = 2$ |

Table 12: Hyperparameter sensitivity of deep models. The columns that represent differences represent the differences between the indicators in Table 15 and Table 1. If it is sensitive to changes in hyperparameters, there will be a large difference in performance when experiments are performed with the optimal hyperparameter combination in the hyperparameter searching space for each dataset, compared to when experiments are performed with the same hyperparameter combination for all datasets.

| Method | AUROC Diff | Avg Rank Diff | Fail Ratio Diff |
|--------|-----------|---------------|-----------------|
| DeepSVDD | 0.0366 | -0.96 | -0.19 |
| GOAD | 0.1124 | -0.66 | -0.03 |
| NeuTraL AD | 0.0310 | -0.51 | -0.03 |
| ICL | 0.0284 | -0.45 | -0.12 |
| MCM | 0.0302 | -0.89 | -0.06 |
| NF-SLT | **0.0116** | **0.10** | **0.07** |

## G  PERFORMANCE COMPARISON NF-SLT BETWEEN NICE AND REALNVP

In this section, we set the flow model used for NF-SLT to RealNVP instead of NICE, and then compared the performance with NICE. RealNVP's implementation method replaced by the additive coupling layer used in NICE with an affine coupling layer. We set the hyperparameters of RealNVP to be the same as those of NICE, which recorded the performance in Table 1, except for the learning rate. We set the learning rate to 1e-3, 5e-3, and 1e-4, and selected 5e-3, the learning rate that recorded the highest AUROC performance, to record the performance in Table 13. Table 13 shows that RealNVP underperformed on all metrics compared to NICE. From this, we can conclude that affine coupling layers, which have higher expressive power than additive coupling layers, are not effective in tabular anomaly detection. In our work, we did not use the Glow architecture as a comparison group for the following reasons. The main elements of the Glow architecture are Actnorm and permutation through 1x1 convolution operations. However, batch normalization was not used in our implementation of NICE and RealNVP, and the main idea of Actnorm is to overcome the case where a large batch size cannot be used, so there is no reason to use the function of Actnorm. In addition, although permutation through 1x1 convolution operations was not described in the paper, it was observed that the performance was lower than that of RealNVP when the methodology was applied. Since removing these two elements makes it very similar to the architecture of RealNVP, we did not select the Glow Architecture as a comparison group.

Table 13: Performance comparison between NICE and RealNVP

| Model | AUROC ↑ | AUPRC ↑ | Avg. Rank ↓ | Top2 Cum. Ratio ↑ | Fail Ratio ↓ |
|-------|---------|---------|-------------|-------------------|--------------|
| NICE | 0.8575 | 0.6398 | 3.36 | 0.47 | 0.04 |
| RealNVP | 0.8480 | 0.6385 | 3.87 | 0.36 | 0.04 |

Surely, Draxler et al. (2024) showed that a volume-preserving model like NICE is not a universal approximator. However, the biased density estimation of NICE is not the reason for its strong performance (i.e., the architecture itself is not a universal approximator). This is evidenced by the fact that RealNVP generally performs better than most models, although it still performs worse than NICE. This can also be considered as a new research direction in the future, and researchers can determine the future research direction by relating the reason why the performance of NICE slightly decreases compared to RealNVP in the tabular domain despite its higher expressive power to the theoretical background of the flow model.

## H  TYPICALITY TEST PERFORMANCE

Anomaly detection using the typicality test is a methodology that started with if normal data having a small region but located in a typical set with high probability (Nalisnick et al., 2019b). To explain the methodology of the typicality test, detection is performed based on whether a given $\mathbf{x} \in \mathbb{R}^d$ is included in the $(\epsilon, N)$-typical set $\mathcal{A}_\epsilon^N[p(\mathbf{x})]$ ($p(\mathbf{x})$ is the generative model trained with

$\mathbf{x}_{train} \in \mathcal{D}_{train}$). Therefore, the method for calculating anomaly score $s$ is as shown in Equation 35 (Zhang et al., 2021).

$$s = \left| -\log p(\mathbf{x}) - \hat{H}_p \right| \ s.t \ \hat{H}_p = -\frac{1}{|\mathcal{D}_{train}|} \Sigma_{\mathbf{x} \in \mathcal{D}_{train}} \log p(\mathbf{x}) \qquad (35)$$

To demonstrate the superiority of the simple likelihood test in the tabular domain, where counter-intuitive phenomenon rarely occur, we trained the normalizing flow model using the final selected hyperparameter combination in NF-SLT in Table 1. We recorded the results of the typicality test using Equation 35 instead of the simple likelihood test in Table 14.

Table 14: Performance comparison between SLT and the typicality test

| Model | AUROC ↑ | AUPRC ↑ | Avg. Rank ↓ | Top2 Cum. Ratio ↑ | Fail Ratio ↓ |
|---|---|---|---|---|---|
| SLT | 0.8575 | 0.6398 | 3.36 | 0.47 | 0.04 |
| Typicality Test | 0.8184 | 0.6270 | 5.11 | 0.26 | 0.15 |

According to the results in Table 14, the typicality test confirms that all performance indicators are lower than those of the simple likelihood tests. In almost all datasets, there were little changes in the performance or the performance deteriorated significantly; however, in the "yeast" dataset, which was a dataset where the simple likelihood test failed to detect, the AUROC improved by as much as 8%. Indeed, this is not a dataset where a counterintuitive phenomenon occurred; however, we confirmed that there are cases where the typicality test works much significant than the simple likelihood.

## I    ADDITIONAL EXPERIMENT RESULT

In this section, we reported the additional experimental results. Table 15 records the AUROC and AUPRC performance per-dataset optimal hyperparameter selection. Table 16 records the AUROC for the shallow model and Table 17 records the AUROC for the deep model. Table 18 shows the AUPRC for the shallow model and Table 19 shows the AUPRC for the deep model. Table 20 shows standard deviation of the AUROC for the shallow model and Table 21 shows standard deviation of the AUROC for the deep model.

Table 15: AUROC and AUPRC performance per-dataset optimal hyperparameter selection.

| Method | AUROC ↑ | AUPRC ↑ | Avg. Rank ↓ | Top2 Cum. Ratio ↑ | Fail Ratio ↓ |
|---|---|---|---|---|---|
| PCA | 0.7752 | 0.5240 | 6.94 | 0.13 | 0.38 |
| LOF | 0.8447 | 0.5979 | 5.55 | 0.30 | 0.30 |
| IF | 0.8036 | 0.5099 | 6.09 | 0.17 | 0.17 |
| OCSVM | 0.6651 | 0.3895 | 9.64 | 0.04 | 0.72 |
| COPOD | 0.7471 | 0.4419 | 8.28 | 0.06 | 0.53 |
| ECOD | 0.7425 | 0.4530 | 8.64 | 0.04 | 0.66 |
| DAGMM | 0.6468 | 0.3473 | 11.21 | 0.00 | 0.91 |
| DeepSVDD | 0.8053 | 0.5840 | 6.00 | 0.15 | 0.15 |
| GOAD | 0.7210 | 0.5225 | 9.06 | 0.06 | 0.57 |
| NeuTralAD | 0.8391 | 0.6262 | 5.06 | 0.30 | 0.3 |
| ICL | 0.8492 | 0.6551 | 4.72 | 0.28 | 0.11 |
| MCM | 0.8166 | 0.5988 | 5.81 | 0.21 | 0.17 |
| NF-SLT | **0.8691** | **0.6749** | **3.53** | **0.38** | **0.09** |

## J    DESCRIPTION OF LLM USAGE

During the preparation of this manuscript, we used large language models (LLMs) to assist with polishing the writing and surveying related work.

Table 16: AUROC performance of shallow models

| Dataset | PCA | LOF | IF | OCSVM | COPOD | ECOD |
|---|---|---|---|---|---|---|
| ALOI | 0.5494 | 0.7571 | 0.5418 | 0.5038 | 0.5153 | 0.5304 |
| Cardiotocography | 0.8285 | 0.7784 | 0.8150 | 0.7349 | 0.6625 | 0.7856 |
| Hepatitis | 0.8165 | 0.8249 | 0.7753 | 0.5905 | 0.7986 | 0.7473 |
| InternetAds | 0.7012 | 0.8975 | 0.4842 | 0.3755 | 0.6764 | 0.6770 |
| Ionosphere | 0.9059 | 0.9550 | 0.9141 | 0.5439 | 0.7975 | 0.7328 |
| Lymphography | 0.9923 | 0.9843 | 0.9979 | 0.8761 | 0.9986 | 0.9974 |
| PageBlocks | 0.9324 | 0.9667 | 0.9284 | 0.8506 | 0.8768 | 0.9149 |
| Pima | 0.7058 | 0.7083 | 0.7337 | 0.5874 | 0.6564 | 0.5965 |
| SpamBase | 0.8073 | 0.6875 | 0.8406 | 0.7484 | 0.6896 | 0.6578 |
| Stamps | 0.9255 | 0.9063 | 0.9312 | 0.8522 | 0.9290 | 0.8717 |
| WBC | 0.9936 | 0.9737 | 0.9970 | 0.9949 | 0.9959 | 0.9959 |
| WDBC | 0.9955 | 0.9959 | 0.9951 | 0.8539 | 0.9957 | 0.9748 |
| WPBC | 0.4914 | 0.5642 | 0.5136 | 0.5852 | 0.5116 | 0.4679 |
| Waveform | 0.6495 | 0.7558 | 0.7354 | 0.6529 | 0.7312 | 0.6019 |
| Wilt | 0.2959 | 0.9218 | 0.4776 | 0.9567 | 0.3440 | 0.3935 |
| annthyroid | 0.8071 | 0.8794 | 0.9151 | 0.7075 | 0.7756 | 0.7888 |
| backdoor | 0.6440 | 0.6941 | 0.7659 | 0.7588 | 0.7891 | 0.8459 |
| breastw | 0.9877 | 0.9561 | 0.9946 | 0.9921 | 0.9938 | 0.9904 |
| campaign | 0.7678 | 0.6713 | 0.7400 | 0.6838 | 0.7828 | 0.7695 |
| cardio | 0.9649 | 0.9088 | 0.9508 | 0.7003 | 0.9216 | 0.9348 |
| celeba | 0.7997 | 0.5231 | 0.7177 | 0.5923 | 0.7505 | 0.7569 |
| census | 0.7073 | 0.5954 | 0.6289 | 0.7198 | 0.6741 | 0.6596 |
| cover | 0.9595 | 0.9899 | 0.8597 | 0.6924 | 0.8840 | 0.9203 |
| donors | 0.8947 | 0.9888 | 0.9008 | 0.8762 | 0.8151 | 0.8886 |
| fault | 0.5463 | 0.7073 | 0.6593 | 0.4821 | 0.4557 | 0.4693 |
| fraud | 0.9537 | 0.8706 | 0.9502 | 0.9288 | 0.9475 | 0.9496 |
| glass | 0.7099 | 0.8423 | 0.8041 | 0.2892 | 0.7605 | 0.7136 |
| http | 0.9994 | 0.9405 | 0.9928 | 0.7980 | 0.9916 | 0.9786 |
| landsat | 0.4452 | 0.7768 | 0.6139 | 0.4463 | 0.4197 | 0.3670 |
| letter | 0.5388 | 0.8572 | 0.6419 | 0.4499 | 0.5592 | 0.5718 |
| magicgamma | 0.7032 | 0.8355 | 0.7749 | 0.8143 | 0.6814 | 0.6387 |
| mammography | 0.8983 | 0.8508 | 0.8806 | 0.6572 | 0.9056 | 0.9064 |
| mnist | 0.9060 | 0.7707 | 0.8735 | 0.7149 | 0.7763 | 0.7484 |
| musk | 1.0000 | 1.0000 | 0.9817 | 0.1856 | 0.9458 | 0.9555 |
| optdigits | 0.5740 | 0.9601 | 0.8448 | 0.7722 | 0.6819 | 0.6037 |
| pendigits | 0.9500 | 0.9968 | 0.9706 | 0.6706 | 0.9051 | 0.9274 |
| satellite | 0.6761 | 0.8614 | 0.8060 | 0.8220 | 0.6339 | 0.5836 |
| satimage-2 | 0.9783 | 0.9958 | 0.9935 | 0.5883 | 0.9745 | 0.9649 |
| shuttle | 0.9932 | 0.9979 | 0.9967 | 0.9869 | 0.9945 | 0.9929 |
| skin | 0.6027 | 0.9384 | 0.8911 | 0.4353 | 0.4702 | 0.4881 |
| smtp | 0.8659 | 0.8541 | 0.9126 | 0.2851 | 0.9117 | 0.8801 |
| speech | 0.4687 | 0.8048 | 0.4689 | 0.5275 | 0.4901 | 0.4687 |
| thyroid | 0.9811 | 0.9344 | 0.9898 | 0.6363 | 0.9402 | 0.9777 |
| vertebral | 0.4698 | 0.5921 | 0.4401 | 0.3849 | 0.3515 | 0.4337 |
| vowels | 0.6781 | 0.9649 | 0.7857 | 0.3771 | 0.4955 | 0.5919 |
| wine | 0.9357 | 0.9620 | 0.9175 | 0.8058 | 0.8725 | 0.7415 |
| yeast | 0.4356 | 0.5004 | 0.4259 | 0.3535 | 0.3828 | 0.4447 |
| Avg AUROC | 0.7752 | 0.8447 | 0.8036 | 0.6562 | 0.7471 | 0.7425 |
| Avg.Rank | 6.51 | 5.53 | 5.62 | 9.47 | 7.68 | 7.98 |
| Top1 Ratio | 0.09 | 0.09 | 0.11 | 0.04 | 0.02 | 0.02 |
| Top2 Ratio | 0.09 | 0.13 | 0.09 | 0.02 | 0.09 | 0.06 |
| Top1,2 Cum Ratio | 0.17 | 0.21 | 0.19 | 0.06 | 0.11 | 0.09 |
| Fail Ratio | 0.40 | 0.23 | 0.13 | 0.72 | 0.49 | 0.49 |

Table 17: AUROC performance of deep models

| Dataset | DAGMM | DeepSVDD | GOAD | NeuTralAD | ICL | MCM | NF-SLT |
|---|---|---|---|---|---|---|---|
| ALOI | 0.5024 | 0.5653 | 0.5319 | 0.5700 | 0.5892 | 0.4831 | 0.5479 |
| Cardiotocography | 0.6067 | 0.6759 | 0.3689 | 0.7626 | 0.6221 | 0.5713 | 0.7558 |
| Hepatitis | 0.6167 | 0.6844 | 0.6054 | 0.6516 | 0.6070 | 0.7985 | 0.7475 |
| InternetAds | 0.5590 | 0.6013 | 0.7356 | 0.7112 | 0.8815 | 0.7973 | 0.8746 |
| Ionosphere | 0.6580 | 0.9597 | 0.9459 | 0.9708 | 0.9556 | 0.9574 | 0.9581 |
| Lymphography | 0.8643 | 0.9756 | 0.9847 | 0.9779 | 0.9624 | 0.9934 | 0.9746 |
| PageBlocks | 0.7966 | 0.8866 | 0.9345 | 0.9803 | 0.9691 | 0.8828 | 0.9656 |
| Pima | 0.5759 | 0.5899 | 0.4432 | 0.5414 | 0.6547 | 0.6891 | 0.7219 |
| SpamBase | 0.5564 | 0.5074 | 0.3366 | 0.6406 | 0.8037 | 0.7379 | 0.7724 |
| Stamps | 0.7417 | 0.8361 | 0.6627 | 0.8836 | 0.8122 | 0.8720 | 0.9327 |
| WBC | 0.8804 | 0.9873 | 0.5576 | 0.8944 | 0.9696 | 0.9254 | 0.9726 |
| WDBC | 0.8121 | 0.9932 | 0.7814 | 0.9970 | 0.9856 | 0.9680 | 0.9785 |
| WPBC | 0.4724 | 0.4579 | 0.3565 | 0.4216 | 0.4328 | 0.5068 | 0.5051 |
| Waveform | 0.5135 | 0.6609 | 0.4180 | 0.8186 | 0.6172 | 0.6641 | 0.7357 |
| Wilt | 0.4790 | 0.6008 | 0.7298 | 0.7527 | 0.8043 | 0.8655 | 0.9066 |
| annthyroid | 0.8361 | 0.7689 | 0.5206 | 0.6685 | 0.8732 | 0.8640 | 0.9181 |
| backdoor | 0.4535 | 0.9473 | 0.2805 | 0.9456 | 0.9767 | 0.9222 | 0.9343 |
| breastw | 0.8740 | 0.9825 | 0.8402 | 0.9505 | 0.9835 | 0.9900 | 0.9842 |
| campaign | 0.5948 | 0.7929 | 0.2837 | 0.7814 | 0.6933 | 0.7987 | 0.8059 |
| cardio | 0.6091 | 0.9139 | 0.5964 | 0.7367 | 0.8456 | 0.8093 | 0.9174 |
| celeba | 0.5308 | 0.3604 | 0.5204 | 0.6888 | 0.4595 | 0.6400 | 0.7340 |
| census | 0.4619 | 0.6421 | 0.4140 | 0.4991 | 0.3917 | 0.7478 | 0.7186 |
| cover | 0.7412 | 0.6819 | 0.2571 | 0.9096 | 0.9859 | 0.8949 | 0.9658 |
| donors | 0.6243 | 0.9970 | 0.4472 | 0.9985 | 0.9630 | 0.9987 | 0.9990 |
| fault | 0.5491 | 0.7146 | 0.7028 | 0.7680 | 0.7991 | 0.5955 | 0.7518 |
| fraud | 0.8141 | 0.9517 | 0.4119 | 0.9448 | 0.9524 | 0.8488 | 0.9564 |
| glass | 0.5561 | 0.4046 | 0.4495 | 0.8305 | 0.9027 | 0.8783 | 0.8867 |
| http | 0.9966 | 0.9996 | 0.7812 | 0.9380 | 0.9999 | 0.9925 | 1.0000 |
| landsat | 0.4874 | 0.6759 | 0.6427 | 0.7582 | 0.7075 | 0.5724 | 0.6543 |
| letter | 0.5177 | 0.7565 | 0.6528 | 0.8227 | 0.9312 | 0.3674 | 0.9258 |
| magicgamma | 0.6099 | 0.7267 | 0.5677 | 0.7317 | 0.7453 | 0.8143 | 0.8863 |
| mammography | 0.5903 | 0.7263 | 0.8477 | 0.6377 | 0.8108 | 0.8321 | 0.8761 |
| mnist | 0.6168 | 0.5940 | 0.7840 | 0.9700 | 0.8931 | 0.8631 | 0.9015 |
| musk | 0.7128 | 0.9868 | 0.9999 | 1.0000 | 1.0000 | 0.9972 | 1.0000 |
| optdigits | 0.4249 | 0.7386 | 0.2742 | 0.8566 | 0.8192 | 0.9279 | 0.9205 |
| pendigits | 0.6830 | 0.9510 | 0.2513 | 0.9701 | 0.9619 | 0.9913 | 0.9930 |
| satellite | 0.6913 | 0.8275 | 0.7789 | 0.8186 | 0.8190 | 0.8008 | 0.8276 |
| satimage-2 | 0.9234 | 0.9971 | 0.9960 | 0.9987 | 0.9974 | 0.9913 | 0.9966 |
| shuttle | 0.6404 | 0.9980 | 0.3801 | 0.9994 | 0.9998 | 0.9979 | 0.9984 |
| skin | 0.7063 | 0.7270 | 0.8856 | 0.9128 | 0.8954 | 0.8175 | 0.9675 |
| smtp | 0.7976 | 0.8855 | 0.6588 | 0.8583 | 0.8805 | 0.9268 | 0.9201 |
| speech | 0.5008 | 0.4688 | 0.5138 | 0.5123 | 0.5888 | 0.4110 | 0.5795 |
| thyroid | 0.9408 | 0.9256 | 0.4299 | 0.9619 | 0.9653 | 0.9584 | 0.9840 |
| vertebral | 0.4778 | 0.4794 | 0.5692 | 0.6381 | 0.5621 | 0.3544 | 0.5483 |
| vowels | 0.5295 | 0.9377 | 0.9316 | 0.7861 | 0.9914 | 0.6873 | 0.9852 |
| wine | 0.7212 | 0.9533 | 0.9430 | 0.9680 | 0.9575 | 0.4924 | 0.9497 |
| yeast | 0.5442 | 0.6315 | 0.5979 | 0.5438 | 0.5574 | 0.4654 | 0.4652 |
| Avg AUROC | 0.6467 | 0.7687 | 0.6086 | 0.8081 | 0.8208 | 0.7864 | 0.8575 |
| Avg.Rank | 10.51 | 6.96 | 9.72 | 5.57 | 5.17 | 6.70 | 3.43 |
| Top1 Ratio | 0.00 | 0.02 | 0.00 | 0.21 | 0.19 | 0.06 | 0.21 |
| Top2 Ratio | 0.00 | 0.04 | 0.04 | 0.04 | 0.13 | 0.04 | 0.23 |
| Top1,2 Cum. Ratio | 0.00 | 0.06 | 0.04 | 0.26 | 0.32 | 0.11 | 0.45 |
| Fail Ratio | 0.85 | 0.34 | 0.60 | 0.26 | 0.23 | 0.23 | 0.02 |

Table 18: AUPRC performance of shallow models

| Dataset | PCA | LOF | IF | OCSVM | COPOD | ECOD |
|---|---|---|---|---|---|---|
| ALOI | 0.0711 | 0.1334 | 0.0649 | 0.0592 | 0.0607 | 0.0637 |
| Cardiotocography | 0.7324 | 0.6321 | 0.7085 | 0.5931 | 0.5516 | 0.6598 |
| Hepatitis | 0.6398 | 0.6612 | 0.5448 | 0.4062 | 0.5783 | 0.4707 |
| InternetAds | 0.5689 | 0.7937 | 0.2782 | 0.2448 | 0.6206 | 0.6216 |
| Ionosphere | 0.9223 | 0.9609 | 0.9252 | 0.7024 | 0.7988 | 0.7743 |
| Lymphography | 0.8923 | 0.8121 | 0.9741 | 0.6735 | 0.9830 | 0.9717 |
| PageBlocks | 0.7340 | 0.8497 | 0.6965 | 0.7058 | 0.5254 | 0.6602 |
| Pima | 0.6961 | 0.6776 | 0.7268 | 0.6157 | 0.6852 | 0.6323 |
| SpamBase | 0.8427 | 0.7184 | 0.8696 | 0.8003 | 0.7039 | 0.6819 |
| Stamps | 0.5743 | 0.5235 | 0.5807 | 0.4142 | 0.5632 | 0.4716 |
| WBC | 0.9427 | 0.7288 | 0.9719 | 0.9510 | 0.9556 | 0.9556 |
| WDBC | 0.9110 | 0.8804 | 0.9002 | 0.4891 | 0.9283 | 0.6830 |
| WPBC | 0.3681 | 0.4122 | 0.3839 | 0.4560 | 0.3742 | 0.3492 |
| Waveform | 0.0861 | 0.2901 | 0.1132 | 0.1099 | 0.1047 | 0.0771 |
| Wilt | 0.0665 | 0.4182 | 0.0876 | 0.5389 | 0.0708 | 0.0799 |
| annthyroid | 0.5127 | 0.5591 | 0.6231 | 0.3465 | 0.2944 | 0.4058 |
| backdoor | 0.0783 | 0.0989 | 0.0942 | 0.1223 | 0.1275 | 0.1677 |
| breastw | 0.9832 | 0.9181 | 0.9944 | 0.9919 | 0.9936 | 0.9906 |
| campaign | 0.4878 | 0.3429 | 0.4645 | 0.3628 | 0.5141 | 0.4994 |
| cardio | 0.8454 | 0.6653 | 0.7977 | 0.3625 | 0.7103 | 0.7070 |
| celeba | 0.2083 | 0.0358 | 0.1306 | 0.0524 | 0.1656 | 0.1697 |
| census | 0.2012 | 0.1174 | 0.1403 | 0.2093 | 0.1610 | 0.1547 |
| cover | 0.1925 | 0.8391 | 0.0827 | 0.0534 | 0.1214 | 0.1865 |
| donors | 0.3618 | 0.7986 | 0.4125 | 0.3834 | 0.3353 | 0.4142 |
| fault | 0.5726 | 0.6218 | 0.6542 | 0.5049 | 0.4751 | 0.4906 |
| fraud | 0.2661 | 0.0287 | 0.2209 | 0.3929 | 0.3669 | 0.3339 |
| glass | 0.1791 | 0.2206 | 0.2177 | 0.0633 | 0.2001 | 0.2535 |
| http | 0.9156 | 0.1117 | 0.5033 | 0.4461 | 0.4636 | 0.2534 |
| landsat | 0.3169 | 0.6913 | 0.4300 | 0.3179 | 0.2977 | 0.2799 |
| letter | 0.1466 | 0.4482 | 0.1661 | 0.1108 | 0.1284 | 0.1432 |
| magicgamma | 0.7480 | 0.8612 | 0.8051 | 0.8429 | 0.7225 | 0.6807 |
| mammography | 0.4513 | 0.3563 | 0.3854 | 0.0753 | 0.5459 | 0.5516 |
| mnist | 0.6753 | 0.3479 | 0.5455 | 0.3116 | 0.3560 | 0.3045 |
| musk | 1.0000 | 0.9995 | 0.7037 | 0.2154 | 0.4869 | 0.6140 |
| optdigits | 0.0591 | 0.3212 | 0.1714 | 0.1326 | 0.0807 | 0.0650 |
| pendigits | 0.4162 | 0.8520 | 0.5394 | 0.0763 | 0.2930 | 0.3924 |
| satellite | 0.7711 | 0.8815 | 0.8455 | 0.8505 | 0.6957 | 0.6596 |
| satimage-2 | 0.8823 | 0.9305 | 0.9371 | 0.1132 | 0.8349 | 0.7419 |
| shuttle | 0.9615 | 0.9784 | 0.9863 | 0.9732 | 0.9773 | 0.9474 |
| skin | 0.3585 | 0.7164 | 0.6368 | 0.3285 | 0.2958 | 0.3030 |
| smtp | 0.4636 | 0.4138 | 0.0088 | 0.0007 | 0.0081 | 0.5760 |
| speech | 0.0361 | 0.0440 | 0.0389 | 0.0404 | 0.0377 | 0.0385 |
| thyroid | 0.7791 | 0.6215 | 0.8265 | 0.3331 | 0.3077 | 0.6362 |
| vertebral | 0.1922 | 0.2003 | 0.1971 | 0.1892 | 0.1680 | 0.2015 |
| vowels | 0.1949 | 0.6011 | 0.2576 | 0.0839 | 0.0660 | 0.1421 |
| wine | 0.7044 | 0.7307 | 0.6605 | 0.5416 | 0.5629 | 0.3364 |
| yeast | 0.4707 | 0.5043 | 0.4787 | 0.4277 | 0.4696 | 0.4998 |
| Avg AUPRC | 0.5209 | 0.5606 | 0.5060 | 0.3833 | 0.4419 | 0.4530 |
| Avg.Rank | 6.81 | 5.66 | 6.02 | 9.23 | 8.26 | 8.19 |
| Top1 Ratio | 0.06 | 0.17 | 0.13 | 0.04 | 0.02 | 0.02 |
| Top2 Ratio | 0.06 | 0.06 | 0.04 | 0.02 | 0.09 | 0.06 |
| Top1,2 Cum Ratio | 0.13 | 0.23 | 0.17 | 0.06 | 0.11 | 0.09 |
| Fail Ratio | 0.36 | 0.19 | 0.21 | 0.66 | 0.53 | 0.51 |

Table 19: AUPRC performance of deep models

| Dataset | DAGMM | DeepSVDD | GOAD | NeuTralAD | ICL | MCM | NF-SLT |
|---|---|---|---|---|---|---|---|
| ALOI | 0.0589 | 0.1186 | 0.0749 | 0.0937 | 0.1058 | 0.0578 | 0.0831 |
| Cardiotocography | 0.4520 | 0.6307 | 0.3829 | 0.6679 | 0.5864 | 0.5245 | 0.6742 |
| Hepatitis | 0.4622 | 0.5382 | 0.4831 | 0.3848 | 0.3773 | 0.5708 | 0.5310 |
| InternetAds | 0.3792 | 0.4077 | 0.5967 | 0.4142 | 0.8345 | 0.7551 | 0.8399 |
| Ionosphere | 0.7007 | 0.9702 | 0.9615 | 0.9761 | 0.9673 | 0.9705 | 0.9676 |
| Lymphography | 0.5683 | 0.7607 | 0.8225 | 0.7839 | 0.7166 | 0.9187 | 0.7691 |
| PageBlocks | 0.5661 | 0.7479 | 0.8474 | 0.9072 | 0.8890 | 0.6263 | 0.8644 |
| Pima | 0.5831 | 0.6349 | 0.5237 | 0.5611 | 0.6805 | 0.6922 | 0.7117 |
| SpamBase | 0.6143 | 0.6307 | 0.5289 | 0.6962 | 0.8289 | 0.7449 | 0.8063 |
| Stamps | 0.3846 | 0.4953 | 0.3601 | 0.5896 | 0.4683 | 0.4703 | 0.6273 |
| WBC | 0.6306 | 0.8947 | 0.1882 | 0.5681 | 0.7894 | 0.7778 | 0.7923 |
| WDBC | 0.3081 | 0.8617 | 0.4889 | 0.9632 | 0.8215 | 0.7424 | 0.6770 |
| WPBC | 0.3795 | 0.3655 | 0.3177 | 0.3349 | 0.3449 | 0.4092 | 0.3727 |
| Waveform | 0.0607 | 0.1041 | 0.0460 | 0.2014 | 0.1301 | 0.1196 | 0.1743 |
| Wilt | 0.1019 | 0.1150 | 0.2689 | 0.2147 | 0.3039 | 0.3129 | 0.3903 |
| annthyroid | 0.5478 | 0.4005 | 0.3897 | 0.3503 | 0.5520 | 0.5679 | 0.6227 |
| backdoor | 0.0461 | 0.6203 | 0.0527 | 0.9027 | 0.8968 | 0.3615 | 0.5688 |
| breastw | 0.8930 | 0.9821 | 0.8605 | 0.9442 | 0.9790 | 0.9887 | 0.9797 |
| campaign | 0.2771 | 0.5130 | 0.1368 | 0.5227 | 0.4218 | 0.5070 | 0.5213 |
| cardio | 0.2816 | 0.7979 | 0.4940 | 0.4121 | 0.6640 | 0.6778 | 0.7557 |
| celeba | 0.0548 | 0.0317 | 0.0426 | 0.0887 | 0.0400 | 0.0663 | 0.0955 |
| census | 0.1036 | 0.1589 | 0.0909 | 0.1163 | 0.0898 | 0.2249 | 0.2084 |
| cover | 0.0998 | 0.0483 | 0.0120 | 0.4044 | 0.7338 | 0.4159 | 0.7204 |
| donors | 0.1729 | 0.9754 | 0.0994 | 0.9853 | 0.7310 | 0.9783 | 0.9879 |
| fault | 0.5743 | 0.7123 | 0.7061 | 0.7222 | 0.7884 | 0.6307 | 0.7412 |
| fraud | 0.0515 | 0.7615 | 0.1315 | 0.6194 | 0.6445 | 0.6662 | 0.7082 |
| glass | 0.1362 | 0.1954 | 0.1929 | 0.2957 | 0.4677 | 0.4202 | 0.3655 |
| http | 0.7535 | 0.9956 | 0.3368 | 0.8768 | 0.9894 | 0.5124 | 0.9917 |
| landsat | 0.3407 | 0.5269 | 0.3988 | 0.5465 | 0.6609 | 0.3967 | 0.4795 |
| letter | 0.1360 | 0.2934 | 0.2592 | 0.3964 | 0.6395 | 0.0893 | 0.6673 |
| magicgamma | 0.6468 | 0.7545 | 0.6289 | 0.7665 | 0.8042 | 0.8549 | 0.9094 |
| mammography | 0.0888 | 0.0964 | 0.3818 | 0.1382 | 0.3841 | 0.3291 | 0.3885 |
| mnist | 0.2500 | 0.2424 | 0.5855 | 0.8695 | 0.6798 | 0.5856 | 0.7080 |
| musk | 0.3674 | 0.7550 | 0.9980 | 1.0000 | 1.0000 | 0.9918 | 1.0000 |
| optdigits | 0.0485 | 0.1621 | 0.0360 | 0.2009 | 0.1563 | 0.2755 | 0.2725 |
| pendigits | 0.1507 | 0.3583 | 0.0283 | 0.5856 | 0.6152 | 0.8148 | 0.8073 |
| satellite | 0.7369 | 0.8576 | 0.7973 | 0.8667 | 0.8632 | 0.8446 | 0.8658 |
| satimage-2 | 0.4356 | 0.9706 | 0.9641 | 0.9735 | 0.9586 | 0.8117 | 0.8984 |
| shuttle | 0.2613 | 0.9818 | 0.2937 | 0.9944 | 0.9972 | 0.9608 | 0.9695 |
| skin | 0.5066 | 0.4875 | 0.6741 | 0.7835 | 0.6725 | 0.5557 | 0.9226 |
| smtp | 0.1050 | 0.6110 | 0.2985 | 0.4615 | 0.5455 | 0.3936 | 0.4675 |
| speech | 0.0409 | 0.0312 | 0.0395 | 0.0362 | 0.0507 | 0.0316 | 0.0529 |
| thyroid | 0.6545 | 0.6129 | 0.2536 | 0.6164 | 0.5694 | 0.6664 | 0.7108 |
| vertebral | 0.2437 | 0.2224 | 0.2879 | 0.3645 | 0.3090 | 0.1731 | 0.2617 |
| vowels | 0.0798 | 0.5534 | 0.6197 | 0.1764 | 0.9254 | 0.1425 | 0.8755 |
| wine | 0.4028 | 0.7368 | 0.7695 | 0.8384 | 0.7650 | 0.1736 | 0.7601 |
| yeast | 0.5627 | 0.5992 | 0.5863 | 0.5513 | 0.5573 | 0.4973 | 0.5052 |
| Avg AUPRC | 0.3468 | 0.5388 | 0.4114 | 0.5694 | 0.6170 | 0.5383 | 0.6398 |
| Avg.Rank | 10.36 | 6.53 | 9.09 | 5.34 | 5.23 | 6.53 | 3.68 |
| Top1 Ratio | 0.00 | 0.09 | 0.00 | 0.21 | 0.09 | 0.02 | 0.17 |
| Top2 Ratio | 0.00 | 0.06 | 0.04 | 0.13 | 0.17 | 0.09 | 0.17 |
| Top1,2 Cum Ratio | 0.00 | 0.15 | 0.04 | 0.34 | 0.26 | 0.11 | 0.34 |
| Fail Ratio | 0.81 | 0.32 | 0.57 | 0.26 | 0.23 | 0.30 | 0.04 |

Table 20: Standard Deviation of the AUROC for Shallow Models

| Dataset | PCA | LOF | IF | OCSVM | COPOD | ECOD |
|---|---|---|---|---|---|---|
| ALOI | 0.0012 | 0.0031 | 0.0025 | 0.0007 | 0.0009 | 0.0008 |
| Cardiotocography | 0.0055 | 0.0208 | 0.0114 | 0.0106 | 0.0046 | 0.0051 |
| Hepatitis | 0.0137 | 0.0284 | 0.0341 | 0.0431 | 0.0275 | 0.0373 |
| InternetAds | 0.0334 | 0.0070 | 0.0526 | 0.0094 | 0.0045 | 0.0044 |
| Ionosphere | 0.0153 | 0.0106 | 0.0162 | 0.0667 | 0.0217 | 0.0161 |
| Lymphography | 0.0061 | 0.0072 | 0.0036 | 0.1011 | 0.0028 | 0.0031 |
| PageBlocks | 0.0032 | 0.0037 | 0.0030 | 0.0346 | 0.0032 | 0.0026 |
| Pima | 0.0143 | 0.0119 | 0.0147 | 0.0381 | 0.0151 | 0.0163 |
| SpamBase | 0.0079 | 0.0183 | 0.0063 | 0.0126 | 0.0062 | 0.0060 |
| Stamps | 0.0251 | 0.0237 | 0.0210 | 0.0500 | 0.0159 | 0.0175 |
| WBC | 0.0034 | 0.0074 | 0.0015 | 0.0014 | 0.0023 | 0.0023 |
| WDBC | 0.0038 | 0.0036 | 0.0035 | 0.1023 | 0.0020 | 0.0042 |
| WPBC | 0.0319 | 0.0283 | 0.0371 | 0.0563 | 0.0332 | 0.0323 |
| Waveform | 0.0043 | 0.0116 | 0.0082 | 0.1012 | 0.0057 | 0.0067 |
| Wilt | 0.0508 | 0.0070 | 0.0154 | 0.0069 | 0.0029 | 0.0033 |
| annthyroid | 0.0192 | 0.0221 | 0.0078 | 0.0752 | 0.0024 | 0.0025 |
| backdoor | 0.0031 | 0.0268 | 0.0115 | 0.0013 | 0.0009 | 0.0009 |
| breastw | 0.0045 | 0.0080 | 0.0016 | 0.0021 | 0.0016 | 0.0019 |
| campaign | 0.0022 | 0.0058 | 0.0105 | 0.0029 | 0.0008 | 0.0008 |
| cardio | 0.0021 | 0.0083 | 0.0063 | 0.0175 | 0.0045 | 0.0041 |
| celeba | 0.0004 | 0.0040 | 0.0103 | 0.0008 | 0.0004 | 0.0005 |
| census | 0.0009 | 0.0025 | 0.0122 | 0.0009 | 0.0007 | 0.0007 |
| cover | 0.0003 | 0.0006 | 0.0107 | 0.0030 | 0.0004 | 0.0002 |
| donors | 0.0088 | 0.0006 | 0.0110 | 0.0182 | 0.0003 | 0.0002 |
| fault | 0.0186 | 0.0243 | 0.0151 | 0.0147 | 0.0059 | 0.0052 |
| fraud | 0.0003 | 0.0230 | 0.0013 | 0.0009 | 0.0001 | 0.0001 |
| glass | 0.0466 | 0.0287 | 0.0263 | 0.1472 | 0.0228 | 0.0248 |
| http | 0.0000 | 0.0060 | 0.0012 | 0.3882 | 0.0002 | 0.0003 |
| landsat | 0.0067 | 0.0056 | 0.0121 | 0.0139 | 0.0061 | 0.0049 |
| letter | 0.0062 | 0.0082 | 0.0148 | 0.0167 | 0.0106 | 0.0096 |
| magicgamma | 0.0019 | 0.0026 | 0.0054 | 0.0015 | 0.0025 | 0.0025 |
| mammography | 0.0046 | 0.0143 | 0.0025 | 0.0134 | 0.0012 | 0.0012 |
| mnist | 0.0027 | 0.0160 | 0.0051 | 0.0109 | 0.0016 | 0.0019 |
| musk | 0.0000 | 0.0000 | 0.0126 | 0.0000 | 0.0026 | 0.0021 |
| optdigits | 0.0089 | 0.0077 | 0.0188 | 0.0124 | 0.0028 | 0.0028 |
| pendigits | 0.0025 | 0.0015 | 0.0037 | 0.0137 | 0.0029 | 0.0023 |
| satellite | 0.0022 | 0.0051 | 0.0082 | 0.0296 | 0.0033 | 0.0026 |
| satimage-2 | 0.0004 | 0.0008 | 0.0007 | 0.1826 | 0.0006 | 0.0009 |
| shuttle | 0.0006 | 0.0005 | 0.0002 | 0.0013 | 0.0001 | 0.0001 |
| skin | 0.0006 | 0.0048 | 0.0029 | 0.2456 | 0.0007 | 0.0008 |
| smtp | 0.0421 | 0.0454 | 0.0019 | 0.0136 | 0.0005 | 0.0003 |
| speech | 0.0042 | 0.0036 | 0.0075 | 0.0368 | 0.0052 | 0.0044 |
| thyroid | 0.0012 | 0.0130 | 0.0015 | 0.2706 | 0.0023 | 0.0012 |
| vertebral | 0.0315 | 0.0250 | 0.0453 | 0.0832 | 0.0189 | 0.0209 |
| vowels | 0.0268 | 0.0039 | 0.0174 | 0.1791 | 0.0068 | 0.0073 |
| wine | 0.0196 | 0.0184 | 0.0336 | 0.1725 | 0.0100 | 0.0095 |
| yeast | 0.0177 | 0.0130 | 0.0125 | 0.0116 | 0.0098 | 0.0102 |

Table 21: Standard Deviation of the AUROC for deep Models

| Dataset | DAGMM | DeepSVDD | GOAD | NeuTralAD | ICL | MCM | NF-SLT |
|---|---|---|---|---|---|---|---|
| ALOI | 0.0182 | 0.0085 | 0.0030 | 0.0096 | 0.0158 | 0.0063 | 0.0017 |
| Cardiotocography | 0.0939 | 0.0109 | 0.0144 | 0.0157 | 0.0314 | 0.0141 | 0.0169 |
| Hepatitis | 0.0835 | 0.0753 | 0.0582 | 0.0815 | 0.0963 | 0.0200 | 0.0787 |
| InternetAds | 0.0430 | 0.0213 | 0.0442 | 0.0179 | 0.0150 | 0.0013 | 0.0114 |
| Ionosphere | 0.0788 | 0.0075 | 0.0164 | 0.0054 | 0.0082 | 0.0011 | 0.0085 |
| Lymphography | 0.1437 | 0.0111 | 0.0087 | 0.0210 | 0.0150 | 0.0020 | 0.0120 |
| PageBlocks | 0.0454 | 0.0105 | 0.0123 | 0.0044 | 0.0049 | 0.0056 | 0.0027 |
| Pima | 0.0464 | 0.0191 | 0.0349 | 0.0311 | 0.0203 | 0.0131 | 0.0163 |
| SpamBase | 0.0646 | 0.0191 | 0.0228 | 0.0157 | 0.0232 | 0.0055 | 0.0200 |
| Stamps | 0.1480 | 0.0776 | 0.1259 | 0.0495 | 0.0875 | 0.0089 | 0.0243 |
| WBC | 0.0968 | 0.0047 | 0.1129 | 0.0460 | 0.0142 | 0.0091 | 0.0118 |
| WDBC | 0.1513 | 0.0031 | 0.1144 | 0.0027 | 0.0095 | 0.0095 | 0.0107 |
| WPBC | 0.0589 | 0.0314 | 0.0347 | 0.0475 | 0.0217 | 0.0144 | 0.0270 |
| Waveform | 0.0560 | 0.0173 | 0.0395 | 0.0061 | 0.0186 | 0.0213 | 0.0224 |
| Wilt | 0.1216 | 0.0185 | 0.0418 | 0.1231 | 0.0649 | 0.0185 | 0.0140 |
| annthyroid | 0.1246 | 0.0202 | 0.0940 | 0.0251 | 0.0332 | 0.0090 | 0.0097 |
| backdoor | 0.1457 | 0.0035 | 0.0149 | 0.0050 | 0.0044 | 0.0070 | 0.0070 |
| breastw | 0.0886 | 0.0049 | 0.1070 | 0.0260 | 0.0041 | 0.0019 | 0.0033 |
| campaign | 0.0401 | 0.0283 | 0.0628 | 0.0134 | 0.0157 | 0.0028 | 0.0024 |
| cardio | 0.1023 | 0.0114 | 0.0385 | 0.0518 | 0.0326 | 0.0123 | 0.0102 |
| celeba | 0.0658 | 0.0786 | 0.0742 | 0.0140 | 0.0869 | 0.0349 | 0.0284 |
| census | 0.0499 | 0.0463 | 0.0455 | 0.0064 | 0.0146 | 0.0034 | 0.0094 |
| cover | 0.1598 | 0.0197 | 0.0697 | 0.1051 | 0.0051 | 0.0187 | 0.0067 |
| donors | 0.1132 | 0.0025 | 0.0824 | 0.0023 | 0.0398 | 0.0004 | 0.0010 |
| fault | 0.0606 | 0.0136 | 0.0299 | 0.0160 | 0.0104 | 0.0084 | 0.0167 |
| fraud | 0.0656 | 0.0045 | 0.1054 | 0.0069 | 0.0024 | 0.0110 | 0.0011 |
| glass | 0.1455 | 0.0382 | 0.0865 | 0.0594 | 0.0378 | 0.0161 | 0.0532 |
| http | 0.0029 | 0.0002 | 0.3625 | 0.1784 | 0.0001 | 0.0030 | 0.0000 |
| landsat | 0.0855 | 0.0105 | 0.0193 | 0.0085 | 0.0054 | 0.0058 | 0.0127 |
| letter | 0.0548 | 0.0087 | 0.0121 | 0.0143 | 0.0102 | 0.0038 | 0.0083 |
| magicgamma | 0.0627 | 0.0049 | 0.0233 | 0.0148 | 0.0177 | 0.0035 | 0.0027 |
| mammography | 0.1788 | 0.0139 | 0.0181 | 0.0321 | 0.0435 | 0.0059 | 0.0067 |
| mnist | 0.0637 | 0.0095 | 0.0318 | 0.0022 | 0.0103 | 0.0277 | 0.0042 |
| musk | 0.2378 | 0.0046 | 0.0002 | 0.0000 | 0.0000 | 0.0005 | 0.0000 |
| optdigits | 0.1469 | 0.0255 | 0.0434 | 0.0225 | 0.0681 | 0.0091 | 0.0147 |
| pendigits | 0.1647 | 0.0104 | 0.0807 | 0.0124 | 0.0541 | 0.0029 | 0.0034 |
| satellite | 0.0745 | 0.0044 | 0.0185 | 0.0030 | 0.0032 | 0.0022 | 0.0034 |
| satimage-2 | 0.0296 | 0.0004 | 0.0012 | 0.0002 | 0.0008 | 0.0026 | 0.0009 |
| shuttle | 0.2304 | 0.0003 | 0.0570 | 0.0004 | 0.0001 | 0.0005 | 0.0009 |
| skin | 0.1689 | 0.0095 | 0.0143 | 0.0220 | 0.0509 | 0.0215 | 0.0113 |
| smtp | 0.0649 | 0.0039 | 0.0525 | 0.0584 | 0.0185 | 0.0195 | 0.0108 |
| speech | 0.0241 | 0.0221 | 0.0375 | 0.0467 | 0.0387 | 0.0057 | 0.0228 |
| thyroid | 0.0434 | 0.0132 | 0.1317 | 0.0121 | 0.0075 | 0.0038 | 0.0028 |
| vertebral | 0.0745 | 0.0316 | 0.0817 | 0.1008 | 0.0590 | 0.0471 | 0.0484 |
| vowels | 0.0895 | 0.0066 | 0.0149 | 0.0502 | 0.0055 | 0.0510 | 0.0049 |
| wine | 0.2031 | 0.0288 | 0.0493 | 0.0223 | 0.0264 | 0.0693 | 0.0544 |
| yeast | 0.0241 | 0.0219 | 0.0217 | 0.0190 | 0.0179 | 0.0124 | 0.0150 |

