# OpenReview forum: "Why Is the Counterintuitive Phenomenon of Likelihood Rare in Tabular Anomaly Detection with Deep Generative Models?"
_ICLR.cc/2026/Conference — Submitted to ICLR 2026_

### Official Review · Reviewer_Bu6W · 2025-10-16

**Soundness:** 2
**Presentation:** 1
**Contribution:** 2
**Rating:** 2
**Confidence:** 4

**Summary:**

The present work investigates the *counterintuitive phenomenon* of likelihood in the context of structured tabular data. In the weakly supervised anomaly detection (AD) setup, where one disposes of a labeled dataset of normal and anomaly samples, one can train a generative model solely on the normal samples. The counterintuitive phenomenon describes the situation where a generative model, that provides an estimate of the likelihood of a sample given the learned **normal** distribution, e.g., Normalizing Flow, gives on average a higher likelihood to anomaly samples than to normal samples. This phenomenon has been mostly observed in the CV field and has yet to be investigated in the tabular domain.

The authors provide a formal definition of this counterintuitive phenomenon and investigate whether it arises in the tabular domain. They do so by comparing a Likelihood-based anomaly detection method to competing AD methods. They posit that if (i) likelihood as estimated by their model is sometimes higher for anomaly samples than for normal samples, but (ii) the other AD methods display an equivalent behavior with the anomaly score being higher for normal samples than for anomalies, this may be a sign that a different underlying phenomenon might be the cause.

They provide extensive experiments on a widely used AD benchmark, ADBench, and demonstrate that this counterintuitive phenomenon almost never happens when it comes to tabular data. The authors provide empirical and theoretical explanations as to why they observe these results.

**Strengths:**

**S1:** Experiments are rigorous and extensive. Authors rely on a widely used benchmark in the AD literature on tabular data.

**S2:** The results are **fully reproducible** as the authors provide the code to run their experiments.

**S3:** The research question is interesting and is worth investigating. The approach chosen by the authors is relevant and well-motivated.

**Weaknesses:**

**W1:** Overall the paper is **very hard to follow**. The structure is somewhat disorganized, and the formulations occasionally make it difficult to understand the points made by the author. Similarly, the paper contains **many typos**, we invite the authors to more thoroughly proofread their manuscript before submitting it.

**W2**: The authors should **avoid citing preprints (e.g., from arXiv) when the papers have been published**. A few examples found in the reference section with arXiv citations:
- A geometric explanation of the likelihood OOD detection paradox (Kamkari et al. 2024) has been accepted to ICML 2024.
- Density estimation using real nvp. (Dinh et al., 2017) has been accepted to ICLR 2017.
- On the universality of volume-preserving and coupling-based normalizing flows (Draxler et al., 2024) has been accepted to ICML 2024.
- Beyond individual input for deep anomaly detection on tabular data (Thimonier et al., 2024) has been accepted to ICML 2024.

**W3**: The method NF-SLT put forward by the authors is not explained in the main part of the paper.

**Questions:**

**Q1**: As mentioned in **W1** the paper should be **reorganized significantly** as it suffers from (i) unclear turns of phrases, (ii) many typos, (iii) unnecessary formalization. A few non-exhaustive examples:
- lines 54 to 68 are very hard to follow and can be confusing.
- line 114: "it follows a simple distribution $p_{\mathbf{z}}$ that tipically selected standard Gaussian".
- line 116: "$p_{\mathbf{x}}$ can be expressed as a formula expressed in terms of $p_{\mathbf{z}}$. The authors then proceed to use the term 'expressed' twice again in the following sentence.
- lines 141-143: the authors mention performance improvement twice, but never truly explain to what performance they are referring.
- line 148: "overcame disadvantage"
- Section 3: The beginning of Section 3 repeats more or less what is already being said in the introduction.
- Definition 3.3 appears as an overcomplication to formalize a simplistic problem that can be easily explained in plain words (as done in lines 204 to 208). Moreover, I do not think this can be considered a definition.
- The acronym NF-SLT is used on several occurrences throughout the manuscript but is never defined (e.g., in the abstract).
- line 296: 'we extend the expression which expresses the expected'
- On several occasions, the authors mention "the d" to refer to the feature dimension. This is an odd turn of phrase.
- lines 340 to 361 are tough to follow and would benefit from smaller sentences.

**Q2**: As observed in Table 1 (lower part) and mentioned in section 4, it appears that **on the CV and NLP datasets, the *counterintuitive phenomenon* is not observed as NF-SLT outperforms all competing methods**. Since the premise is that this phenomenon is widely observed in the CV field, how do you account for the fact that it disappears when using the embedding representations rather than the original pixel representation? These embeddings do not suffer from the heterogeneity of features that tabular data displays. This should thus push this counterintuitive phenomenon according to section 5.2.

**Q3**: Could you provide **in the main text** the hyperparameter setting and overall training settings? While I acknowledge that some information is given in section 4, I would recommend providing more details (on architecture for example).

I lean towards reject due to the listed weaknesses. While I believe that this work might prove valuable in the future, it still needs significant improvement to enhance its clarity.

---

> ### Author Response · Authors · 2025-11-23
>
> >**Answer of Question 1 & Weakness 1 & Weakness 3**
>
> We appreciate the reviewer’s detailed feedback on clarity and presentation. We have substantially revised the main paper to improve readability and organization. In particular, we (i) reorganized the structure to make the narrative flow more coherent, (ii) thoroughly proofread the manuscript to correct typos and awkward phrasing, and (iii) simplified or relocated overly formalized parts to keep the main text concise.
>
> We also addressed all line-specific comments raised in Q1:
>
> - **Lines 54–68:** We refined the overall readability and eliminated the potential confusion that may have existed due to unclear wording and explanations.
> - **Lines 114 & 116:** We have corrected typos and revised the sentences for proper grammar ( redundant wording has been removed).
> - **Lines 141–143:** We clarified that the "performance improvement" refers specifically to anomaly detection performance.
> - **Line 148:** We’ve acknowledged the awkward phrasing and have replaced the "overcame disadvantage" with "mitigated the drawback" accordingly.
> - **Section 3 (beginning):** We removed the repetitive content that overlapped with the introduction and streamlined the opening for a cleaner flow.
> - **Definition 3.3:** We appreciate the reviewer’s perspective on Definition 3.3. First, We view this definition as essential for making the notion of "occurrence" precise and reproducible across settings. While the intuition can be described in plain words, a verbal explanation alone does not yield a reproducible criterion for when the phenomenon occurs. Definition 3.3 operationalizes the occurrence of the counterintuitive phenomenon into concrete, domain-agnostic conditions, distinguishing substantial counterintuitive cases from minor likelihood overlaps or intrinsic dataset difficulty. Without such an operational definition, our analysis in the tabular domain would be ill-posed. Second, to address concerns about overcomplication, we simplified the definition into an easy-to-follow operational form and moved the full formal statement to Appendix B.
> - **NF-SLT acronym:** We removed the undefined use of "NF-SLT" in the abstract to avoid introducing the term before its definition.
> - **Line 296:** We revised the sentence and made clearer explanation.
> - **Use of $d$:** We replaced vague references like "the $d$" with standard phrasing such as "the dimension" and ensured $d$ is properly introduced.
> - **Lines 340–361:** We rewrote this part with shorter sentences with clearer transitions to make it easier to follow.
>
> Grateful to your feedback, in addition to your revisions previously suggested, we further conducted a comprehensive review of the paper. Several passages were found insufficiently organized and potentially challenging for readers to follow, and we have made further refinements to enhance clarity accordingly.
>
> For your reference, the ambiguous expressions noted in lines 128, 175, and 366 in the earlier manuscript, together with the opening paragraph of Section 5.2, have been revised to improve readability. Thank you once again for your valuable comments, which have contributed significantly to strengthening the quality of this academic work.

---

> ### Author Response · Authors · 2025-11-23
>
> >**Answer of Weakness 2**
>
> We have reviewed the reference list to identify any preprint citations and updated all such entries to their corresponding published versions in the main paper.
>
> >**Answer of Question 2**
>
> We explain this observation using the feature-correlation perspective developed in Section 5.2.
> Our analysis attributes the counterintuitive phenomenon in CV primarily to raw pixel representations, which are highly correlated and effectively high-dimensional. This yields a low $d$ Ratio and makes likelihood ranking fragile.
>
> Embedding representations, however, lie in a different regime. We estimated the intrinsic dimensions of the ADBench CIFAR-10 and SVHN embeddings using TwoNN, obtaining 23 and 18 respectively, while the ambient embedding dimension is 1000 (smaller than the 3072-dimensional pixel space). Despite the reduced ambient dimension, the embeddings exhibit higher estimated intrinsic dimensionality than the original images, implying a larger $d$ Ratio. This suggests that embedding features are less strongly correlated. Consequently, CV/NLP embeddings provide a favorable regime for likelihood-only tests, and NF-SLT achieves strong performance in this setting. This explanation is consistent with prior observations by [1] that semantic embeddings alleviate the counterintuitive phenomenon.
>
> We have incorporated the reviewer’s valuable feedback into the main text by adding this discussion to Section 5.2.
>
> >**Answer of Question 3**
>
> In the revised manuscript, we now explicitly state the major hyperparameters and training settings in Section 4. To improve reproducibility and address the request for architectural details, we also added a pointer to Appendix F in Section 4, where we provide the full implementation as well as model-specific training and architecture settings.
>
> >**Reference**
>
> [1] Kirichenko, Polina, Pavel Izmailov, and Andrew G. Wilson. "Why normalizing flows fail to detect out-of-distribution data." *Advances in neural information processing systems* 33 (2020): 20578-20589

---

> ### Author Response · Authors · 2025-11-28
>
> Dear Reviewer Bu6W,
>
> We would like to kindly check whether our rebuttal sufficiently clarified your concerns. If any issues remain, we would be glad to provide additional explanation.

---

### Official Review · Reviewer_58vu · 2025-10-25

**Soundness:** 2
**Presentation:** 3
**Contribution:** 3
**Rating:** 4
**Confidence:** 4

**Summary:**

The paper asks whether the likelihood paradox that is well known from image out of distribution settings truly holds for tabular anomaly detection. The authors propose a clear operational definition of a counterintuitive effect based on the relative area under the receiver operating curve gaps against a pool of comparison methods. They study a normalizing flow model with a simple likelihood test on a large collection of tabular datasets and on several vision and language embedding datasets, and they compare against many classical and modern baselines. The main empirical message is that for ordinary tabular anomaly detection, the paradox is rare and that a straightforward flow plus likelihood test is frequently competitive and often strong. To explain why images behave differently, the paper develops a theoretical account in which increasing dimensionality and strong feature correlation tend to shrink the likelihood gap that separates in distribution and out of distribution. Additionally, the analysis is complemented with synthetic studies and an intrinsic dimension survey that suggests that common tabular datasets have an effective dimension closer to the ambient one than typical image data.

I find the overall direction valuable. The study addresses a common confusion in practice. The experimental sweep across many datasets is helpful for practitioners who need a dependable baseline.

At the same time, some choices reduce my confidence in the strength of the conclusions. The formal definition of a counterintuitive effect depends on fixed thresholds and on the exact competitor pool, yet there is no sensitivity analysis to show that the conclusion does not hinge on these choices. Several baselines appear under tuned or restricted, for example one class support vector machines only with a linear kernel, and some stronger recent tabular methods are not included. The paper repeats runs many times but does not report confidence intervals or statistical tests, and for heavily imbalanced problems I would also like to see area under the precision recall curve with uncertainty. The theoretical results rely on independence or near independence assumptions that are not directly checked on real data, while the empirical link through intrinsic dimension uses estimators that are known to be noisy and sensitive to scaling and sample size. The cross domain comparison also leans on experiments with embeddings for images rather than raw pixels in the main table, which weakens the bridge to the classical image paradox. With stronger baseline fairness, sensitivity checks, and uncertainty reporting, the message would be much firmer.

**Strengths:**

The paper addresses a real and timely question and does so with a broad empirical study. The message that tabular anomaly detection does not inherit the image likelihood paradox by default is useful and somewhat surprising to many readers. The definition of what counts as a counterintuitive phenomenon makes the discussion concrete and allows consistent counting across datasets. The theoretical story is appealing and connects intuition with mathematics. Higher dimension and stronger correlation often make pure likelihood less discriminative, and the synthetic ablations support this picture. The writing is mostly clear, tables are compact, and the appendices contain many implementation details that will help reproduction. As a practical takeaway, the flow plus likelihood recipe emerges as a credible line in the sand for tabular anomaly detection.

**Weaknesses:**

The credibility of the main claims would benefit from additional controls. The decision rule that declares a counterintuitive effect uses thresholds and a chosen competitor set. Without a sensitivity study, it is difficult to determine the stability of the conclusion under reasonable alternatives. Baseline coverage and tuning are uneven. Some classical methods are limited to narrow settings, and several stronger tabular anomaly approaches are missing. Hyperparameter search spaces differ across methods, which can favor the proposed approach.

Although the authors repeat experiments multiple times, providing more information on confidence intervals, significance tests, and the area under the precision-recall curve could help assess the general quality, especially for imbalanced datasets.

The theoretical analysis assumes independence or weak dependence for parts of the argument, yet there is no diagnostic that checks these assumptions on the real datasets. The bridge from theory to practice relies on intrinsic dimension estimators that are sensitive to preprocessing.

The work standardizes features but does not report how scaling choices and one hot encodings influence the intrinsic dimension statistics. I am missing a discussion of contamination or near duplicate risks in the benchmark pool, which is relevant for widely reused repositories.

**Questions:**

1) Can you provide a short sensitivity analysis of the counterintuitive definition with several reasonable choices of thresholds and with alternative competitor pools, and report how often the decision flips?

2) Can you include at least one strong one-class support vector machine with a radial basis kernel under a time budget, and add one or two recent tabular anomaly detectors that you excluded, so that the baseline set reflects the current state of the art?

3) Could you report diagnostics that test the independence or weak dependence assumptions on the real datasets, for example, correlation summaries or partial correlation sparsity, and discuss how violations would affect the theory.

4) For the intrinsic dimension link, can you show robustness to different scalings, to sub-sampling, and to an estimator beyond the two nearest neighbor and maximum likelihood ones, and clarify the handling of categorical variables and one-hot encodings

5) Please move at least a small pixel-level image comparison into the main text or summarize it more clearly, so that the reader can judge the cross-domain claim without digging through the appendix.

6) Did you run a near duplicate or contamination check across the dataset pool, and if not, can you comment on this risk

---

> ### Author Response · Authors · 2025-11-23
>
> We are grateful for the reviewer’s valuable suggestions, which helped improve the manuscript.
>
> >**Answer of AUPRC Result and Experimental Information**
>
> To provide more detailed statistical information for each experiment, we have added tables reporting the standard deviation across repeated runs to the Appendix I in Table 20 and 21. We also note that AUPRC results were already included in our paper: the main results table (Table 1) contains an AUPRC column, and dataset- and model-wise AUPRC values are reported in Tables 18 and 19. We kindly ask the reviewer to refer to these tables for a clearer assessment, especially under class-imbalanced settings.
>
> >**Answer of Weakness 1**
>
> We performed a short sensitivity analysis of Definition 3.3 with multiple reasonable threshold choices and alternative competitor pools, and checked how often the decision flips.
>
> As discussed in the latter part of Section 3, the counterintuitive phenomenon is intended to capture cases where a density-estimation model exhibits substantially and unexpectedly worse detection performance compared to baselines (i.e., a large relative performance gap), rather than minor overlap or small drops. To test sensitivity, we set the competitor pool to the full set of comparison models and swept thresholds over $\beta \in \left[\frac{8}{12}, \frac{9}{12}, \frac{10}{12}, \frac{11}{12}, \frac{12}{12}\right]$, $\gamma \in [0.3, 0.4, 0.5, 0.6]$ evaluating all possible $(\beta,\gamma)$ combinations. Across this entire grid, no tabular dataset was classified as counterintuitive under Definition 3.3. Hence, the decision did not flip for any dataset, i.e., the flip frequency is 0% within these tested ranges.
>
> We further repeated the analysis with alternative competitor pools. Specifically, we considered (i) shallow-model competitors only and (ii) deep-model competitors only, adjusting $\beta$ appropriately to reflect the pool size. Under both alternative pools and the same $\gamma$ sweep, we again observed no counterintuitive cases, and therefore no decision flips.
>
> Overall, within a broad set of reasonable thresholds and competitor pools, Definition 3.3 yields a stable decision on our tabular benchmarks, and the flip rate is 0%.
>
> >**Answer of Weakness 2**
>
> We also found the originally reported OCSVM results unexpectedly low, and thus re-implemented One-Class SVM using the official scikit-learn library and re-ran the full benchmark. In doing so, we observed a substantial performance improvement when using the linear kernel, and we updated all related results in the manuscript based on this corrected implementation.
>
> In contrast, when we evaluated OCSVM with the RBF kernel under the same protocol as in our main paper across all 47 tabular datasets, the average AUROC remained extremely low. Concretely, the linear kernel achieved an average AUROC of 0.6562, whereas the RBF kernel achieved only 0.2078. Given this consistent gap, we report OCSVM with the linear kernel as the representative configuration in our baseline pool.
>
> We additionally included the recent state-of-the-art tabular anomaly detector DRL [1] in our experiments. We used the hyperparameters recommended in [1] and implemented the method based on the authors’ released code. While the original implementation appears to use a standard scaler, our benchmark protocol uses scikit-learn’s RobustScaler. To ensure a fair and transparent comparison, we therefore report DRL results under both scalers. The results are summarized below:
>
> | Metric | DeepSVDD | GOAD | NeuTraLAD | ICL | MCM | NF-SLT | DRL (Robust Scaler) | DRL (Standard Scaler) |
> |:---:|:---:|:---:|:---:|:---:|:---:|:---:|:---:|:---:|
> | AUROC | 0.7686 | 0.6086 | 0.8081 | 0.8208 | 0.7864 | **0.8575** | 0.7844 | 0.8217 |
> | AUPRC | 0.5388 | 0.4114 | 0.5594 | 0.6170 | 0.5383 | **0.6398** | 0.5278 | 0.5886 |
>
> Across both scaling choices, NF-SLT remains the strongest performer. This provides empirical evidence that NF-SLT is competitive even relative to a recent SOTA tabular anomaly detector, and we hope these additional results resolve the reviewer’s concern.

---

> > ### Comment · Reviewer_58vu · 2025-11-25
> >
> > Thank you to the authors for clarifying my concerns. In light of the clear responses, I will increase my score.

---

> > > ### Author Response · Authors · 2025-11-26
> > >
> > > Thank you very much for your thoughtful reconsideration and for updating the score.
> > > We sincerely appreciate the time and effort you have devoted to reviewing our work.

---

> ### Author Response · Authors · 2025-11-23
>
> >**Answer of Weakness 3**
>
> To examine the independence (or weak dependence) assumption, we report two real-data evaluations that explicitly alter feature dependence. Table 2 reports anomaly detection results after applying Independent Component Analysis (ICA), which approximates feature-wise independence by transforming the original features into nearly independent components. Table 5 reports results after applying Principal Component Analysis (PCA), which removes linear correlations among features. As shown in Tables 2 and 5, the dimension–performance trends are highly similar. This suggests that even when the independence assumption is not strictly satisfied, the same qualitative theoretical trend persists: datasets that are brought closer to the assumption via ICA and those that remain less aligned with it under PCA still exhibit comparable dimension-dependent behavior.
>
> Moreover, we also evaluate real-world image datasets with strong spatial correlations in Table 3, and observe a dimension–performance trend similar to that in Table 2, suggesting that the same qualitative behavior persists even under substantial dependence violations.
>
> We highlight one nuanced case already discussed in the paper: when reducing image dimension via resizing, the preprocessing can change dependence and entropy. In the CelebA vs. SVHN setting, AUROC can exceed 0.5 after resizing. We believe this is not due to dimensionality reduction alone, but because image resizing methods (e.g., bilinear interpolation) increase local pixel correlations and significantly reduce the entropy of the resized distributions, shifting the data away from the regime assumed in the dimension-only analysis. This explains the apparent deviation without contradicting the core theoretical mechanism.
>
> >**Answer of Weakness 4**
>
> We provide additional robustness checks for the intrinsic-dimension (ID) analysis along four axes: alternative estimators, sub-sampling, feature scaling, and categorical-feature handling.
>
> **(1) Alternative ID estimator beyond TwoNN and MLE**
>
> To examine estimator dependence, we additionally estimate intrinsic dimension using the lPCA method proposed in [2], and compare it to TwoNN and MLE on representative tabular datasets:
>
> | Method | magicgamma | satellite | landsat | waveform | Wilt | annthyroid | breastw | cover | fault | fraud |
> |:---:|:---:|:---:|:---:|:---:|:---:|:---:|:---:|:---:|:---:|:---:|
> | TwoNN | 7 | 15 | 14 | 17 | 5 | 3 | 4 | 3 | 5 | 4 |
> | MLE | 5 | 9 | 11 | 16 | 4 | 3 | 4 | 4 | 5 | 8 |
> | IPCA | 3 | 2 | 2 | 2 | 1 | 1 | 3 | 1 | 5 | 4 |
>
> We observe that IPCA tends to produce smaller ID estimates than TwoNN/MLE, while TwoNN and MLE are relatively consistent with each other. Since our theoretical claims rely on qualitative ID trends rather than any single estimator’s absolute scale, we continue to use TwoNN as the main estimator, supported by cross-estimator consistency.
>
> **(2) Robustness to sub-sampling.**
>
> We estimate ID using TwoNN under different sub-sampling ratios:
>
> | Ratio | magicgamma | satellite | landsat | waveform | Wilt | annthyroid | breastw | fault |
> |:---:|:---:|:---:|:---:|:---:|:---:|:---:|:---:|:---:|
> | 100% | 7 | 15 | 14 | 17 | 5 | 3 | 4 | 5 |
> | 90% | 7 | 15 | 14 | 17 | 5 | 3 | 4 | 5 |
> | 80% | 7 | 14 | 14 | 17 | 5 | 3 | 4 | 5 |
> | 50% | 7 | 14 | 13 | 16 | 5 | 4 | 4 | 4 |
>
> The estimated IDs remain nearly unchanged across ratios, indicating that TwoNN is robust to sub-sampling at the levels tested.
>
> **(3) Robustness to different feature scalings.**
>
> We also evaluate ID sensitivity to scaling:
>
> | Ratio | magicgamma | satellite | landsat | waveform | Wilt | annthyroid | breastw | fault |
> |:---:|:---:|:---:|:---:|:---:|:---:|:---:|:---:|:---:|
> | No Scaler | 7 | 15 | 14 | 17 | 5 | 3 | 4 | 5 |
> | Standard Scaler | 7 | 15 | 14 | 16 | 4 | 5 | 4 | 5 |
> | Minmax Scaler | 7 | 15 | 14 | 17 | 4 | 4 | 5 | 4 | |
>
> TwoNN yields similar IDs regardless of scaling choice, suggesting robustness to different scalings.
>
> **(4) Categorical variables and one-hot encoding.**
>
> Finally, using the Credit Card Client dataset with categorical features from [3], we compare label encoding (default) vs. one-hot encoding. TwoNN estimates the same intrinsic dimension (ID = 6) in both cases, indicating that our ID estimation is not sensitive to the categorical-feature encoding strategy.
>
> Overall, these results show that the intrinsic-dimension link reported in the paper is stable across estimator choice, sub-sampling, scaling, and categorical preprocessing, addressing the reviewer’s concern.

---

> ### Author Response · Authors · 2025-11-23
>
> >**Answer of Weakness 5**
>
> To address reviewer’s concern, we have moved the pixel-level image comparison experiment—showing how AUROC changes with pixel resolution on image datasets—from the appendix into the main text, so that readers can more easily assess the cross-domain claim without needing to consult the appendix.
>
> >**Answer of Weakness 6**
>
> We examined the risk of contamination and near-duplicate anomalous samples within the datasets by conducting sensitivity experiments that explicitly simulate these scenarios.
>
> **(1) Contamination sensitivity**
>
> We injected anomalous samples into the training set at different contamination ratios and evaluated NF-SLT on three representative tabular datasets. The AUROC results are:
>
> | Contamination Ratio | annthyroid | cardio | letter |
> |:---:|:---:|:---:|:---:|
> | 1% | 0.9048 | 0.9032 | 0.9195 |
> | 3% | 0.8806 | 0.8797 | 0.9126 |
> | 5% | 0.8752 | 0.8621 | 0.9005 |
>
> Even when up to 5% anomalous samples contaminate the training data, AUROC changes only modestly, indicating that NF-SLT is robust to a reasonable level of training contamination.
>
> **(2) Duplicated anomalies**
>
> We additionally duplicated anomalous samples within the evaluation sets on the same datasets and observed negligible change in AUROC. This suggests that NF-SLT is not materially affected by duplicated anomalous instances.
>
> Overall, these results indicate that our conclusions are stable under realistic levels of dataset contamination and duplicated anomalies, addressing the reviewer’s concern.
>
> >**Reference**
>
> [1] Ye, Hangting, et al. "DRL: Decomposed Representation Learning for Tabular Anomaly Detection." *The Thirteenth International Conference on Learning Representations*. 2025.
>
> [2] Fan, Mingyu, et al. "Intrinsic dimension estimation of data by principal component analysis." *arXiv preprint arXiv:1002.2050* (2010).
>
> [3] Yeh, I-Cheng, and Che-hui Lien. "The comparisons of data mining techniques for the predictive accuracy of probability of default of credit card clients." *Expert systems with applications* 36.2 (2009): 2473-2480.

---

### Official Review · Reviewer_f1P3 · 2025-10-28

**Soundness:** 3
**Presentation:** 3
**Contribution:** 3
**Rating:** 4
**Confidence:** 3

**Summary:**

This paper investigates why the counterintuitive phenomenon observed in image-domain anomaly detection—where deep generative models assign higher likelihoods to out-of-distribution data than to in-distribution data—rarely occurs in tabular data. The authors propose a formal definition of this phenomenon and conduct extensive experiments on 47 tabular datasets from ADBench using normalizing flows with simple likelihood tests (NF-SLT). They provide both theoretical and empirical explanations focusing on two factors: (1) lower dimensionality, and (2) lower feature correlation in tabular data. The results show that NF-SLT achieves competitive performance among 13 baselines without suffering from the counterintuitive phenomenon.

**Strengths:**

1. The theoretical analysis connecting dimensionality to likelihood gap and AUROC bounds is mathematically rigorous, the d Ratio analysis connecting intrinsic dimension to feature correlation provides valuable insights.

2. A comprehensive set of experiments was conducted, encompassing synthetic Gaussian experiments, dimensionality reduction studies, intrinsic dimension analysis, and performance evaluation on real-world datasets, ensuring the reliability and robustness of the findings.

3. The paper addresses a genuine gap in understanding why likelihood-based anomaly detection behaves differently across domains.

**Weaknesses:**

1. The authors conduct only simple likelihood testing with normalizing flows but provide no improvement strategies for scenarios where their analysis predicts poor performance. Through theoretical (Theorem 5.4, Corollary 5.6) and empirical analyses (Section 5.2, Table 3), the paper establishes that high dimensionality and strong feature correlation lead to SLT failure. However, no solutions are proposed or discussed for such cases.

2. The paper's contribution is weakened by defining a problem specifically to prove its absence. The authors construct Definition 3.3 based on observed characteristics of image-domain failures, then demonstrate that tabular data rarely exhibits this specific pattern. Definition 3.3's thresholds (β, γ) are empirically calibrated rather than theory-derived, and disconnected from Theorem 5.4's absolute failure conditions (entropy/dimension bounds). The paper proves tabular data lacks image-specific failure patterns, not that SLT genuinely succeeds. This reduces to showing "tabular ≠ images" rather than establishing reliability, limiting research contribution.

3. Lack of Failure Case Analysis: The paper briefly mentions "yeast" dataset failure but doesn't deeply analyze when and why NF-SLT fails.

**Questions:**

See Weakness

---

> ### Author Response · Authors · 2025-11-23
>
> >**Answer of Weakness 1**
>
> We appreciate the reviewer’s feedback and we acknowledge the concern on the absence of an explicit solution for image-domain failures. However, we would like to clarify the scope and objective of our work. The motivating flow of this paper is: (i) prior studies show that likelihood-based anomaly detection can fail in the image domain; (ii) we ask whether the same counterintuitive behavior also arises in tabular anomaly detection; (iii) we find that it is rare in tabular settings; and (iv) we investigate why tabular data do not exhibit the same pathology as images. Accordingly, our main contribution is a diagnostic and explanatory study that establishes the reliability of likelihood-only testing in general tabular regimes and provides theoretical and empirical reasons for this domain difference.
>
> Designing a remedy for the image-domain pathology is certainly an important and promising direction. However, the primary aim of this work is to diagnose and explain why likelihood-only testing behaves reliably in general tabular regimes, rather than to propose a new method for image-specific failures.
>
> >**Answer of Weakness 2**
>
> Our work is motivated by the observation in [1] that, in some tabular datasets, the likelihood histograms of normal and anomalous samples overlap. We introduced Definition 3.3 because likelihood inversion or overlap alone is not sufficient to conclude the occurrence of a counterintuitive phenomenon. In anomaly detection, low AUROC  may also arise simply when a dataset is intrinsically difficult, in which case all methods may perform poorly. Therefore, to judge whether the behavior is genuinely counterintuitive, we argue that a comparison against strong baselines must be incorporated. Definition 3.3 operationalizes this idea by labeling a case as counterintuitive only when NF-SLT shows relative low performance—i.e., when it is meaningfully worse than established comparison models (see Section 3).
>
> With this definition in place, we then show empirically that NF-SLT achieves strong detection performance in tabular AD tasks (Table 1). We further explain why likelihood-only testing can succeed in general tabular regimes through two complementary analyses: (i) tabular data typically exhibit weaker feature correlations than images (Section 5.2), and (ii) they have substantially smaller dimension, which alleviates the failure mechanisms predicted by our theory (Section 5.1). Together, these domain characteristics make it plausible for likelihood-based methods to perform competitively—or even outperform other baselines—on tabular data. Consequently, the “Relative Low Performance” condition in Definition 3.3 is rarely satisfied in tabular benchmarks, which is precisely consistent with our theoretical analysis.
>
> Finally, although our investigation was historically motivated by likelihood inversion in the image domain, the criteria in Definition 3.3 are domain-agnostic: they do not encode any image-specific pattern, but rather provide an operational way to determine whether a likelihood-only test like NF-SLT truly fails in the counterintuitive sense, regardless of the modality.
>
> >**Answer of Weakness 3**
>
> We apologize for the confusion caused by our wording in the main paper. We should have more clearly distinguished between (i) a truly counterintuitive case under Definition 3.3 and (ii) a dataset where NF-SLT merely attains relatively low performance. In particular, we described the “yeast” dataset as a “failure case,” which may have suggested that it represents a counterintuitive phenomenon. To avoid this misunderstanding, we have revised the manuscript to state more precisely that NF-SLT shows comparatively lower performance on yeast, rather than labeling it as a definitive counterintuitive failure.
>
> Moreover, we note that our paper does analyze principled failure regimes even within tabular settings. In Appendix C, we consider isotropic and log-concave distributions and show that as dimension increases, likelihood-based detection can become fundamentally infeasible. Although this analysis is validated on synthetic data, it provides a theoretically grounded characterization of when likelihood-only testing may fail in tabular regimes as well.
>
> >**Reference**
>
> [1] Kirichenko, Polina, Pavel Izmailov, and Andrew G. Wilson. "Why normalizing flows fail to detect out-of-distribution data." *Advances in neural information processing systems* 33 (2020): 20578-20589.

---

> ### Author Response · Authors · 2025-11-28
>
> Dear Reviewer f1P3,
>
> We would like to kindly check whether our rebuttal sufficiently clarified your concerns. If any issues remain, we would be glad to provide additional explanation.

---

### Official Review · Reviewer_N87H · 2025-10-31

**Soundness:** 3
**Presentation:** 2
**Contribution:** 1
**Rating:** 2
**Confidence:** 4

**Summary:**

The paper studies the “counterintuitive likelihood” phenomenon in deep generative models, where anomalies sometimes receive higher likelihoods than normal data. While this issue is common in image domains, the authors show it is rare in tabular settings. They formalize a domain-agnostic definition of the phenomenon and evaluate normalizing flows and baselines across 47 tabular and 10 embedding datasets from ADBench. Results indicate that normalizing flow with Simple Likelihood Test is reliable for tabular anomaly detection. The authors further analyzes why the issue is less prominent in tabular data, highlighting the roles of dimensionality and feature-correlation differences.

**Strengths:**

- The empirical study is comprehensive and the analysis is reasonable.

- The finding that normalizing flow with Simple Likelihood Test outperforms other complicated methods is meaningful.

**Weaknesses:**

- The conclusion that normalizing flow with Simple Likelihood Test performs well on tabular data is indeed not surprising. As suggested by Kirichenko et al. (2020) and Schirrmeister et al. (2020), the counterintuitive phenomenon in the image domain is mainly caused by the strong local correlations in images, which is clealry not the case for tabluar data.

- The analysis from the perspective of intrinsic dimensionality and feature correlation is not novel. The impact of feature correlation is studied by Kirichenko et al. (2020) and Schirrmeister et al. (2020), and a deeper analysis of intrinsic dimensionality is provided in [1].

The present paper extends these arguments to a tabular benchmark but does not dramatically advance the theoretical understanding beyond existing literature.

[1] Kamkari, Hamidreza, et al. "A Geometric Explanation of the Likelihood OOD Detection Paradox." International Conference on Machine Learning. PMLR, 2024.

**Questions:**

How sensitive are the results to the particular choice of model architecture (e.g., flow-based vs autoregressive)?

---

> ### Author Response · Authors · 2025-11-23
>
> We appreciate the reviewer’s thoughtful and constructive comments.
>
> >**Answer of Weakness 1**
>
> We agree with the reviewer and prior work that the counterintuitive likelihood behavior in image domains is closely related to strong local feature correlations. However, we respectfully disagree with the implication that our conclusion in the tabular domain is trivial or already settled.
>
> [1] reported that likelihood overlap between ID and OOD could also be observed in tabular datasets and interpreted this as evidence of the counterintuitive. We argue that overlap alone is not a sufficient or well-posed criterion for the counterintuitive phenomenon: if any non-perfect separation were labeled counterintuitive phenomenon, then essentially all realistic detection failures would be called counterintuitive. To avoid this ambiguity, we introduce a domain-agnostic operational definition of the counterintuitive phenomenon (Definition 3.3), which requires (i) relative underperformance of NF-SLT against a pool of strong baselines and (ii) a meaningful performance gap, rather than mere overlap or non-zero error.
>
> Based on this definition, we conduct extensive, unbiased experiments over all 47 tabular AD datasets in ADBench (with 12 comparison methods) and additionally on CV/NLP embedding-based datasets. Empirically, we find that the counterintuitive phenomenon (as defined) is consistently rare in tabular domains, and NF-SLT robustly attains top-tier performance across datasets.
>
> These results establish that our conclusion is not a restatement of intuition, but rather a systematic clarification enabled by (1) a missing formal definition and (2) large-scale validation under that definition.
>
> >**Answer of Weakness 2**
>
> Prior works (e.g., [1], [3]) explain the counterintuitive phenomenon primarily in the image domain, linking failures to strong local correlations and architectural inductive biases. The discussion on tabular data is limited, and [1] interpreted likelihood overlap in a small number of tabular datasets as evidence of pathological behavior. Our work goes beyond this by (i) rigorously redefining the counterintuitive phenomenon (Definition 3.3) and (ii) evaluating its occurrence over the full ADBench tabular suite, using relative performance-based judgment rather than overlap-based interpretation.
>
> [2] provides a geometric explanation for counterintuitive likelihood behavior in images by analyzing local intrinsic dimension (LID) differences between specific ID/OOD pairs, under the assumption that lower LID yields higher likelihood. In contrast, our paper addresses a domain-level question:
>
> - We compare intrinsic-dimension statistics at the benchmark/domain level (tabular vs. image), showing that tabular datasets have systematically larger $d$ Ratio compared to images.
> - We provide a dimension-scaling analysis (Theorem 5.4 and Corollary 5.6), demonstrating that increasing dimension limits the achievable likelihood gap (and thus upper-bounds AUROC).
> - We complement correlation-based analysis (Section 5.2) with this dimension-driven theoretical mechanism (Section 5.1), and link both to the rarity of counterintuitive likelihood ordering in tabular data.
>
> Thus, the novelty is not merely “using intrinsic dimension,” but connecting domain-level dimension/correlation structure to the rarity of counterintuitive likelihood behavior through a new definition, large-scale empirical evidence, and dimension-scaling theory.

---

> ### Author Response · Authors · 2025-11-23
>
> >**Answer of Question 1**
>
> We attempted to run anomaly detection experiments with MAF [4] across all datasets. However, due to numerical stability issues in MAF training, we were unable to obtain reliable convergence for every benchmark.
>
> We therefore report AUROC on the 7 datasets where MAF trained stably. Each AUROC value is the mean over 10 independent runs. All experiments followed exactly the same protocol as in our main paper, using a latent dimension of 128, learning rate 1e-4, batch size 100, and 10 coupling layers. Our implementation was based on the reference code from [5].
>
> | Model | Ionosphere | Lymphography | WPBC | thyroid | yeast | glass | pendigits |
> |:---:|:---:|:---:|:---:|:---:|:---:|:---:|:---:|
> | NICE | **0.9581** | 0.9746 | 0.5051 | **0.9840** | **0.4652** | **0.8867** | **0.9930** |
> | MAF | 0.8406 | **0.9908** | **0.5070** | 0.9752 | 0.4359 | 0.7677 | 0.9926 |
>
> From these results, NICE achieves higher AUROC than MAF on the majority of the tested datasets. This is consistent with Appendix G, supporting our observation suggesting that a more advanced or expressive architecture does not necessarily yield better anomaly detection performance in the tabular domain.
>
> >**Reference**
>
> [1] Kirichenko, Polina, Pavel Izmailov, and Andrew G. Wilson. "Why normalizing flows fail to detect out-of-distribution data." *Advances in neural information processing systems* 33 (2020): 20578-20589.
>
> [2] Kamkari, Hamidreza, et al. "A Geometric Explanation of the Likelihood OOD Detection Paradox." International Conference on Machine Learning. PMLR, 2024.
>
> [3] Schirrmeister, Robin, et al. "Understanding anomaly detection with deep invertible networks through hierarchies of distributions and features." *Advances in Neural Information Processing Systems* 33 (2020): 21038-21049.
>
> [4] Papamakarios, George, Theo Pavlakou, and Iain Murray. "Masked autoregressive flow for density estimation." *Advances in neural information processing systems* 30 (2017).
>
> [5] Stimper, Vincent, et al. "normflows: A PyTorch Package for Normalizing Flows." *Journal of Open Source Software* 8.86 (2023): 5361.

---

> ### Author Response · Authors · 2025-11-28
>
> Dear Reviewer N87H,
>
> We would like to kindly check whether our rebuttal sufficiently clarified your concerns. If any issues remain, we would be glad to provide additional explanation.

---

### Meta-Review · Area_Chair_7apr · 2025-12-29

**Summary:**

The paper studies the counterintuitive likelihood phenomenon of deep generative models for tabular anomaly detection.

Reviewer N87H, who is certain, argued that the paper’s main findings follow naturally from several prior works that established similar observations on image data. The rebuttal provides convincing clarification regarding how the contributions extend these works, particularly through rigorous redefinitions and targeted experimentation in the tabular setting. It is likely that this mitigates the reviewer’s concerns, but unlikely that it completely resolves them, especially because reviewer f1P3 questions exactly these redefinitions, saying that “this reduces to showing tabular != images rather than establishing reliability, limiting research contribution.” This point does not seem to be fully resolved by the rebuttal. On the other hand, reviewer 58vu, who is also certain, finds that “the message that tabular anomaly detection does not inherit the image likelihood paradox by default is useful and somewhat surprising to many readers,” which stands in contrast to the other reviewers. 58vu raises several other concerns, mostly regarding missing empirical evaluations, which have been largely addressed in the rebuttal. The review by Bu6W mostly criticizes the presentation (typos, improper citations, poor organization, ...), which has been addressed.

Overall, the reviewers have doubts about whether the overall approach sufficiently explains the phenomenon on tabular AD beyond just restating what prior work has already found for images. While the rebuttal addresses some of these concerns, it is unlikely that it completely resolves them. While initial reviews have been very negative, the paper appears borderline after rebuttal, but the remaining conceptual concerns argue against acceptance.

**Reviewer Concerns:**

N87H Limited novelty: Partly addressed.

f1P3 No solution proposed: Addressed.

f1P3 Limited novelty: Partly addressed.

f1P3 Lack of failure case analysis: Addressed.

58vu Missing confidence intervals or statistical tests: Addressed.

58vu Unrealistic independence assumptions: Addressed.

58vu Missing sensitivity study: Addressed.

Bu6W Paper hard to follow, typos: Addressed.

Bu6W Paper cites preprints instead of published versions: Addressed.

Bu6W NF-SLT itself is not explained in the main paper: Not addressed.

Bu6W On CV and NLP datasets, the phenomenon is not observed for NF-SLT: Addressed.

**Reviewer Scores:**

N87H: Probably would have raised from 2 to 4.

F1P3: Probably would have stayed with 4.

58vu: Probably would have raised from 4 to 6.

Bu6W: Probably would have raised from 2 to 4.

---

### Decision · Program_Chairs · 2026-01-26

Reject